# GVFi: Learning 3D Gaussian Velocity Fields from Dynamic Videos

## Abstract

In this paper, we aim to model 3D scene geometry, appearance, and physical information just from dynamic multi-view videos in the absence of any human labels. By leveraging physics-informed losses as soft constraints or integrating simple physics models into neural networks, existing works often fail to learn complex motion physics, or doing so requires additional labels such as object types or masks. In this paper, we propose a new framework named **GVFi** to model the motion physics of complex dynamic 3D scenes. The key novelty of our approach is that, by formulating each 3D point as a rigid particle with size and orientation in space, we choose to directly learn a translation rotation dynamics system for each particle, explicitly estimating a complete set of physical parameters to govern the particle's motion over time. Extensive experiments on three existing dynamic datasets and two newly created challenging synthetic and real-world datasets demonstrate the extraordinary performance of our method over baselines in the task of future frame extrapolation. A nice property of our framework is that multiple objects or parts can be easily segmented just by clustering the learned physical parameters. Our datasets and code will be released at https://github.com/.

## 1 Introduction

Regarding our daily dynamic 3D scenes such as falling balls, rotating fans, and folding chairs, precisely modeling their geometry, appearance, and physical properties, and further predicting their future states are crucial for emerging applications in robotics, mixed reality, and embodied AI. With the advancement of recent 3D representations such as NeRF (Mildenhall et al., 2020) and 3DGS (Kerbl et al., 2023), a plethora of works (Pumarola et al., 2021; Yang et al., 2024; Wu et al., 2024) have been proposed to model various dynamic 3D scenes, achieving excellent performance in interpolating novel views within the observed time. However, they often fail to extrapolate future frames, fundamentally because they cannot learn the underlying physics priors of complex 3D scenes.

To learn physics priors, existing methods mainly consist of two categories: 1) physics-informed neural network (PINN) based methods (Raissi et al., 2019) which integrate the governing partial differential equations (PDEs) into loss functions to drive neural networks to learn physically plausible dynamic 3D scenes such as floating smoke (Chu et al., 2022) and simple moving objects (Li et al., 2023b). Although demonstrating promising results in modeling 3D geometry and physics such as velocity and viscosity, these methods usually need boundary constraints such as accurate object/foreground masks which may not always be available in practice. In addition, adding PINN losses is not a free lunch, but significantly sacrificing the efficiency in training and accuracy at boundary regions. 2) Physics model based methods (Jonathan et al., 2020; Zhong et al., 2024; Whitney et al., 2024) which encode various physics systems into neural networks to model elastic objects, fluids, *etc.*. Thanks to the explicit physics priors, these methods obtain impressive results in physical properties learning and simulation. Nevertheless, they are often limited to specific types of objects, materials, or motions due to the lack of generality of encoded physics priors, thus being unable to predict future motions of complex dynamic 3D objects and scenes.

In this paper, we aim to introduce a new framework to model dynamic 3D scenes just from multi-view RGB videos, without needing any additional human labels such as object types or masks, ultimately being able to predict future frames viewing from arbitrary angles. Among various physical properties of a dynamic 3D scene, following the recent work NVFi (Li et al., 2023a), we also

Figure 1: An illustration of the overall framework.

choose to learn a velocity field as it directly governs 3D scene movement. However, to accurately learn the physical velocity from RGB videos is extremely challenging, essentially due to the lack of sufficient physics constraints from raw color pixels. This problem is even harder when multiple objects or parts are undergoing rather different motion patterns. For example, regarding two adjacent objects moving in opposite directions in 3D space, the velocity of neighboring 3D surface points at the intersection region tends to have particularly distinct patterns. This means that the latent representation of per-point dynamics in 3D space could be discrete in nature. Therefore, it is more desirable to model per-point dynamics independently, thus every point having its unique motion. For generality, we regard each 3D point in space as a rigid particle with its size and orientation. If its size is zero, the rigid particle degenerates to a point.

With this insight, for each rigid particle in space, we propose to learn an independent dynamics system that includes a complete set of physical parameters to govern its motion over time. According to the laws of classical mechanics, for a specific rigid particle traversing 3D space over time, its motion can always be regarded as a rotational movement about a rotation center which has its own translation. Given this general foundation, we choose to learn a translation rotation dynamics system for each rigid particle, allowing its future motion to be derived accordingly. Alongside learning the core dynamics, we must also model the geometry and appearance of 3D scenes. In this paper, we naturally choose the recent 3D Gaussian Splatting (3DGS) (Kerbl et al., 2023) as our scene representation, thanks to its unprecedented fidelity in reconstruction and its particle (a Gaussian kernel) based representation in nature, which shares the basic concept of our defined rigid particle.

As illustrated in Figure 1, our framework consists of two major components: 1) a **3D scene representation module** to learn dynamic scene geometry and appearance at a canonical timestamp, which is implemented by a vanilla 3DGS (Kerbl et al., 2023), though other variants can be adopted as well; 2) a **translation rotation dynamics system module** to learn a full set of physical parameters for each input rigid particle, which is just realized by multilayer perceptrons (MLPs). Based on these system parameters, the rigid particle's velocity is then derived according to the laws of classical mechanics, without needing additional physics priors such as PINN (Raissi et al., 2019) in training.

The key to our framework is the second module which simply regards each 3D Gaussian kernel as a rigid particle and takes it as input into MLPs. Nevertheless, we empirically find that it is hard to optimize this module due to the inaccuracy and instability of Gaussian kernels regressed at early training epochs. To tackle this issue, we simply train an auxiliary deformation field in parallel with our second module using an existing work such as (Yang et al., 2024) and (Wu et al., 2024).

Different from current works for modeling dynamic scenes, including NeRF-based methods, *e.g.*, D-NeRF(Pumarola et al., 2021)/TiNeuVox(Fang et al., 2022)/HexPlane(Cao & Johnson, 2023), and 3DGS-based methods such as DefGS (Yang et al., 2024), 4DGS (Wu et al., 2024), and E-D3DGS (Bae et al., 2024), our core novelty is the introduced translation rotation dynamics system together with its effective optimization strategy, which allows us to truly learn physical parameters, ultimately achieving future frame extrapolation. By comparison, all those existing methods fail to do so, though they perform well for past frame interpolation, as extensively verified in Tables 1&2.

Our method, named **GVFi**, leverages 3D **G**aussians to model scene geometry and appearance, while learning **v**elocity **fi**elds via estimating translation rotation dynamics systems. Our contributions are:

- We introduce a new framework to model motion physics of complex dynamic 3D scenes, without needing prior knowledge of object shapes, types, or masks.
- We propose to learn a translation rotation dynamics system for each 3D rigid particle, thus allowing the velocity field to be derived without needing additional physics constraints in training.
- We demonstrate superior results in future frame extrapolation on three existing datasets, and two newly collected synthetic and real-world datasets with extremely challenging dynamics.

## 2 RELATED WORKS

**3D Shape Representations**: Static 3D objects and scenes are traditionally represented by voxels, point clouds, meshes, *etc.*, but they usually have limited representation capabilities due to the nature of discretization. Recently, implicit representations have been developed in the literature, including occupancy fields (OF) (Mescheder et al., 2019; Chen & Zhang, 2019), un/signed distance fields (U/SDF) (Park et al., 2019; Chibane et al., 2020), and radiance fields (NeRF) (Mildenhall et al., 2020). Although demonstrating excellent performance in novel view synthesis and shape reconstruction, they are time-consuming to render 2D images or extract 3D shapes due to the integration of their continuous coordinate-based representations. To tackle this issue, the very recent 3D Gaussian Splatting (Kerbl et al., 2023) turns to represent a 3D shape as a set of explicit Gaussian kernels with various properties, achieving real-time rendering speed. In our framework, we adopt this particle-based representation, as it is amenable to our particle-based physics learning framework.

**Dynamic 3D Reconstruction**: Recent advances in dynamic 3D reconstruction primarily follow the development of static 3D techniques such as SDF, NeRF, and Gaussian Splatting. To model the temporal relationship, existing works (Tretschk et al., 2021; Li et al., 2021; Barron et al., 2021; Gao et al., 2021; Tian et al., 2023; Liu et al., 2023; Cao & Johnson, 2023; Fridovich-Keil et al., 2023; Cai et al., 2022; Fang et al., 2022; Li et al., 2022; Park et al., 2021; You & Hou, 2023; Du et al., 2021; Park et al., 2023; Xian et al., 2021; Wang et al., 2021; Liu et al., 2024b) usually add the time dimension into static 3D representations to learn a motion or deformation field for rigid or deformable objects and scenes. Despite achieving excellent performance in novel view synthesis, especially when integrating 3DGS as the backbone (Wu et al., 2024; Yang et al., 2024; Li et al., 2024; Lei et al., 2024; Lin et al., 2024; Lu et al., 2024), these works can only interpolate 2D views within the observed time, instead of predicting physically meaningful future frames. Basically, this is because the commonly learned motion or deformation field does not encode physics priors in nature, but just fits the correlation between pixels. ==In this paper, the key difference between these works and us is that we separately learn translation rotation dynamics systems for 3D rigid particles, thus enabling us to estimate physically meaningful future frames, whereas they cannot.==

**3D Physics Learning**: To learn various physical properties for 3D objects and scenes, the recent physics-informed neural networks (PINN) (Raissi et al., 2019; Mishra & Molinaro, 2023; Raissi et al., 2020; Hao et al., 2023; Baieri et al., 2023; Chalapathi et al., 2024; Wang et al., 2024; Zhao et al., 2024) are widely applied to convert PDEs into loss functions as soft constraints, driving neural networks to learn physically meaningful targets. However, it is often inefficient to train PINNs due to the large amount of data samples needed to regularize, and the soft constraints are usually not sufficient to obtain satisfactory results. ==In this paper, we do not rely on such inefficient PINN losses to incorporate physics priors to train neural networks.== Another line of works (Qiao et al., 2022; Deng et al., 2023; Xue et al., 2023; Franz et al., 2023; Whitney et al., 2024) integrate explicit physics systems such as springs, graphs, *etc.*, into the learning process to model elastic objects (Zhong et al., 2024; Zhang et al., 2024; Liu et al., 2024a), fluids (Jonathan et al., 2020; Lienen et al., 2024), *etc.*, achieving impressive results in physics learning and simulation. ==In this paper, we also opt to learn physics systems. However, the core difference is that we learn a translation rotation dynamics system which is applicable to common deformation and transformation dynamics, whereas existing works often learn a spring or fluid system only applicable to elastic objects or fluids.==

## 3 GVFI

Our framework mainly comprises two modules together with an auxiliary deformation field to model 3D geometry, appearance, and physics. Given dynamic multi-view RGB videos with known camera poses and intrinsics, the 3D scene representation module aims to learn a set of 3D Gaussian kernels to represent the 3D scene geometry and appearance in a canonical space. The auxiliary deformation field is designed to predict the translation and distortion of each Gaussian kernel given the current training time $t$. For these two components, we simply follow the design of existing works (Kerbl et al., 2023; Yang et al., 2024) briefly elaborated in Section 3.1. Notably, the deformation field alone cannot extrapolate frames beyond the training time. Our core module of the translation rotation dynamics system aims to learn a set of physical parameters for each 3D rigid particle, governing its motion dynamics over time, which is detailed in Section 3.2.

## 3.1 PRELIMINARY

For the input multi-view RGB videos, $T$ represents the greatest timestamp in training and $N$ the total number of cameras. For training stability, we first use all frames $\{I_0^1 \cdots I_0^n \cdots I_0^N\}$ at time $t = 0$ to train a reasonable static 3DGS model as an initialization of the 3D scene geometry and appearance, and then use the remaining frames to jointly optimize our translation and rotation dynamics system and the auxiliary deformation field.

**Canonical 3D Gaussians**: Following the vanilla 3DGS (Kerbl et al., 2023), we employ a set of learnable 3D Gaussian kernels $G_0$ to represent the canonical scene geometry and appearance at $t = 0$. Each kernel is parameterized by a 3D position $x_0$, covariance matrix obtained from quaternion $r_0$, scaling $s_0$, opacity $\sigma$, and color $c$ computed from spherical harmonics (SH). Following prior works (Yang et al., 2024; Wu et al., 2024), we assume the opacity $\sigma$ and color $c$ of each Gaussian will not be updated, but constantly associated with the kernel and transported over time.

Given the $N$ images at timestamp $t = 0$, we either initialize all canonical 3D Gaussian kernels $G_0$ randomly or based on sparse points created by SfM (Schonberger & Frahm, 2016). To train all kernels, we exactly follow the process of 3DGS (Kerbl et al., 2023) by 1) projecting Gaussian kernels into camera space, 2) rendering the projected kernels into image space, and 3) optimizing all kernel parameters via $\ell_1$ and $\ell_{ssim}$ losses used in 3DGS as follows.

$$\underbrace{\left\{\cdots(x_0, r_0, s_0, \sigma, c)\cdots\right\}}_{G_0} \xrightleftharpoons[\ell_1+\ell_{\text{ssim}}]{\text{project+render}} \left\{I_0^1 \cdots I_0^n \cdots I_0^N\right\} \quad (1)$$

**Auxiliary Deformation Field**: To aid the learning of our translation and rotation dynamics system, we leverage an existing deformation field (Yang et al., 2024), but we are also amenable to other deformable Gaussian methods such as 4DGS (Wu et al., 2024), as demonstrated in our experiments in Section 4.1. In particular, the 3D position $x_0$ of each canonical Gaussian kernel and the current timestamp $t$ are fed into an MLP-based deformation network, denoted as $f_{defo}$, directly predicting the corresponding position displacement $\delta x$, and the change of quaternion $\delta r$ and scaling $\delta s$ from timestamp 0 to $t$. All Gaussians $G_t$ at time $t$ can be easily computed as follows, where the operations $\circ$ and $\odot$ follow (Yang et al., 2024).

$$\underbrace{\left\{\cdots\left((x_t = x_0 + \delta x),(r_t = r_0 \circ \delta r),(s_t = s_0 \odot \delta s),\sigma,c\right)\cdots\right\}}_{G_t}, (\delta x, \delta r, \delta s) = f_{defo}(x_0, t) \quad (2)$$

All these deformed Gaussians will be projected and optimized by visual images at timestamp $t$ in a later stage as clarified in Section 3.3, where the deformation net $f_{defo}$ will be optimized from scratch. All details are provided in Appendix A.1 and A.2.

## 3.2 TRANSLATION ROTATION DYNAMICS SYSTEM

This module aims to learn physical parameters that govern the motion of 3D scenes. However, the dynamics of an entire space are extremely complex. Here, we simplify this problem and formulate it into just learning per rigid particle dynamics, where we treat each (canonical or deformed) Gaussian kernel as a rigid particle with size and orientation. According to the laws of classical mechanics, for a specific rigid particle $P \in \mathcal{R}^3$, its motion in a 3D world coordinate system can be regarded as a rotational movement about a rotation center which has its own translation. To this end, we aim to learn the following two groups of physical parameters for each 3D rigid particle $P$:

- Group #1 - Rotation Center Parameters including: 1) the center's position $P_c \in \mathcal{R}^3$, 2) the center's velocity $v_c \in \mathcal{R}^3$, and 3) acceleration $a_c \in \mathcal{R}^3$ in the world coordinate system.
- Group #2 - Rigid Particle Rotational Parameters including: 1) the rigid particle's rotation vector $w_p \in \mathcal{R}^3$ with regard to its center $P_c$, and 2) the rigid particle's angular acceleration $\epsilon_p$.

As illustrated in Figure 2, our translation rotation dynamics system module, denoted as $f_{trd}$, takes a rigid particle $P$ as input, directly predicting the physical parameters of its rotation center and its own rotational information. Then, this rigid particle will be naturally driven by its learned physical parameters, forming its motion dynamics, as illustrated by the trajectory in Figure 2. This module is implemented by simple MLPs:

$$\{(P_c, v_c, a_c),(w_p, \epsilon_p)\} = f_{trd}(P) \quad (3)$$

Figure 2: The proposed translation rotation dynamics system for a specific rigid particle. The rigid particle will be driven by its learned physical parameters over time, forming a trajectory in 3D space.

Notably, for an input rigid particle $\boldsymbol{P}$, the elegance of this module $f_{trd}$ is that it only needs to learn this full translation rotation dynamics system at its canonical timestamp, *i.e.*, $t = 0$, and that particle's future motion will be governed by the learned dynamics system when $t > 0$ according to the laws of mechanics. We will now derive the rigid particle's future motion as follows.

For a specific rigid particle $\boldsymbol{P}$, now we have its estimated translation rotation dynamics parameters $\{(\boldsymbol{P}_c, \boldsymbol{v}_c, \boldsymbol{a}_c), (\boldsymbol{w}_p, \epsilon_p)\}$. Given a future timestamp $t$, we now calculate its updated parameters for both the rotation center and the particle itself as follows:

$$\boldsymbol{P}_c^t = \boldsymbol{P}_c + \boldsymbol{v}_c t + \frac{1}{2}\boldsymbol{a}_c t^2, \quad \boldsymbol{v}_c^t = \boldsymbol{v}_c + \boldsymbol{a}_c t, \quad \boldsymbol{a}_c^t = \boldsymbol{a}_c, \quad \boldsymbol{w}_p^t = (\|\boldsymbol{w}_p\| + \epsilon_p t)\frac{\boldsymbol{w}_p}{\|\boldsymbol{w}_p\|}, \quad \epsilon_p^t = \epsilon_p \quad (4)$$

Theoretically, our above updating scheme can be naturally extended to higher orders or reduced to lower orders with regard to future time $t$. Intuitively, a higher order relationship from time 0 to $t$ is expected to capture extremely complex dynamics such as a rolling ball suddenly breaking up into pieces due to unknown explosives inside, whereas a much lower order relationship tends to only capture static or constant speed scenes, thus being oversimplified. In this paper, we opt to the above second-order scheme to update dynamics parameters from time 0 to $t$ for two primary reasons:

- In many applications such as robot manipulation, the need for future prediction usually involves a relatively short interval, *i.e.*, $|t - 0|$ is rather small, *e.g.*, in milliseconds. In this case, a second-order relationship is usually sufficient to achieve decent approximations. In addition, a simple sliding window based approach can be applied to continuously and incrementally predict future frames given the newest visual observations from sensors.

- In our daily life, the majority of common physical movements such as rolling balls or moving cars can be generally described by a second-order relationship. In fact, both Newton's First and Second Law of Motion can be captured. Notably, since the whole 3D scene comprises a large number of rigid particles, each particle has up to second-order dynamics, *i.e.*, with a constant acceleration between $0 \sim t$. Therefore, the compounded dynamics for the entire 3D scene can be rather complex, including various deformations and transformations in our daily lives.

Nevertheless, it is still interesting yet non-trivial to learn much higher-order relationships and we leave it for future exploration. More implementation details of this module are in Appendix A.3.

## 3.3 TRAINING

With our translation rotation dynamics module and the auxiliary deformation field, we now discuss how to connect and train them together, such that physical parameters can be truly learned.

For two timestamps $t'$ and $t$, where $t$ is usually sampled from the training set and $\Delta t = t - t'$ is predefined to be small enough, we can easily obtain Gaussians $G_{t'}$ from the deformation field $f_{defo}$.

Having our translation rotation dynamics module $f_{trd}$ at hand, we naturally regard the transportation of all kernels from $t'$ to $t$ is governed by the corresponding physical parameters estimated by $f_{trd}$ at time $t'$. From Equations 4, at time $t'$, the physical parameters of a rigid particle $\boldsymbol{P}$ are:

$$\{(\boldsymbol{P}_c^{t'}, \boldsymbol{v}_c^{t'}, \boldsymbol{a}_c^{t'}), (\boldsymbol{w}_p^{t'}, \epsilon_p^{t'})\} \xleftarrow{t'} \{(\boldsymbol{P}_c, \boldsymbol{v}_c, \boldsymbol{a}_c), (\boldsymbol{w}_p, \epsilon_p)\} = f_{trd}(\boldsymbol{P}) \quad (5)$$

Now we can easily compute the kernel's orientation change $\Delta \boldsymbol{r}$ from $\boldsymbol{r}_{t'}$ to $\boldsymbol{r}_t$ as follows:

$$\Delta \boldsymbol{r} = (\cos\frac{\Delta\theta}{2}, \sin\frac{\Delta\theta}{2} \cdot \frac{\boldsymbol{w}_p}{\|\boldsymbol{w}_p\|}), \quad \text{where } \Delta\theta = (\|\boldsymbol{w}_p\| + \epsilon_p t')(t - t') \quad (6)$$

Table 1: Quantitative results of all methods for both future frame extrapolation and novel view interpolation on Dynamic Object Dataset and Dynamic Indoor Scene Dataset.

| | Dynamic Object Dataset | | | | | | Dynamic Indoor Scene Dataset | | | | | |
| | Interpolation | | | Extrapolation | | | Interpolation | | | Extrapolation | | |
| | PSNR↑ | SSIM↑ | LPIPS↓ | PSNR↑ | SSIM↑ | LPIPS↓ | PSNR↑ | SSIM↑ | LPIPS↓ | PSNR↑ | SSIM↑ | LPIPS↓ |
|---|---|---|---|---|---|---|---|---|---|---|---|---|
| T-NeRF(Pumarola et al., 2021) | 13.163 | 0.709 | 0.353 | 13.818 | 0.739 | 0.324 | 24.944 | 0.742 | 0.336 | 22.242 | 0.700 | 0.363 |
| D-NeRF(Pumarola et al., 2021) | 14.158 | 0.697 | 0.352 | 14.660 | 0.737 | 0.312 | 25.380 | 0.766 | 0.300 | 20.791 | 0.692 | 0.349 |
| TiNeuVox(Fang et al., 2022) | 27.988 | 0.960 | 0.063 | 19.612 | 0.940 | 0.073 | 29.982 | 0.864 | 0.213 | 21.029 | 0.770 | 0.281 |
| T-NeRF$_{PINN}$ | 15.286 | 0.794 | 0.293 | 16.189 | 0.835 | 0.230 | 16.250 | 0.441 | 0.638 | 17.290 | 0.477 | 0.618 |
| HexPlane$_{PINN}$ | 27.042 | 0.958 | 0.057 | 21.419 | 0.946 | 0.067 | 25.215 | 0.763 | 0.389 | 23.091 | 0.742 | 0.401 |
| NSFF(Li et al., 2021) | - | - | - | - | - | - | 29.365 | 0.829 | 0.278 | 24.163 | 0.795 | 0.289 |
| NVFi(Li et al., 2023a) | 29.027 | 0.970 | 0.039 | 27.594 | 0.972 | 0.036 | 30.675 | 0.877 | 0.211 | 29.745 | 0.876 | 0.204 |
| DefGS(Yang et al., 2024) | 37.865 | 0.994 | 0.007 | 19.849 | 0.949 | 0.045 | 29.926 | 0.916 | 0.130 | 21.380 | 0.819 | 0.188 |
| DefGS$_{nvfi}$ | 37.316 | 0.994 | 0.008 | 28.749 | 0.984 | 0.013 | 30.170 | 0.915 | 0.133 | 31.096 | 0.945 | 0.077 |
| E-D3DGS(Bae et al., 2024) | 28.075 | 0.963 | 0.049 | 18.526 | 0.923 | 0.087 | 29.267 | 0.874 | 0.222 | 20.374 | 0.772 | 0.307 |
| 4DGS(Wu et al., 2024) | 37.285 | 0.986 | 0.020 | 20.354 | 0.950 | 0.052 | 29.381 | 0.889 | 0.212 | 21.107 | 0.793 | 0.274 |
| GVFi$_{4dgs}$ (Ours) | 35.961 | 0.985 | 0.021 | 28.316 | 0.978 | 0.023 | 27.932 | 0.860 | 0.252 | 31.590 | 0.909 | 0.194 |
| GVFi (Ours) | 38.788 | 0.995 | 0.006 | 28.758 | 0.982 | 0.011 | 32.202 | 0.928 | 0.089 | 34.556 | 0.964 | 0.046 |

Table 2: Quantitative results of all methods for future frame extrapolation optionally with novel view interpolation on NVIDIA Dynamic Scene Dataset, Dynamic Multipart Dataset, and GoPro Dataset.

| | NVIDIA Dynamic Scene Dataset | | | | | | Dynamic Multipart Dataset | | | | | | GoPro Dataset | | |
| | Interpolation | | | Extrapolation | | | Interpolation | | | Extrapolation | | | Extrapolation | | |
| | PSNR↑ | SSIM↑ | LPIPS↓ | PSNR↑ | SSIM↑ | LPIPS↓ | PSNR↑ | SSIM↑ | LPIPS↓ | PSNR↑ | SSIM↑ | LPIPS↓ | PSNR↑ | SSIM↑ | LPIPS↓ |
|---|---|---|---|---|---|---|---|---|---|---|---|---|---|---|---|
| T-NeRF(Pumarola et al., 2021) | 23.078 | 0.684 | 0.355 | 21.120 | 0.707 | 0.358 | 9.833 | 0.567 | 0.550 | 10.064 | 0.576 | 0.537 | - | - | - |
| D-NeRF(Pumarola et al., 2021) | 22.827 | 0.711 | 0.309 | 20.633 | 0.709 | 0.327 | 13.279 | 0.747 | 0.378 | 13.344 | 0.767 | 0.340 | - | - | - |
| TiNeuVox(Fang et al., 2022) | 28.304 | 0.868 | 0.216 | 24.556 | 0.863 | 0.215 | 29.957 | 0.966 | 0.067 | 20.804 | 0.923 | 0.090 | 20.323 | 0.738 | 0.318 |
| T-NeRF$_{PINN}$ | 18.443 | 0.597 | 0.439 | 17.975 | 0.605 | 0.428 | - | - | - | - | - | - | - | - | - |
| HexPlane$_{PINN}$ | 24.971 | 0.890 | 0.281 | 24.473 | 0.818 | 0.279 | - | - | - | - | - | - | - | - | - |
| NVFi(Li et al., 2023a) | 27.138 | 0.844 | 0.231 | 28.462 | 0.876 | 0.214 | 27.516 | 0.960 | 0.052 | 25.235 | 0.955 | 0.046 | 19.879 | 0.736 | 0.415 |
| DefGS(Yang et al., 2024) | 26.662 | 0.893 | 0.127 | 24.240 | 0.895 | 0.140 | 34.635 | 0.990 | 0.019 | 20.664 | 0.930 | 0.067 | 21.193 | 0.842 | 0.185 |
| DefGS$_{nvfi}$ | 26.972 | 0.890 | 0.128 | 27.529 | 0.927 | 0.102 | 34.637 | 0.990 | 0.018 | 28.455 | 0.979 | 0.017 | 25.469 | 0.882 | 0.141 |
| E-D3DGS(Bae et al., 2024) | 20.848 | 0.541 | 0.532 | 20.301 | 0.565 | 0.522 | 26.180 | 0.955 | 0.062 | 18.615 | 0.904 | 0.114 | - | - | - |
| 4DGS(Wu et al., 2024) | 19.411 | 0.462 | 0.532 | 22.510 | 0.703 | 0.408 | 37.021 | 0.992 | 0.014 | 20.564 | 0.935 | 0.067 | - | - | - |
| GVFi$_{4dgs}$ (Ours) | 18.995 | 0.448 | 0.544 | 22.706 | 0.714 | 0.400 | 36.542 | 0.991 | 0.015 | 30.801 | 0.983 | 0.016 | - | - | - |
| GVFi (Ours) | 26.943 | 0.891 | 0.102 | 29.388 | 0.938 | 0.067 | 34.807 | 0.991 | 0.011 | 30.721 | 0.986 | 0.012 | 26.276 | 0.890 | 0.131 |

Then, we compute the kernel's position translation $\Delta \boldsymbol{x}$ from $\boldsymbol{x}_{t'}$ to $\boldsymbol{x}_t$, which consists of two parts: 1) the translation of its rotation center, and 2) the displacement caused by the kernel's rotation with regard to its center. In particular, they are:

$$\Delta \boldsymbol{x} = \left[ \boldsymbol{v}_c^{t'}(t - t') + \frac{1}{2}\boldsymbol{a}_c^{t'}(t - t')^2 \right] + \left[ (\Delta \mathbf{R} - \boldsymbol{I})(\boldsymbol{x}_{t'} - \boldsymbol{P}_c^{t'}) \right] \tag{7}$$

where $\Delta \mathbf{R}$ is a $3 \times 3$ rotation matrix converted from quaternion $\Delta \boldsymbol{r}$. Since the rotation change $\Delta \boldsymbol{r}$ will update both the orientation and position of a Gaussian kernel, so it is also used in Equation 7.

With the above relationships, we optimize all learnable parameters of canonical Gaussian kernels $G_0$, the deformation field $f_{defo}$ and our translation rotation dynamics module $f_{trd}$ as follows:

- Step #1: We sample two close timestamps $t'$ and $t$, where $t$ is a timestamp appears in training dataset and $t'$ can be greater or smaller than $t$, but $\Delta t = |t - t'|$ is appropriately small.
- Step #2: We get $\{(\boldsymbol{P}_c^{t'}, \boldsymbol{v}_c^{t'}, \boldsymbol{a}_c^{t'}, \boldsymbol{w}_p^{t'}, \epsilon_p^{t'}), (\boldsymbol{P}_c, \boldsymbol{v}_c, \boldsymbol{a}_c, \boldsymbol{w}_p, \epsilon_p)\} \leftarrow f_{trd}(\boldsymbol{P})$, and then calculate $\Delta \boldsymbol{x}$ and $\Delta \boldsymbol{r}$ based on $(\boldsymbol{x}_{t'}, \boldsymbol{r}_{t'}, \boldsymbol{s}_{t'}, \sigma, \boldsymbol{c}) \leftarrow f_{defo}(\boldsymbol{x}_0, t')$, according to Equations 6&7.
- Step #3: We obtain the kernel information at time $t$: $(\boldsymbol{x}_{t'} + \Delta \boldsymbol{x}, \boldsymbol{r}_{t'} \circ \Delta \boldsymbol{r}, \boldsymbol{s}_{t'}, \sigma, \boldsymbol{c})$ which is transported by our physicals parameters from time $t'$, where $\circ$ represents quaternion multiplication.
- Step #4: Lastly, we render all the above kernels at time $t$ to 2D image space following 3DGS, comparing with the training images at time $t$. All parameters are supervised by $\ell_1$ and $\ell_{ssim}$:

$$(\boldsymbol{G}_0, f_{defo}, f_{trd}) \longleftarrow (\ell_1 + \ell_{ssim}) \tag{8}$$

## 4 EXPERIMENTS

**Datasets:** Our method is designed to learn meaningful physical information of 3D dynamic scenes, aiming at accurately predicting future motions, instead of just fitting observed video frames. In this regard, the closest work to us is the recent NVFi (Li et al., 2023a). Following NVFi, we primarily evaluate our method on its three dynamic datasets: **1) Dynamic Object dataset**. It consists of 6 dynamic objects. Each object displays a unique motion pattern belonging to either rigid or deformable movement. **2) Dynamic Indoor Scene dataset**. It has 4 complex indoor scenes. Each scene has multiple objects undergoing different rigid body motions. **3) NVIDIA Dynamic Scene dataset** (Yoon et al., 2020). It consists of two real-world dynamic 3D scenes.

Upon a closer look at the above three datasets, we find that their dynamics captured are relatively simple. In our daily life, the majority of objects and scenes consist of multiple parts undergoing radically different motions over time, showing extremely challenging physical patterns to learn. To further evaluate the effectiveness of our design, we collect a new synthetic dataset, named **4) Dynamic Multipart dataset**, and a new real-world dataset by 20 GoPros, named **5) GoPro dataset**.

Our new synthetic dataset comprises 4 objects. Each has 2 to 5 distinct motion patterns on different object parts. Following (Li et al., 2023a), for each object, we collect RGBs at 15 different viewing angles over 1 (virtual) second after normalization, where each viewing angle has 60 frames captured. We reserve the first 46 frames at randomly picked 12 viewing angles as the training split, *i.e.*, 552 frames, while leaving the 46 frames at the remaining 3 viewing angles for testing *interpolation* ability, *i.e.*, 138 frames for novel view synthesis within the training time period, and keeping the last 14 frames at all 15 viewing angles for evaluating future frame *extrapolation*, *i.e.*, 210 frames.

Our new real-world dataset captures 4 dynamic scenes with 20 GoPro cameras. For each dynamic scene, we select 89 frames from each view, and resize images to be a resolution of $960 \times 540$. We reserve the first 67 frames at 17 picked viewing angles as the training split, *i.e.*, 1139 frames, while leaving the 67 frames at the remaining 3 viewing angles for evaluating *novel view interpolation* within the training time period, *i.e.*, 201 frames. We keep the last 22 frames at all 20 viewing angles for evaluating *future frame extrapolation*, *i.e.*, 440 frames in total. More details are in Appendix A.6.

**Baselines:** We select the following baselines: **1) NVFi** (Li et al., 2023a): This is the closest work to us, but differs from us in two folds. First, NVFi relies on PINN losses to learn physics priors, but we directly learn physical parameters. Second, NVFi adopts NeRF as a backbone, being short in 3D scene geometry and appearance modeling, but our method is amenable to and adopts the powerful 3DGS in nature. **2) T-NeRF** (Pumarola et al., 2021). **3) D-NeRF** (Pumarola et al., 2021). **4) NSFF** (Li et al., 2021). **5) TiNeuVox** (Fang et al., 2022). The latter four methods are based on NeRF and designed for novel view interpolation. Therefore they are expected to be rather weak for future frame extrapolation. For a fair and extensive comparison, we also include the following two baselines. **6) DefGS** (Yang et al., 2024), **7) 4DGS** (Wu et al., 2024), and **8) E-D3DGS** (Bae et al., 2024). These very recent deformable 3D Gaussians methods are particularly strong to model dynamic 3D scenes for novel view synthesis using 3DGS as a backbone. **9) DefGS**$_{nvfi}$. We build this baseline by combining DefGS with the velocity field proposed by NVFi. This baseline has the powerful 3DGS as a backbone as well as the current state-of-the-art NVFi learning strategy. It is trained with exactly the same settings as our method. To demonstrate the flexibility of our framework, we also adopt 4DGS as our auxiliary deformation field, denoted as GVFi$_{4dgs}$.

**Metrics:** The standard metrics **PSNR**, **SSIM**, and **LPIPS** are reported for RGB view synthesis in two tasks: interpolation and future frame extrapolation.

### 4.1 MAIN RESULTS OF FUTURE FRAME EXTRAPOLATION

All methods are trained in a scene-specific fashion. Since NSFF (Li et al., 2021) is not suitable for white-background images, it is not compared on our new Dynamic Multipart dataset. When training our method, we set $\Delta t$ to be 2 divided by the training set frame rate. The time $t'$ for our auxiliary deformation field $f_{defo}$ is set to be 0.7 in the extrapolation task, and target time $t$ is chosen as the frame time. The time difference $\Delta t$ is dynamically computed.

Our primary goal is to extrapolate meaningful future frames as a continuum of the last training observations. In our evaluation, we follow Steps #1∼#4 in Section 3.3 to extrapolate future frames from the last timestamp of training frames. For benchmarking, we also compare novel view (past frame) interpolation with baselines, but this is less important to us. Particularly, we also follow Steps #1∼#4 to render past frames. Though this can also be achieved by progressively querying our translation rotation dynamics module $f_{trd}$, it is inferior due to accumulated errors as detailed in Appendix A.9.

**Results & Analysis:** Tables 1&2 compare all methods on the five datasets. It can be seen that:

• Compared with NeRF and 3DGS based dynamic scene modeling methods such as T-NeRF/ D-NeRF/ TiNeuVox/ NSFF/ DefGS/ 4DGS/ E-D3DGS, both versions of our method achieve about 10 points higher on PSNR for future frame extrapolation. This means that, without explicitly

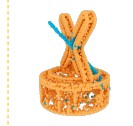 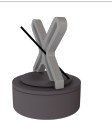 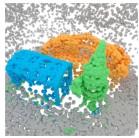 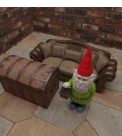

| Dynamic Object Dataset | Dynamic Multipart Dataset | Dynamic Indoor Scene Dataset |

Figure 3: Qualitative results of clustering translation rotation physics parameters. Gaussian kernels can be autonomously grouped into meaningful objects or parts according to their motion patterns.

learning physical information like us, these dynamic methods completely fail to predict the future, highlighting the core value of our method.

- Compared with the closest and also strongest baselines NVFi/ DefGS$_{nvfi}$, our method is still constantly better than them on all datasets for future frame extrapolation. Notably, on the Dynamic Indoor Scene dataset and our newly collected Dynamic Multipart dataset, there are much more complex motion dynamics such as different objects or parts moving in distinct directions, but our best results are constantly about 3 points higher on PSNR than them. Fundamentally, this is because both NVFi and DefGS$_{nvfi}$ rely on PINN losses as soft constraints to incorporate physics priors, whereas we directly integrate hard physics by learning translation rotation dynamics system parameters, thus being more effective in learning dynamics.

- Lastly, our framework is indeed amenable to existing deformation fields such as DefGS and 4DGS, and both versions achieve good results for future frame extrapolation on most datasets.

Figure 4 shows qualitative results. More results of the total 5 datasets are in Appendix A.10 / A.11 / A.12 / A.13/A.14. We also report the training/test time, memory cost, *etc.*, in Appendix A.4.

## 4.2 ANALYSIS OF DYNAMICS PARAMETERS

Our core translation rotation dynamics system module is designed to learn per rigid particle's physical parameters. Ideally, for those rigid particles undergoing the same motion pattern such as all surface points of a single rigid part, they should have the same or similar physical parameters. Given this, multi-

Table 3: Quantitative results of motion segmentation results on Dynamic Indoor Scene dataset.

| | AP↑ | PQ↑ | F1↑ | Pre↑ | Rec↑ | mIoU↑ |
|---|---|---|---|---|---|---|
| M2F(Cheng et al., 2022) | 65.37 | 73.14 | 78.29 | 94.83 | 68.88 | 64.42 |
| D-NeRF(Pumarola et al., 2021) | 57.26 | 46.15 | 59.02 | 56.55 | 62.94 | 46.58 |
| NVFi(Li et al., 2023a) | 91.21 | 78.74 | 93.75 | 93.76 | 93.74 | 67.64 |
| DefGS(Yang et al., 2024) | 51.73 | 57.60 | 66.43 | 63.21 | 70.07 | 54.46 |
| DefGS$_{nvfi}$ | 55.26 | 62.75 | 69.83 | 69.39 | 72.91 | 56.82 |
| **GVFi (Ours)** | **95.82** | **93.28** | **97.90** | **96.21** | **99.86** | **79.55** |

ple dynamic objects or parts with distinct motions can be automatically segmented based on the similarity of learned physical parameters. By comparison, the prior work NVFi (Li et al., 2023a) can hardly achieve this autonomous dynamic segmentation by its own design, unless an external motion grouping method is applied. To further evaluate this nice property of our method, we conduct the following steps to analyze the learned dynamics parameters.

First, after training our method on the Dynamic Object dataset, Dynamic Multipart dataset, and Dynamic Indoor Scene dataset, for each dynamic scene, we have a set of well-trained canonical 3D Gaussians, an auxiliary deformation field, and our translation rotation dynamics parameters.

Then, we use the auxiliary deformation field $f_{defo}$ to deform the canonical 3D Gaussians $G_0$ to time $t = 0.7$, which is the maximum time the deformation field can query in our training. At this timestamp, the motions of different objects and parts normally achieve a steady state.

Lastly, we query all the physical parameters at this time, *i.e.*, $\{(P_c^t, v_c^t, a_c^t), (w_p^t, \epsilon_p^t)\}$. We choose $(\|v_c^t\|, v_c^t/\|v_c^t\|, \|w_p^t\|, w_p^t/\|w_p^t\|)$ as the features to cluster Gaussian particles via a simple K-means algorithm. As shown in Figure 3, all Gaussian particles can be grouped into physically meaningful objects or parts according to their actual motion patterns. More results are in Appendix A.17.

We further quantitatively evaluate our motion grouping results on Dynamic Indoor Scene Dataset. In particular, we follow Gaussian Grouping (Ye et al., 2024) to render 2D object segmentation masks for all 30 views over 60 timestamps on all 4 scenes, *i.e.*, 7200 images in total. We compare with **D-NeRF**, **NVFi**, **DefGS** and **DefGS**$_{nvfi}$. We follow NVFi to obtain segmentation results of D-NeRF and NVFi. For the 3DGS-based baselines, we also adopt OGC (Song & Yang, 2022) to segment Gaussians based on scene flows induced from their learned deformation fields. All imple-

mentation details are in Appendix. Additionally, we include a strong image-based 2D object segmentation method, Mask2Former (Cheng et al., 2022) pre-trained by human annotations on COCO dataset (Lin et al., 2014) as a fully-supervised baseline.

As shown in Table 3, our method achieves almost perfect object segmentation results on all metrics, significantly outperforming all baselines. This shows that our learned physical parameters correctly model object physical motion patterns and can be easily leveraged to identify individual objects according to their motions, without needing any human annotations.

### 4.3 ABLATION STUDY

Our framework mainly comprises 3DGS as the backbone and our core translation rotation dynamics system module, together with an auxiliary deformation field. To verify different choices of our method, we conduct the following three groups of ablation experiments.

**(1) Different choices of time difference $\Delta t$ in training stage**: Given the time interval between two consecutive frames in the training set as $\delta t$, we compare three choices of the time difference $\Delta t$ in training stage: $\{\delta t, 2\delta t, 3\delta t\}$. We choose $\Delta t = 2\delta t$ in our main experiments.

**(2) Removing the auxiliary deformation field $f_{defo}$**: In particular, we feed the Gaussian particles at time $t' = 0$ directly into $f_{trd}$, and use the output physical parameters to directly move particles to a target future timestamp $t$. Note that, here $\Delta t$ is meaningless.

**(3) Learning time-dependent physical parameters at $t'$:** Instead of using Equation 4 to derive physical parameters at time $t'$, we directly learn them by a 6-layer MLPs as: $f_{\mathrm{trd}'}(\boldsymbol{P}, t')$. Theoretically, such a complex function can learn higher order relationships to approximate arbitrary motions than our second-order Equation 4. Nevertheless, it would be more challenging to learn and unable to guarantee physical parameters of the same motion pattern to be consistent over time.

**Results & Analysis**: Table 4 shows all ablation results for future frame extrapolation on our new Dynamic Multipart dataset. It can be seen that: 1) The greatest impact is caused by the removal of the deformation field $f_{defo}$. Although this deformation field itself is unable to learn physics, it significantly aids our core translation rotation dynamics system module to learn physical parameters given the motion information. 2) The choice of time difference $\Delta t$ is also important. Once it is as small as the interval between two consecutive frames, the performance

Table 4: Quantitative results of ablation studies on Dynamic Multipart dataset.

| | $f_{defo}$ | $f_{trd}$ | Extrapolation | | |
| | | | PSNR↑ | SSIM↑ | LPIPS↓ |
|---|---|---|---|---|---|
| (1) $\delta t$ | ✓ | ✓ | 29.441 | 0.984 | 0.013 |
| (1) $2\delta t$ | ✓ | ✓ | **30.721** | **0.986** | **0.012** |
| (1) $3\delta t$ | ✓ | ✓ | 30.246 | 0.985 | **0.012** |
| (2) - | ✗ | ✓ | 27.081 | 0.981 | 0.018 |
| (3) $2\delta t$ | ✓ | ✗ | 29.986 | 0.985 | **0.012** |

drops apparently, because the motion in short intervals is too subtle to be distinguished. For example, a rotation may be learned as a translation. However, if $\Delta t$ is too large, the appearance fitting could be sacrificed, so the performance is slightly weaker. 3) If physical parameters are learned but not derived, the lack of physics consistency will influence motion learning.

Detailed ablation settings and results are in Appendix A.7. More ablations of $\Delta t$ on four datasets, and more ablations of using 1st-/ 2nd-order relationships in Equation 4 are in Appendix A.8.

## 5 CONCLUSION

In this paper, we have demonstrated that complex motion dynamics can be explicitly learned just from multi-view RGB videos without needing additional human labels such as object types and masks. This is achieved by a new generic framework that simultaneously models 3D scene geometry, appearance and physics by extending the appealing 3D Gaussian Splatting technique. In contrast to existing works which usually rely on PINN losses as soft constraints to learn physics priors, we instead directly learn a complete set of physical parameters to govern the motion pattern of each 3D rigid particle in space via our core translation rotation dynamics system module. Extensive experiments on three public dynamic datasets and a newly created dynamic multipart dataset have shown the extraordinary performance of our method in the challenging task of future frame extrapolation over all baselines. In addition, the learned physical parameters can be directly used to segment objects or parts according to the similarity of parameters.

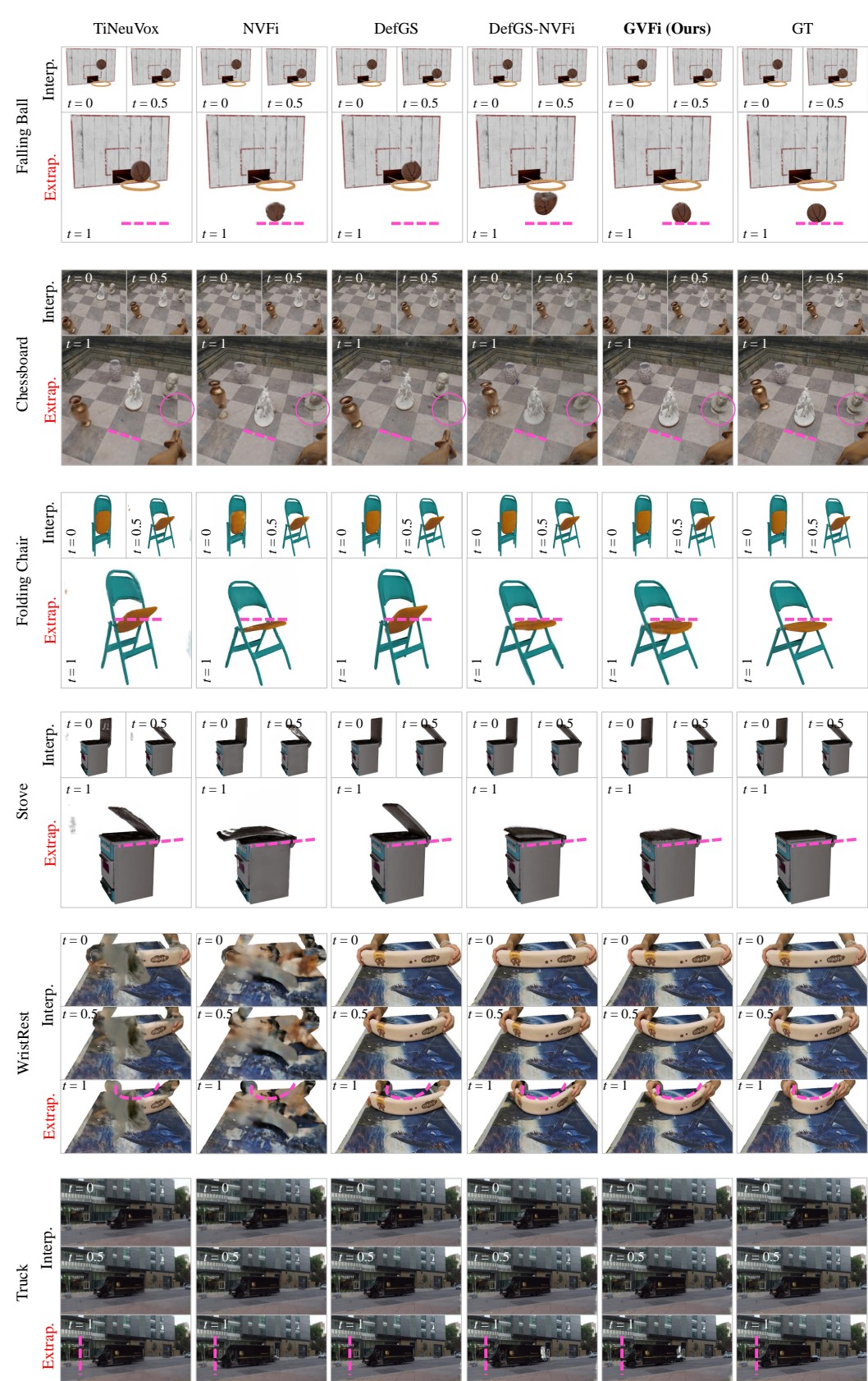

Figure 4: Qualitative results of RGB view synthesis for interpolation and extrapolation tasks.

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

# A APPENDIX

The appendix includes:

- Rendering equation and preliminary for vanilla 3DGS.
- Implementation details of Auxiliary Deformation Field.
- Implementation details of translation and rotation system.
- Training and evaluation resources for the model.
- Additional incremental learning experiments for self-propelled objects.
- Additional details of datasets.
- Additional quantitative results for ablation study in the main context.
- Additional ablation studies.
- Analysis of different ways for interpolation period.
- Additional quantitative results for GoPro dataset.
- Additional quantitative & qualitative results for future frame extrapolation.
- Additional qualitative results for motion segmentation.
- Additional qualitative results for extrapolation beyond dataset time span.

## A.1 PRELIMINARY FOR VANILLA 3DGS

3D Gaussian Splatting (Kerbl et al., 2023) represents a 3D scene by a set of colored 3D Gaussian kernels. Specifically, each Gaussian kernel is parameterized by a 3D position $\boldsymbol{P} \in \mathcal{R}^3$, an orientation represented by a quaternion $\boldsymbol{r}$, and a scaling $\boldsymbol{s}$. By transforming the orientation $\boldsymbol{r}$ and scaling $\boldsymbol{s}$ into the rotation matrix $\mathbf{R}$ and scaling matrix $\mathbf{S}$, a 3D covariance matrix $\Sigma$ can be composed as $\Sigma = \mathbf{R}\mathbf{S}\mathbf{S}^T\mathbf{R}^T$. Then the Gaussian kernel can be evaluated at any location $\boldsymbol{x} \in \mathcal{R}^3$ in the 3D space:

$$G(\boldsymbol{x}) = e^{-\frac{1}{2}(\boldsymbol{x}-\boldsymbol{P})^T\Sigma^{-1}(\boldsymbol{x}-\boldsymbol{P})}. \tag{9}$$

Besides, each Gaussian kernel has an opacity $\sigma$ indicating its influence in rendering, and a color $\boldsymbol{c}$ computed from spherical harmonics (SH) for view-dependent appearance.

The rendering of Gaussian kernels on the image consists of two steps. Firstly, The Gaussian kernels are projected onto the image plane, following the differentiable rasterization pipeline proposed in (Zwicker et al., 2001). The 3D position $\boldsymbol{P}$ and covariance matrix $\Sigma$ of each Gaussian kernel are projected into 2D position $\boldsymbol{P}' = JW\boldsymbol{P}$ and covariance matrix $\Sigma' = JW\Sigma W^T J^T$ respectively, where $J$ denotes the Jacobian of the approximated projective transformation and $W$ denotes the transformation from the world to camera coordinates. Secondly, the color of a pixel $\boldsymbol{\mu}$ on the image can be rendered by $\alpha$-blending as follows:

$$\boldsymbol{C}(\boldsymbol{\mu}) = \sum_i T_i \alpha_i \boldsymbol{c}_i, \quad T_i = \prod_{j=1}^{i-1}(1-\alpha_j), \tag{10}$$

where $\alpha_i$ is obtained by evaluating the projection of the Gaussian kernel $G_i$ on the pixel $\boldsymbol{\mu}$, *i.e.*, $\alpha_i = \sigma_i e^{-\frac{1}{2}(\boldsymbol{\mu}-\boldsymbol{P}')^T\Sigma'^{-1}(\boldsymbol{\mu}-\boldsymbol{P}')}$. By adjusting the parameters of Gaussian kernels mentioned above and adaptively controlling the Gaussian density, a high-fidelity representation of a 3D scene can be obtained from multi-view images. We refer readers to (Kerbl et al., 2023) for more details.

## A.2 IMPLEMENTATION DETAILS OF AUXILIARY DEFORMATION FIELD

We leverage an existing deformation field introduced in (Yang et al., 2024) as our auxiliary deformation field. In particular, the 3D position $\boldsymbol{x}_0$ of each canonical Gaussian kernel and the current timestamp $t$ are fed into an MLP-based deformation network, denoted as $f_{defo}$. The implementation of this MLP is directly adapted from (Yang et al., 2024), *i.e.*, an MLP with 8 layers in total and 256 hidden sizes for each layer, plus a ResNet layer at layer 4. At the input layer, an 8-degree

positional embedding is applied onto the 3D position $x_0$ and a 5-degree positional embedding onto time $t$.

Mathematically, the deformation field $f_{defo}(\boldsymbol{x}, t) : \mathcal{R}^4 \rightarrow \mathcal{R}^{10}$ is defined as

$$(\delta\boldsymbol{x}, \delta\boldsymbol{r}, \delta\boldsymbol{s}) = f_{defo}(\boldsymbol{x}_0, t), \tag{11}$$

where $\delta\boldsymbol{x}$ represents the translation of the center of Gaussian kernel, $\delta\boldsymbol{r}$ represents the rotation for the pose of Gaussian kernel in quaternion representation, and $\delta\boldsymbol{s}$ is the difference of Gaussian sizes. Note a Gaussian size vector is parametrized by $\boldsymbol{z} = \log(\boldsymbol{s})$, so the difference can be defined as $\delta\boldsymbol{s} = \exp(\delta\boldsymbol{z})$.

After applying the deformation field onto Gaussian kernels, we can deform the Gaussian kernels from canonical time to time $t$ as:

$$\boldsymbol{x}_t = \boldsymbol{x}_0 + \delta\boldsymbol{x} \tag{12}$$
$$\boldsymbol{r}_t = \delta\boldsymbol{r} \circ \boldsymbol{r}_0 \tag{13}$$
$$\boldsymbol{s}_t = \exp(\boldsymbol{z}_0 + \delta\boldsymbol{z}) = \boldsymbol{s}_0 \odot \delta\boldsymbol{s}, \tag{14}$$

where $\circ$ is quaternion multiplication and $\odot$ is element-wise multiplication.

### A.3  IMPLEMENTATION DETAILS OF TRANSLATION AND ROTATION SYSTEM

As discussed in the main context, we learn the following two groups of physical parameters for each 3D particle $\boldsymbol{P}$ as function $f_{trd}(\boldsymbol{x}) : \mathcal{R}^3 \rightarrow \mathcal{R}^{13}$:

$$\{(\boldsymbol{P}_c, \boldsymbol{v}_c, \boldsymbol{a}_c), (\boldsymbol{w}_p, \epsilon_p)\} = f_{trd}(\boldsymbol{P}), \tag{15}$$

where angular velocity is defined as $\omega = \|\boldsymbol{w}_p\|_2$ around rotation axes direction $\hat{\boldsymbol{k}} = \boldsymbol{w}_c/\omega$ following right-hand rule.

This module is implemented by a simple $6 \times 128$ MLPs with 6 layers in total and 128 hidden sizes for each layer. In addition, a 5-degree positional embedding is applied onto the input 3D position $\boldsymbol{x}$, and relu is chosen as the activation functions.

### A.4  TRAINING AND EVALUATION RESOURCES FOR THE MODEL

As the complexity of different scenes varies, the total number of Gaussians learned for each scene varies from 40k to 1.6M. In general, our training time is 1.05 times longer than DefGS (or 4DGS if built on it). For example, on the *bat* of Dynamic Object Dataset, DefGS/4DGS need 25 minutes, while we need 27 minutes, with a slight training cost addition. Since our additional module is a tiny MLPs, we only need 367.4kB larger storage. Our rendering speed is 0.85 times slower than DefGS (or 0.8 times slower than 4DGS if built on it). For example, on the *bat* of Dynamic Object Dataset, they achieve 40fps and ours 32fps. We train all our models on a single NVIDIA 3090 24G GPU.

### A.5  INCREMENTAL LEARNING FOR SELF-PROPELLED OBJECTS

We include an additional incremental learning experiment to show that our framework can easily adapt to new observations when internal forces change for self-propelled objects. We choose three self-propelled objects from the Dynamic Object Dataset for this experiment. We keep the same viewing angles in training and testing split, and incrementally train the network.

To be specific, we first feed time $t = 0 \sim 0.15$ to train the network, and evaluate novel view interpolation on $t = 0 \sim 0.15$, future frame extrapolation on $t = 0.15 \sim 0.30$. Next, we include $t = 0.15 \sim 0.30$ to train, and evaluate novel view interpolation on $t = 0 \sim 0.30$, future frame extrapolation on $t = 0.30 \sim 0.45$. We keep adding a time interval of 0.15 till we train from $t = 0 \sim 0.75$, and extrapolate from $t = 0.75 \sim 0.9$.

We compare our performance with DefGS. Table 5 shows quantitative results. It can be seen that DefGS suffers from overfitting the previous timestamps and results in a decrease in its interpolation ability, while our model can stably adapt to new observations and achieve excellent past and future frame predictions. This means that even though the internal forces are changing for self-propelled objects, our model can easily adapt to new observations.

Table 5: Quantitative results (PSNR) of incremental learning.

| Interpolation | $0.15 \rightarrow 0.30$ | $0.30 \rightarrow 0.45$ | $0.45 \rightarrow 0.60$ | $0.60 \rightarrow 0.75$ | $0.75 \rightarrow 0.90$ | Average |
|---|---|---|---|---|---|---|
| DefGS(Yang et al., 2024) | 39.386 | 38.745 | 35.818 | 34.531 | 27.904 | 35.277 |
| **GVFi (Ours)** | **40.032** | **40.706** | **41.013** | **40.466** | **39.971** | **40.438** |
| Extrapolation | $0.15 \rightarrow 0.30$ | $0.30 \rightarrow 0.45$ | $0.45 \rightarrow 0.60$ | $0.60 \rightarrow 0.75$ | $0.75 \rightarrow 0.90$ | Average |
| DefGS(Yang et al., 2024) | 23.438 | 21.360 | 19.989 | 19.670 | 17.629 | 20.417 |
| **GVFi (Ours)** | **29.958** | **32.260** | **31.384** | **29.527** | **28.958** | **30.417** |

## A.6 ADDITIONAL DETAILS OF NEW DATASETS

**Dynamic Multipart dataset:** This dataset comprises 4 distinct objects [1], including a variety of challenging motions. Details of the 4 dynamic objects are:

- **Foldingchair:** A folingchair is given. This chair is composed of three parts. The whole motion is unfolding this chair, so all three parts are undergoing different rotating motions.

- **Hypoerbolic Slot:** This is an extremely hard case, where a stick is rotating through a hypoerbolic slot. Note that, only the stick in this hyperbolic shape is dynamic, this introduces more challenges in motion extrapolation.

- **Satellite:** This object is a satellite with two wing doors opening and one main door opening, all rotating in different directions.

- **Stove:** A home stove is given. The motion is mainly closing its top cover plate.

**GoPro Dataset**: This dataset includes 4 challenging real-world dynamic scenes.

- **Scene #1: Box**. This scene contains a drawer-like box, and a person is trying to close it. The difficulty lies in a tight combination of the moving part and the static part of the box, especially in the future.

- **Scene #2: Hammer**. This scene contains a hammer moving on the topside of a box. The difficulty lies in the direct contact of moving objects and static objects, which requires sharp separation of diverse motion patterns in order to keep the right static/moving states in the future.

- **Scene #3: Collision**. This scene contains a cube and a cup moving towards each other. The difficulty is the different directions of two motions. It is hard to keep the shapes of these two objects in the future.

- **Scene #4: Wrist Rest**. A person is trying to bend a wrist rest. The difficulty is that the object is deformable and the motion is thus not rigid or part-wise rigid.

## A.7 ADDITIONAL QUANTITATIVE RESULTS FOR ABLATION STUDY FOR THE MAIN CONTEXT

We first elaborate how we implement the ablation study (3). Our original design of the translation and rotation dynamics system module $f_{trd}(\boldsymbol{x})$ is only relevant to space, but not time. To obtain the corresponding physics parameters at time $t$, we use Equation 4 to derive the queried physics parameters. In our ablation study (3), we aim to keep the physics parameters changing over time, thus making it more complex in theory. Particularly, we use the same network architecture of $f_{trd}$, except changing the input from $f_{trd}(\boldsymbol{x})$ to $f_{trd}(\boldsymbol{x}, t)$, to force the change of physics parameters.

Here we show the total results for the ablation study in Table 6, both for interpolation and extrapolation.

---

[1] All objects are purchased from SketchFab, licensed under the SketchFab Standard License: https://sketchfab.com/licenses, and are all allowed for AI generation model usage

Table 6: Quantitative results of ablation studies on Dynamic Multipart dataset.

|  | $f_{defo}$ | $f_{trd}$ | Interpolation | | | Extrapolation | | |
|---|---|---|---|---|---|---|---|---|
|  |  |  | PSNR↑ | SSIM↑ | LPIPS↓ | PSNR↑ | SSIM↑ | LPIPS↓ |
| (1) $\delta t$ | ✓ | ✓ | 35.128 | **0.991** | **0.011** | 29.441 | 0.984 | 0.013 |
| (1) $2\delta t$ | ✓ | ✓ | 34.807 | **0.991** | **0.011** | **30.721** | **0.986** | **0.012** |
| (1) $3\delta t$ | ✓ | ✓ | 35.223 | **0.991** | **0.011** | 30.246 | 0.985 | **0.012** |
| (2) - | ✗ | ✓ | 32.266 | 0.987 | 0.016 | 27.081 | 0.981 | 0.018 |
| (3) $2\delta t$ | ✓ | ✗ | **35.225** | **0.991** | **0.011** | 29.986 | 0.985 | **0.012** |

## A.8 ADDITIONAL ABLATION STUDIES

Since the range of motion between two consecutive frames is different across different datasets, we conduct extensive ablations about different choices of $\Delta t$ on all 4 datasets, and the results are listed in Table 7. We observe that $3\delta t$ works better in extrapolation on three datasets (Dynamic Object/ Dynamic Indoor Scene/ NVIDIA Dynamic Scenes). The basic rule to select an appropriate $\delta t$ is based on the motion range. If the motion changes fast, so the motion between two consecutive frames is apparent enough, then a smaller $\delta t$ is good enough. Otherwise, if the motion is rather slow, then a larger $\delta t$ is preferred.

Table 7: Quantitative results of ablation studies for $\delta t$ on all four datasets.

|  | Dynamic Multipart Dataset | | | | | | Dynamic Object Dataset | | | | | |
|---|---|---|---|---|---|---|---|---|---|---|---|---|
|  | Interpolation | | | Extrapolation | | | Interpolation | | | Extrapolation | | |
|  | PSNR↑ | SSIM↑ | LPIPS↓ | PSNR↑ | SSIM↑ | LPIPS↓ | PSNR↑ | SSIM↑ | LPIPS↓ | PSNR↑ | SSIM↑ | LPIPS↓ |
| $\delta t$ | 35.128 | **0.991** | **0.011** | 29.441 | 0.984 | 0.013 | **38.929** | **0.995** | **0.005** | 28.506 | 0.981 | 0.013 |
| $2\delta t$ | 34.807 | **0.991** | **0.011** | **30.721** | **0.986** | **0.012** | 38.788 | **0.995** | 0.006 | 28.758 | 0.982 | **0.011** |
| $3\delta t$ | **35.223** | **0.991** | **0.011** | 30.246 | 0.985 | **0.012** | 38.693 | **0.995** | 0.006 | **29.414** | **0.983** | 0.012 |
|  | Dynamic Indoor Scene Dataset | | | | | | NVIDIA Dynamic Scenes Dataset | | | | | |
|  | Interpolation | | | Extrapolation | | | Interpolation | | | Extrapolation | | |
|  | PSNR↑ | SSIM↑ | LPIPS↓ | PSNR↑ | SSIM↑ | LPIPS↓ | PSNR↑ | SSIM↑ | LPIPS↓ | PSNR↑ | SSIM↑ | LPIPS↓ |
| $\delta t$ | 32.179 | **0.929** | **0.089** | 34.387 | 0.964 | 0.046 | 26.823 | **0.891** | **0.101** | 28.781 | 0.934 | 0.070 |
| $2\delta t$ | 32.202 | 0.928 | 0.089 | 34.556 | 0.964 | 0.046 | 26.943 | **0.891** | 0.102 | 29.388 | **0.938** | 0.067 |
| $3\delta t$ | **32.296** | 0.928 | **0.089** | **35.242** | **0.967** | **0.045** | **27.099** | 0.890 | 0.103 | **29.440** | **0.938** | 0.067 |

The updating scheme of our translation rotation dynamics system is chosen as a second-order relationship in Equation 4, *i.e.*, each rigid particle can have a constant acceleration. We also evaluate the third-order scheme (acceleration of acceleration) and first-order scheme (no acceleration) on Dynamic Multipart Dataset and Dynamic Object Dataset. Table 8 shows the results. We can see that, in Dynamic Object Dataset which has several self-propelled objects whose internal forces tend to change over time, not surprisingly, the third-order variant performs better. Nevertheless, due to the inherent over-parametrization, the third-order scheme tends to learn excessive rotation information to represent simple acceleration motions, thus incurring inferior performance on the Dynamic Multipart Dataset which does not have self-propelled objects.

Table 8: Quantitative results of ablation studies about 3 orders of Taylor expansion in Equation 4 on Dynamic Multipart Dataset and Dynamic Object Dataset.

|  | Dynamic Multipart Dataset | | | | | | Dynamic Object Dataset | | | | | |
|---|---|---|---|---|---|---|---|---|---|---|---|---|
|  | Interpolation | | | Extrapolation | | | Interpolation | | | Extrapolation | | |
|  | PSNR↑ | SSIM↑ | LPIPS↓ | PSNR↑ | SSIM↑ | LPIPS↓ | PSNR↑ | SSIM↑ | LPIPS↓ | PSNR↑ | SSIM↑ | LPIPS↓ |
| $1^{st}$-order | 34.776 | 0.990 | 0.013 | 26.729 | 0.976 | 0.018 | 38.892 | **0.995** | **0.005** | 28.536 | **0.983** | 0.012 |
| $2^{nd}$-order | 34.807 | **0.991** | **0.011** | **30.721** | **0.986** | **0.012** | 38.788 | **0.995** | 0.006 | 28.758 | 0.982 | **0.011** |
| $3^{rd}$-order | **35.268** | **0.991** | 0.012 | 30.503 | 0.985 | 0.013 | **39.164** | **0.995** | **0.005** | **29.378** | **0.983** | **0.011** |

## A.9 ANALYSIS OF DIFFERENT WAYS FOR INTERPOLATION PERIOD

Our method has three possible strategies for interpolation rendering: (1) directly using $f_{defo}$ to predict the deformation at the given time $t$, (2) progressively calculating the Gaussian deformation at the given time $t$ from time 0 using the motion parameters predicted by $f_{trd}$, or (3) following the steps described in Section 3.3. We choose the third one in our main experiments. Here, we evaluate the other two strategies in Table 9. We can see that, the first and the third strategies are not strictly consistent, but achieve very similar performance. However, for the second strategy, the performance clearly decreases, we hypothesize that this is due to the accumulated errors in the autoregressive process.

Table 9: Quantitative results of different interpolation strategies of our method on all four datasets.

| | Multipart | | | Object | | | Indoor | | | NVIDIA | | |
|---|---|---|---|---|---|---|---|---|---|---|---|---|
| | PSNR↑ | SSIM↑ | LPIPS↓ | PSNR↑ | SSIM↑ | LPIPS↓ | PSNR↑ | SSIM↑ | LPIPS↓ | PSNR↑ | SSIM↑ | LPIPS↓ |
| (1)$f_{defo}$ | **35.040** | **0.991** | **0.011** | 38.406 | **0.995** | **0.005** | **32.569** | **0.930** | **0.088** | **26.951** | **0.891** | **0.102** |
| (2)$f_{trd}$ | 30.310 | 0.984 | 0.017 | 33.527 | 0.991 | 0.009 | 31.776 | 0.926 | 0.092 | 25.899 | 0.875 | 0.118 |
| (3)$f_{defo} + f_{trd}$ | 34.807 | **0.991** | **0.011** | **38.788** | **0.995** | 0.006 | 32.202 | 0.928 | 0.089 | 26.943 | **0.891** | **0.102** |

## A.10 ADDITIONAL QUANTITATIVE RESULTS ON DYNAMIC OBJECT DATASET

Here we show the total per-scene results on Dynamic Object Dataset in Table 10 and qualitative results in Figures 8, 9&10, both for interpolation and extrapolation.

## A.11 ADDITIONAL QUANTITATIVE RESULTS ON DYNAMIC INDOOR SCENE DATASET

Here we show the total per-scene results in Dynamic Indoor Scene Datasets in Table 11 and qualitative results in Figures 10&11, both for interpolation and extrapolation.

## A.12 ADDITIONAL QUANTITATIVE RESULTS ON NVIDIA DYNAMIC SCENE DATASET

Here we show the total per-scene results on NVIDIA Dynamic Scene Dataset in Table 12 and qualitative results in Figure 13, both for interpolation and extrapolation.

## A.13 ADDITIONAL QUANTITATIVE RESULTS ON DYNAMIC MULTIPART DATASET

Here we show the total per-scene results on our Dynamic Multipart Dataset in Table 13 and qualitative results in Figure 12, both for interpolation and extrapolation.

## A.14 ADDITIONAL QUANTITATIVE & QUALITATIVE RESULTS FOR GOPRO DATASET

Here we show the total results on our GoPro Dataset in Table 14 and qualitative results in Figures 14,15,&16, both for interpolation and extrapolation.

## A.15 ADDITIONAL QUALITATIVE RESULTS FOR EXTRAPOLATION BEYOND DATASET TIME SPANS

We list some meaningful longer extrapolation results from each dataset here in Figure 21. In our dataset, the training period lasts from $t = 0$ to $t = 0.75$ and the extrapolation period lasts from $t = 0.75$ to $t = 1.0$. Here we show the qualitative results till $t = 1.5$, which is already twice the training period. We can see that our method can still obtain physically meaningful future frame prediction in particularly high quality.

## A.16 ADDITIONAL QUALITATIVE RESULTS FOR OBJECT/PART SEGMENTATION

Figures 5, 6 and 7 show more qualitative results for the autonomous object or part segmentation based on the learned physical parameters via the simple K-means clustering algorithm.

Table 10: Per-scene quantitative results on Dynamic Object dataset.

| | Falling Ball | | | | | | Bat | | | | | |
|---|---|---|---|---|---|---|---|---|---|---|---|---|
| Methods | Interpolation | | | Extrapolation | | | Interpolation | | | Extrapolation | | |
| | PSNR↑ | SSIM↑ | LPIPS↓ | PSNR↑ | SSIM↑ | LPIPS↓ | PSNR↑ | SSIM↑ | LPIPS↓ | PSNR↑ | SSIM↑ | LPIPS↓ |
| T-NeRF(Pumarola et al., 2021) | 14.921 | 0.782 | 0.326 | 15.418 | 0.793 | 0.308 | 13.070 | 0.836 | 0.234 | 13.897 | 0.834 | 0.230 |
| D-NeRF(Pumarola et al., 2021) | 15.548 | 0.665 | 0.435 | 15.116 | 0.644 | 0.427 | 14.087 | 0.845 | 0.212 | 15.406 | 0.887 | 0.175 |
| TiNeuVox(Fang et al., 2022) | 35.458 | 0.974 | 0.052 | 20.242 | 0.959 | 0.067 | 16.080 | 0.908 | 0.108 | 16.952 | 0.930 | 0.115 |
| T-NeRF$_{PINN}$ | 17.687 | 0.775 | 0.368 | 17.857 | 0.829 | 0.265 | 16.412 | 0.903 | 0.197 | 18.983 | 0.930 | 0.132 |
| HexPlane$_{PINN}$ | 32.144 | 0.965 | 0.065 | 20.762 | 0.951 | 0.081 | 23.399 | 0.958 | 0.057 | 21.144 | 0.951 | 0.064 |
| NVFi(Li et al., 2023a) | 35.826 | 0.978 | 0.041 | 31.369 | 0.978 | 0.041 | 23.325 | 0.964 | 0.046 | 25.015 | 0.968 | 0.042 |
| DefGS(Yang et al., 2024) | 37.535 | 0.995 | 0.009 | 20.442 | 0.976 | 0.033 | 38.750 | **0.997** | 0.004 | 17.063 | 0.936 | 0.072 |
| DefGS$_{NVFi}$ | **38.606** | **0.996** | 0.010 | 24.873 | 0.985 | 0.015 | 38.075 | **0.997** | 0.004 | **28.950** | **0.980** | **0.015** |
| **GVFi (Ours)** | 38.071 | 0.995 | 0.008 | 35.949 | 0.995 | 0.004 | 39.626 | 0.997 | 0.003 | 24.352 | 0.973 | 0.023 |
| | Fan | | | | | | Telescope | | | | | |
| Methods | Interpolation | | | Extrapolation | | | Interpolation | | | Extrapolation | | |
| | PSNR↑ | SSIM↑ | LPIPS↓ | PSNR↑ | SSIM↑ | LPIPS↓ | PSNR↑ | SSIM↑ | LPIPS↓ | PSNR↑ | SSIM↑ | LPIPS↓ |
| T-NeRFPumarola et al. (2021) | 8.001 | 0.308 | 0.646 | 8.494 | 0.392 | 0.593 | 13.031 | 0.615 | 0.472 | 13.892 | 0.670 | 0.417 |
| D-NeRFPumarola et al. (2021) | 7.915 | 0.262 | 0.690 | 8.624 | 0.370 | 0.623 | 13.295 | 0.609 | 0.469 | 14.967 | 0.700 | 0.385 |
| TiNeuVoxFang et al. (2022) | 24.088 | 0.930 | 0.104 | 20.932 | 0.935 | 0.078 | 31.666 | 0.982 | 0.041 | 20.456 | 0.921 | 0.067 |
| T-NeRF$_{PINN}$ | 9.233 | 0.541 | 0.508 | 9.828 | 0.606 | 0.443 | 14.293 | 0.739 | 0.366 | 15.752 | 0.804 | 0.298 |
| HexPlane$_{PINN}$ | 22.822 | 0.921 | 0.079 | 19.724 | 0.919 | 0.080 | 25.381 | 0.948 | 0.066 | 23.165 | 0.932 | 0.074 |
| NVFiLi et al. (2023a) | 25.213 | 0.948 | 0.049 | 27.172 | 0.963 | 0.037 | 26.487 | 0.959 | 0.048 | 27.101 | 0.963 | 0.046 |
| DefGSYang et al. (2024) | **35.858** | 0.985 | 0.017 | 20.932 | 0.948 | 0.038 | 37.502 | 0.996 | 0.003 | 20.684 | 0.927 | 0.048 |
| DefGS$_{NVFi}$ | 35.217 | 0.984 | 0.019 | 26.648 | 0.972 | 0.023 | 37.568 | 0.996 | 0.003 | **34.096** | **0.994** | **0.005** |
| **GVFi (Ours)** | 35.577 | **0.986** | **0.013** | **29.533** | **0.979** | **0.012** | 40.614 | 0.998 | 0.002 | 29.744 | 0.983 | 0.007 |
| | Shark | | | | | | Whale | | | | | |
| Methods | Interpolation | | | Extrapolation | | | Interpolation | | | Extrapolation | | |
| | PSNR↑ | SSIM↑ | LPIPS↓ | PSNR↑ | SSIM↑ | LPIPS↓ | PSNR↑ | SSIM↑ | LPIPS↓ | PSNR↑ | SSIM↑ | LPIPS↓ |
| T-NeRFPumarola et al. (2021) | 13.813 | 0.853 | 0.223 | 15.325 | 0.882 | 0.193 | 16.141 | 0.860 | 0.212 | 15.880 | 0.860 | 0.203 |
| D-NeRFPumarola et al. (2021) | 17.727 | 0.903 | 0.150 | 19.078 | 0.936 | 0.092 | 16.373 | 0.898 | 0.154 | 14.771 | 0.883 | 0.171 |
| TiNeuVoxFang et al. (2022) | 23.178 | 0.971 | 0.059 | 19.463 | 0.950 | 0.050 | 37.455 | 0.994 | 0.016 | 19.624 | 0.943 | 0.063 |
| T-NeRF$_{PINN}$ | 17.315 | 0.878 | 0.177 | 18.739 | 0.921 | 0.115 | 16.778 | 0.927 | 0.141 | 15.974 | 0.919 | 0.127 |
| HexPlane$_{PINN}$ | 28.874 | 0.976 | 0.040 | 22.330 | 0.961 | 0.047 | 29.634 | 0.981 | 0.035 | 21.391 | 0.961 | 0.053 |
| NVFiLi et al. (2023a) | 32.072 | 0.984 | 0.024 | 28.874 | 0.982 | 0.021 | 31.240 | 0.986 | 0.025 | 26.032 | 0.978 | 0.029 |
| DefGSYang et al. (2024) | 37.802 | 0.994 | 0.006 | 19.924 | 0.957 | 0.034 | 39.740 | **0.997** | 0.004 | 20.048 | 0.951 | 0.046 |
| DefGS$_{NVFi}$ | 37.327 | 0.994 | 0.006 | **29.240** | **0.987** | **0.007** | 37.101 | 0.996 | 0.005 | **28.686** | **0.986** | **0.012** |
| **GVFi (Ours)** | **40.464** | **0.997** | **0.004** | 26.680 | 0.979 | 0.009 | 38.376 | 0.997 | 0.003 | 26.288 | 0.982 | 0.013 |

Table 11: Per-scene quantitative results on Dynamic Indoor Scene dataset.

| | Gnome House | | | | | | Chessboard | | | | | |
|---|---|---|---|---|---|---|---|---|---|---|---|---|
| Methods | Interpolation | | | Extrapolation | | | Interpolation | | | Extrapolation | | |
| | PSNR↑ | SSIM↑ | LPIPS↓ | PSNR↑ | SSIM↑ | LPIPS↓ | PSNR↑ | SSIM↑ | LPIPS↓ | PSNR↑ | SSIM↑ | LPIPS↓ |
| T-NeRF(Pumarola et al., 2021) | 26.094 | 0.716 | 0.383 | 23.485 | 0.643 | 0.419 | 25.517 | 0.796 | 0.294 | 20.228 | 0.708 | 0.365 |
| D-NeRF(Pumarola et al., 2021) | 27.000 | 0.745 | 0.319 | 21.714 | 0.641 | 0.367 | 24.852 | 0.774 | 0.308 | 19.455 | 0.675 | 0.384 |
| TiNeuVox(Fang et al., 2022) | 30.646 | 0.831 | 0.253 | 21.418 | 0.699 | 0.326 | 33.001 | 0.917 | 0.177 | 19.718 | 0.765 | 0.310 |
| T-NeRF$_{PINN}$ | 15.008 | 0.375 | 0.668 | 16.200 | 0.409 | 0.651 | 16.549 | 0.457 | 0.621 | 17.197 | 0.472 | 0.618 |
| HexPlane$_{PINN}$ | 23.764 | 0.658 | 0.510 | 22.867 | 0.658 | 0.510 | 24.605 | 0.778 | 0.412 | 21.518 | 0.748 | 0.428 |
| NSFF(Li et al., 2021) | 31.418 | 0.821 | 0.294 | 25.892 | 0.750 | 0.327 | 32.514 | 0.810 | 0.201 | 21.501 | 0.805 | 0.282 |
| NVFi(Li et al., 2023a) | 30.667 | 0.824 | 0.277 | 30.408 | 0.826 | 0.273 | 30.394 | 0.888 | 0.215 | 27.840 | 0.872 | 0.219 |
| DefGS(Yang et al., 2024) | 32.041 | 0.918 | 0.132 | 21.703 | 0.775 | 0.207 | 27.355 | 0.912 | 0.147 | 20.032 | 0.808 | 0.218 |
| DefGS$_{NVFi}$ | **32.881** | 0.919 | 0.132 | 33.630 | 0.953 | 0.077 | 26.200 | 0.907 | 0.156 | 26.730 | 0.917 | 0.110 |
| **GVFi (Ours)** | 32.698 | **0.921** | **0.101** | 36.578 | 0.962 | 0.055 | 35.138 | 0.960 | 0.060 | 33.685 | 0.966 | 0.042 |
| | Factory | | | | | | Dining Table | | | | | |
| Methods | Interpolation | | | Extrapolation | | | Interpolation | | | Extrapolation | | |
| | PSNR↑ | SSIM↑ | LPIPS↓ | PSNR↑ | SSIM↑ | LPIPS↓ | PSNR↑ | SSIM↑ | LPIPS↓ | PSNR↑ | SSIM↑ | LPIPS↓ |
| T-NeRFPumarola et al. (2021) | 26.467 | 0.741 | 0.328 | 24.276 | 0.722 | 0.344 | 21.699 | 0.716 | 0.338 | 20.977 | 0.725 | 0.324 |
| D-NeRFPumarola et al. (2021) | 28.818 | 0.818 | 0.252 | 22.959 | 0.746 | 0.303 | 20.851 | 0.725 | 0.319 | 19.035 | 0.705 | 0.341 |
| TiNeuVoxFang et al. (2022) | 32.684 | 0.909 | 0.148 | 22.622 | 0.810 | 0.229 | 23.596 | 0.798 | 0.274 | 20.357 | 0.804 | 0.258 |
| T-NeRF$_{PINN}$ | 16.634 | 0.446 | 0.624 | 17.546 | 0.480 | 0.609 | 16.807 | 0.486 | 0.640 | 18.215 | 0.548 | 0.595 |
| HexPlane$_{PINN}$ | 27.200 | 0.826 | 0.283 | 24.998 | 0.792 | 0.312 | 25.291 | 0.788 | 0.350 | 22.979 | 0.771 | 0.355 |
| NSFFLi et al. (2021) | **33.975** | 0.919 | 0.152 | 26.647 | 0.855 | 0.196 | 19.552 | 0.665 | 0.464 | 22.612 | 0.770 | 0.351 |
| NVFiLi et al. (2023a) | 32.460 | 0.912 | 0.151 | 31.719 | 0.908 | 0.154 | **29.179** | 0.885 | 0.199 | 29.011 | 0.898 | 0.171 |
| DefGSYang et al. (2024) | 33.629 | **0.943** | 0.096 | 22.820 | 0.839 | 0.169 | 27.680 | 0.890 | 0.145 | 20.965 | 0.855 | 0.157 |
| DefGS$_{NVFi}$ | 33.643 | **0.943** | **0.097** | 33.049 | 0.954 | 0.062 | 27.957 | **0.891** | 0.145 | 30.975 | 0.955 | 0.060 |
| **GVFi (Ours)** | 33.423 | 0.941 | 0.076 | 34.906 | 0.963 | 0.045 | 27.547 | **0.891** | 0.118 | 33.056 | 0.965 | 0.043 |

## A.17 ADDITIONAL QUALITATIVE RESULTS FOR SEGMENTATION ON DYNAMIC INDOOR SCENE DATASET

Figures 17, 18, 19, &20 shows qualitative results for the rendered mask on Dynamic Indoor Scene dataset.

Table 12: Quantitative results of our method and baselines on the NVIDIA Dynamic Scene dataset.

| | Truck | | | | | | Skating | | | | | |
| --- | --- | --- | --- | --- | --- | --- | --- | --- | --- | --- | --- | --- |
| | Interpolation | | | Extrapolation | | | Interpolation | | | Extrapolation | | |
| | PSNR↑ | SSIM↑ | LPIPS↓ | PSNR↑ | SSIM↑ | LPIPS↓ | PSNR↑ | SSIM↑ | LPIPS↓ | PSNR↑ | SSIM↑ | LPIPS↓ |
| T-NeRF(Pumarola et al., 2021) | 18.673 | 0.548 | 0.447 | 18.176 | 0.567 | 0.447 | 27.483 | 0.820 | 0.263 | 24.063 | 0.846 | 0.269 |
| D-NeRF(Pumarola et al., 2021) | 17.660 | 0.554 | 0.431 | 16.905 | 0.544 | 0.445 | 27.994 | 0.869 | 0.187 | 24.361 | 0.873 | 0.208 |
| TiNeuVox(Fang et al., 2022) | 27.230 | 0.846 | 0.229 | 24.887 | 0.848 | 0.209 | 29.377 | 0.889 | 0.202 | 24.224 | 0.878 | 0.220 |
| T-NeRF$_{PINN}$ | 15.241 | 0.413 | 0.540 | 14.959 | 0.395 | 0.552 | 21.644 | 0.780 | 0.338 | 20.990 | 0.814 | 0.303 |
| HexPlane$_{PINN}$ | 25.494 | 0.768 | 0.337 | 24.991 | 0.768 | 0.325 | 24.447 | 0.867 | 0.225 | 23.955 | 0.868 | 0.232 |
| NVFi(Li et al., 2023a) | 27.276 | 0.840 | 0.235 | 28.269 | 0.855 | 0.220 | **26.999** | 0.848 | 0.227 | 28.654 | 0.896 | 0.208 |
| DefGS(Yang et al., 2024) | **28.327** | **0.885** | 0.115 | 24.947 | 0.875 | 0.131 | 24.997 | 0.900 | 0.138 | 23.532 | 0.914 | 0.148 |
| DefGS$_{nvfi}$ | 28.169 | 0.884 | 0.114 | 28.481 | 0.922 | 0.088 | 25.774 | 0.896 | 0.141 | 26.577 | 0.931 | 0.115 |
| **GVFi (Ours)** | 27.977 | 0.880 | **0.097** | **29.655** | **0.931** | **0.063** | 25.909 | **0.901** | **0.106** | **29.120** | **0.944** | **0.071** |

Table 13: Per-scene quantitative results on Dynamic Multipart dataset.

| Methods | Folding Chair | | | | | | Hyperbolic Slot | | | | | |
| --- | --- | --- | --- | --- | --- | --- | --- | --- | --- | --- | --- | --- |
| | Interpolation | | | Extrapolation | | | Interpolation | | | Extrapolation | | |
| | PSNR↑ | SSIM↑ | LPIPS↓ | PSNR↑ | SSIM↑ | LPIPS↓ | PSNR↑ | SSIM↑ | LPIPS↓ | PSNR↑ | SSIM↑ | LPIPS↓ |
| T-NeRF(Pumarola et al., 2021) | 10.146 | 0.598 | 0.537 | 10.260 | 0.586 | 0.548 | 7.437 | 0.424 | 0.749 | 7.098 | 0.404 | 0.739 |
| D-NeRF (Pumarola et al., 2021) | 11.681 | 0.717 | 0.437 | 13.177 | 0.765 | 0.357 | 7.279 | 0.485 | 0.714 | 7.547 | 0.468 | 0.695 |
| TiNeuVox(Fang et al., 2022) | 34.160 | 0.984 | 0.039 | 13.391 | 0.808 | 0.199 | 28.637 | 0.955 | 0.083 | 25.436 | 0.973 | 0.040 |
| NVFi(Li et al., 2023a) | 27.748 | 0.962 | 0.049 | 23.433 | 0.940 | 0.063 | 25.487 | 0.944 | 0.057 | 25.757 | 0.956 | 0.039 |
| DefGS(Yang et al., 2024) | 37.319 | **0.995** | 0.009 | 13.682 | 0.820 | 0.169 | 31.780 | 0.983 | 0.030 | 25.631 | 0.981 | 0.020 |
| DefGS$_{NVFi}$ | 37.269 | 0.994 | 0.009 | 25.404 | 0.962 | 0.022 | **32.506** | **0.985** | 0.025 | 29.351 | 0.988 | 0.012 |
| **GVFi (Ours)** | **37.910** | 0.995 | **0.005** | **27.869** | **0.978** | **0.015** | 31.740 | 0.985 | **0.018** | **34.185** | **0.993** | **0.007** |
| Methods | Satellite | | | | | | Stove | | | | | |
| | Interpolation | | | Extrapolation | | | Interpolation | | | Extrapolation | | |
| | PSNR↑ | SSIM↑ | LPIPS↓ | PSNR↑ | SSIM↑ | LPIPS↓ | PSNR↑ | SSIM↑ | LPIPS↓ | PSNR↑ | SSIM↑ | LPIPS↓ |
| T-NeRF(Pumarola et al., 2021) | 14.614 | 0.754 | 0.307 | 14.468 | 0.751 | 0.328 | 7.134 | 0.490 | 0.605 | 8.429 | 0.562 | 0.531 |
| D-NeRF(Pumarola et al., 2021) | 17.991 | 0.930 | 0.100 | 17.252 | 0.926 | 0.102 | 16.165 | 0.856 | 0.262 | 15.400 | 0.908 | 0.205 |
| TiNeuVox(Fang et al., 2022) | 33.061 | 0.983 | 0.035 | 28.627 | 0.978 | 0.032 | 23.969 | 0.943 | 0.109 | 15.760 | 0.934 | 0.087 |
| NVFi(Li et al., 2023a) | 29.644 | 0.973 | 0.029 | 30.075 | 0.975 | 0.027 | 27.186 | 0.959 | 0.072 | 21.675 | 0.950 | 0.054 |
| DefGS(Yang et al., 2024) | **36.832** | 0.993 | 0.007 | 27.622 | 0.979 | 0.016 | 32.607 | 0.989 | 0.029 | 15.721 | 0.941 | 0.063 |
| DefGS$_{NVFi}$ | 36.640 | 0.993 | 0.007 | **34.282** | **0.990** | **0.007** | 21.134 | 0.988 | 0.029 | 24.781 | 0.977 | 0.027 |
| **GVFi (Ours)** | 36.687 | **0.994** | **0.006** | 31.383 | 0.987 | 0.009 | **32.892** | **0.990** | **0.016** | **29.446** | **0.985** | **0.017** |

Table 14: Quantitative results for both novel view interpolation and future frame extrapolation on GoPro Dataset.

| | GoPro Dataset | | | | | |
| --- | --- | --- | --- | --- | --- | --- |
| | Interpolation | | | Extrapolation | | |
| | PSNR↑ | SSIM↑ | LPIPS↓ | PSNR↑ | SSIM↑ | LPIPS↓ |
| TiNeuVox(Fang et al., 2022) | 15.306 | 0.588 | 0.516 | 20.323 | 0.738 | 0.318 |
| NVFi(Li et al., 2023a) | 14.229 | 0.568 | 0.569 | 19.879 | 0.736 | 0.415 |
| DefGS(Yang et al., 2024) | 20.018 | **0.838** | **0.167** | 21.193 | 0.842 | 0.185 |
| DefGS$_{nvfi}$ | **20.254** | **0.838** | **0.167** | 25.469 | 0.882 | 0.141 |
| **GVFi (Ours)** | 20.124 | 0.834 | 0.168 | **26.276** | **0.890** | **0.131** |

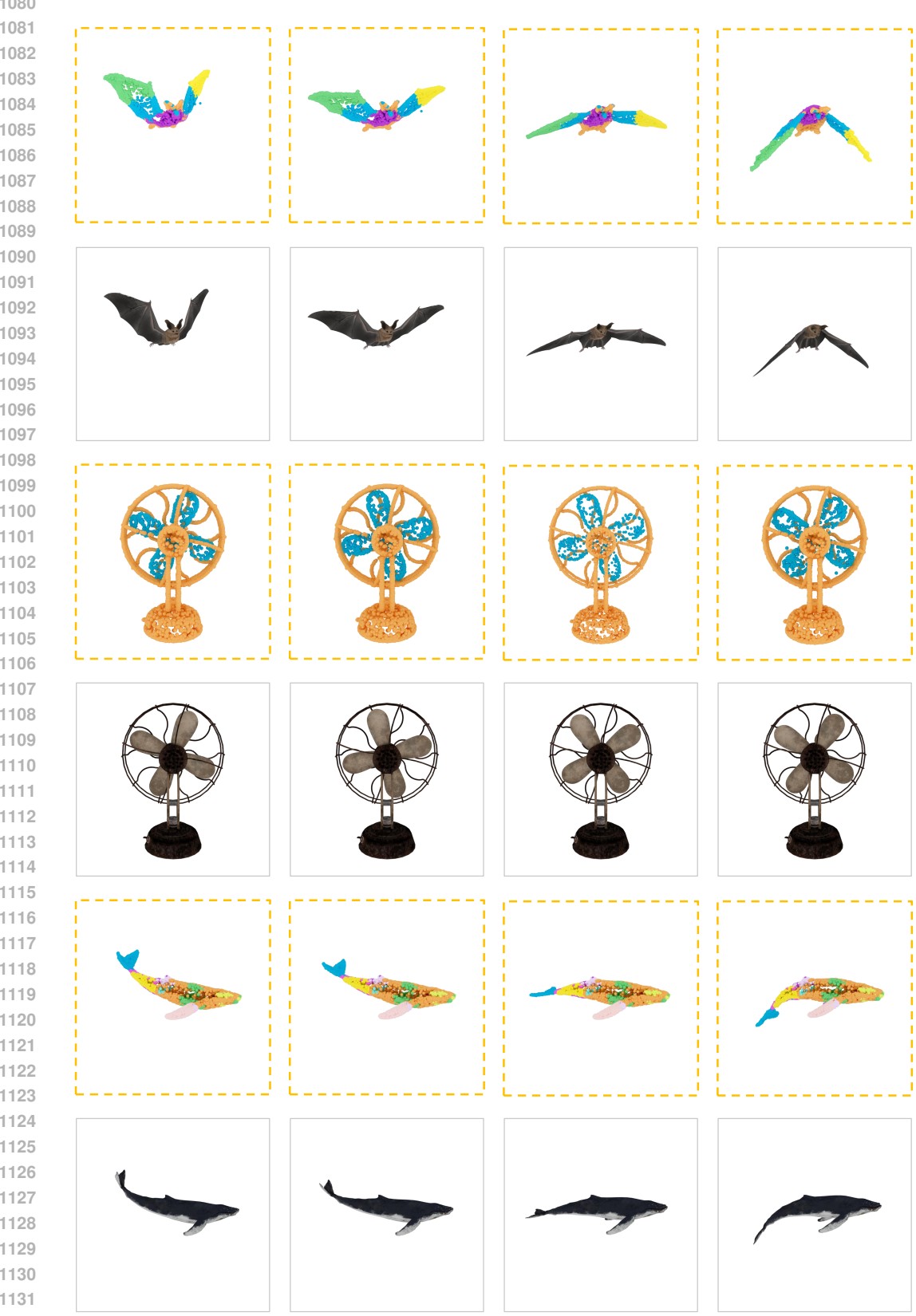

Figure 5: Qualitative results for Object/Part Segmentation on Dynamic Object dataset.

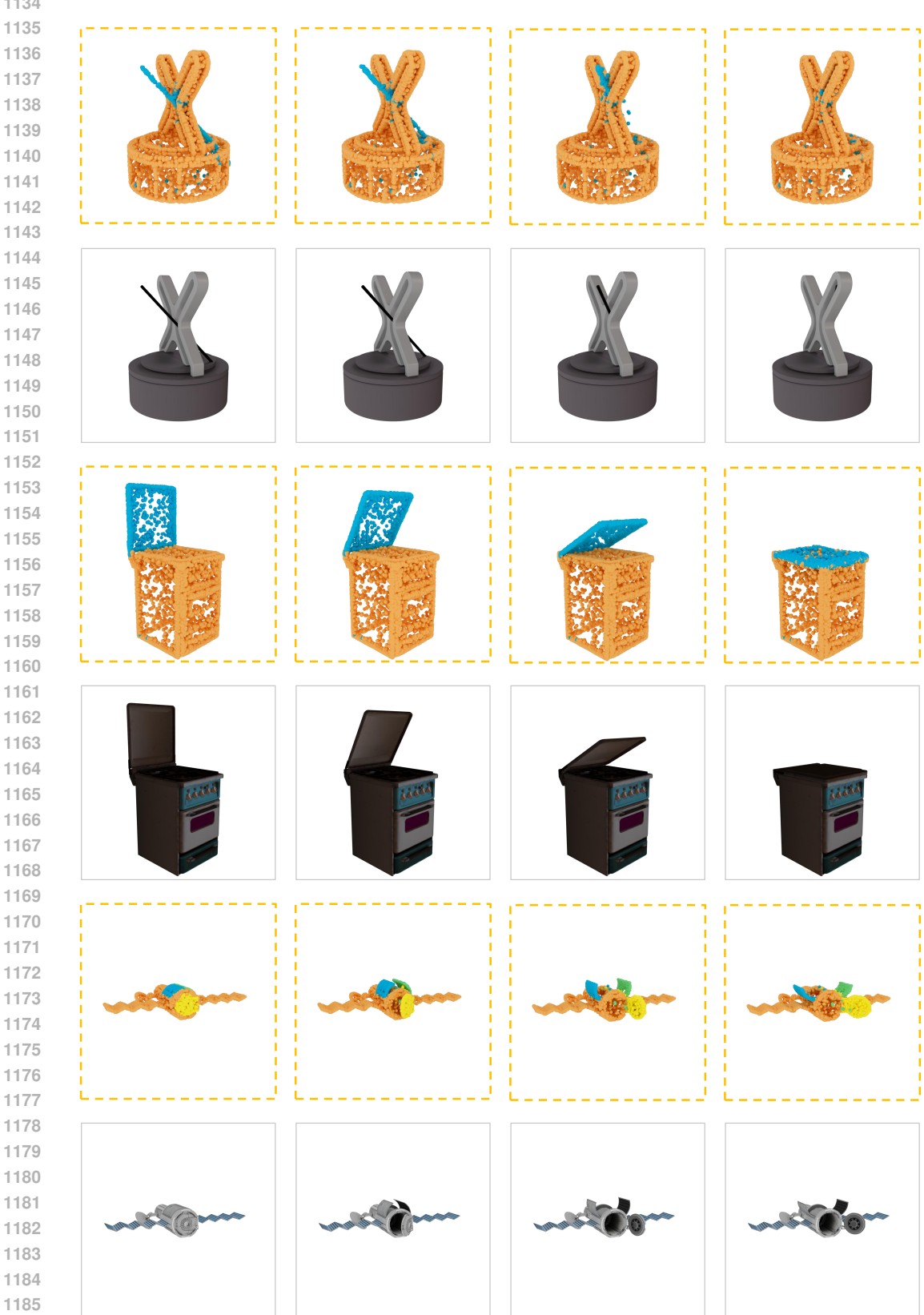

Figure 6: Qualitative results for Object/Part Segmentation on Dynamic Multipart dataset.

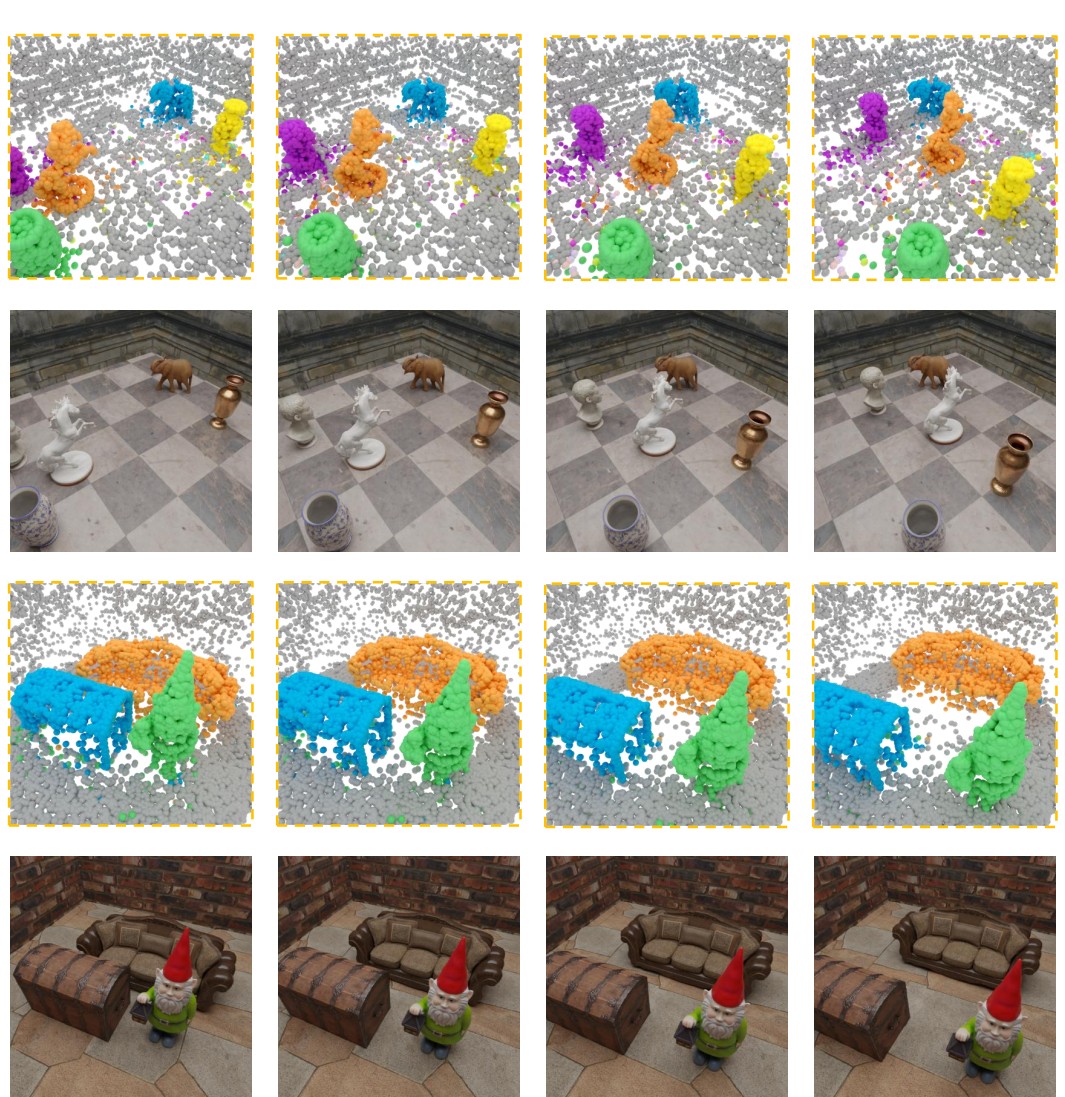

Figure 7: Qualitative results for Object/Part Segmentation on Dynamic Indoor Scene dataset.

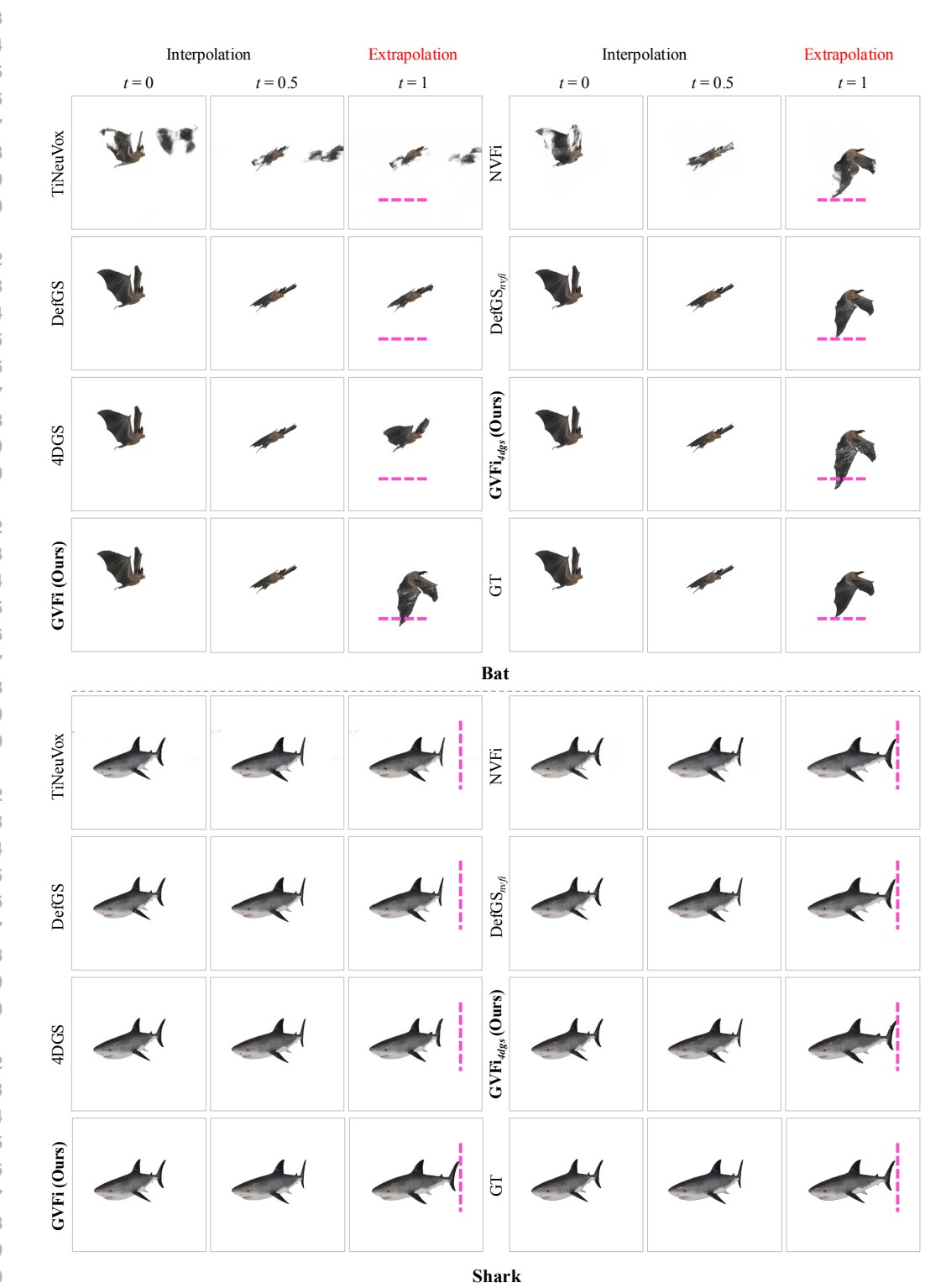

Figure 8: Qualitative results of RGB view synthesis for interpolation and extrapolation tasks on Dynamic Object dataset.

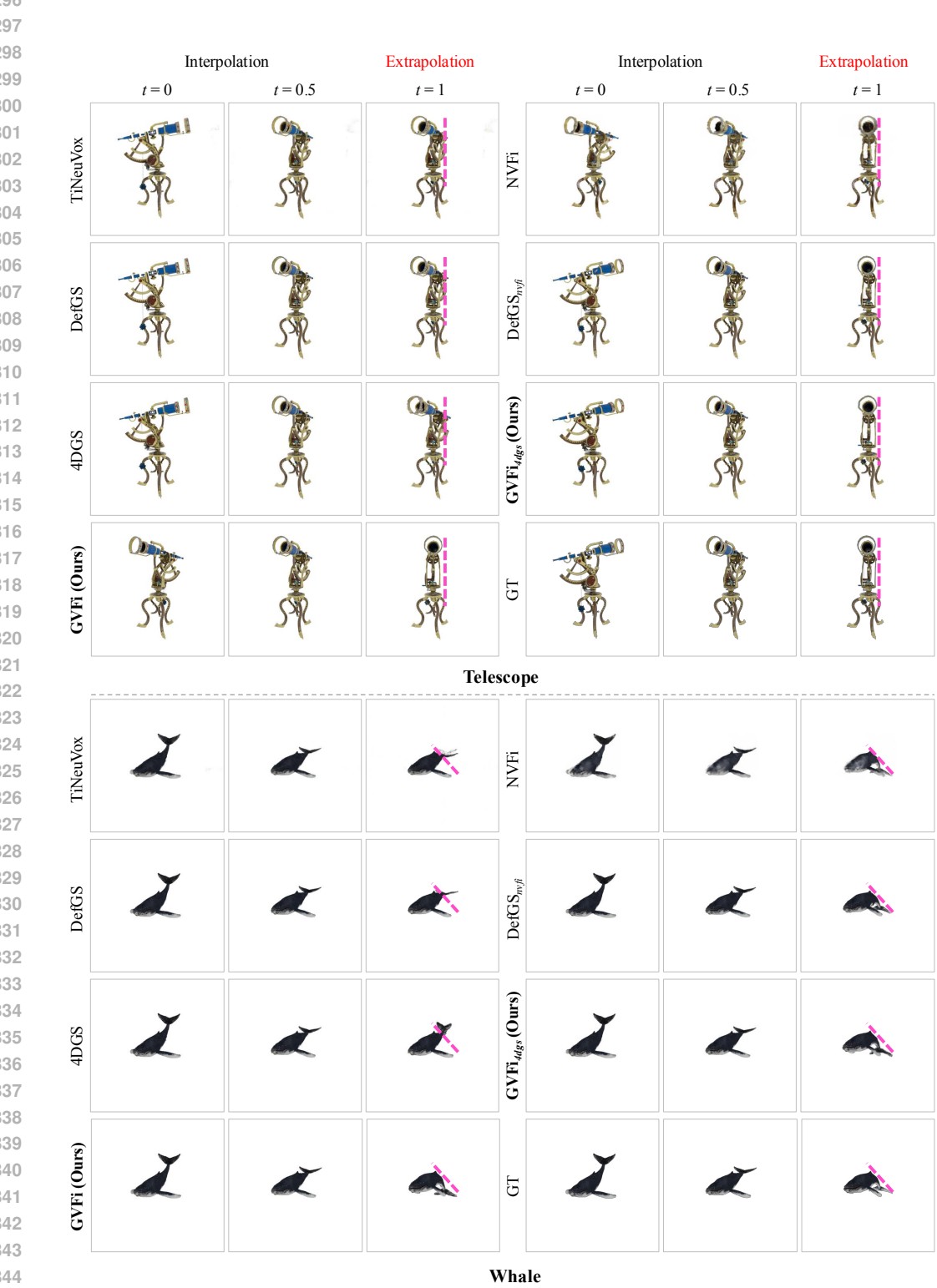

Figure 9: Qualitative results of RGB view synthesis for interpolation and extrapolation tasks on Dynamic Object dataset.

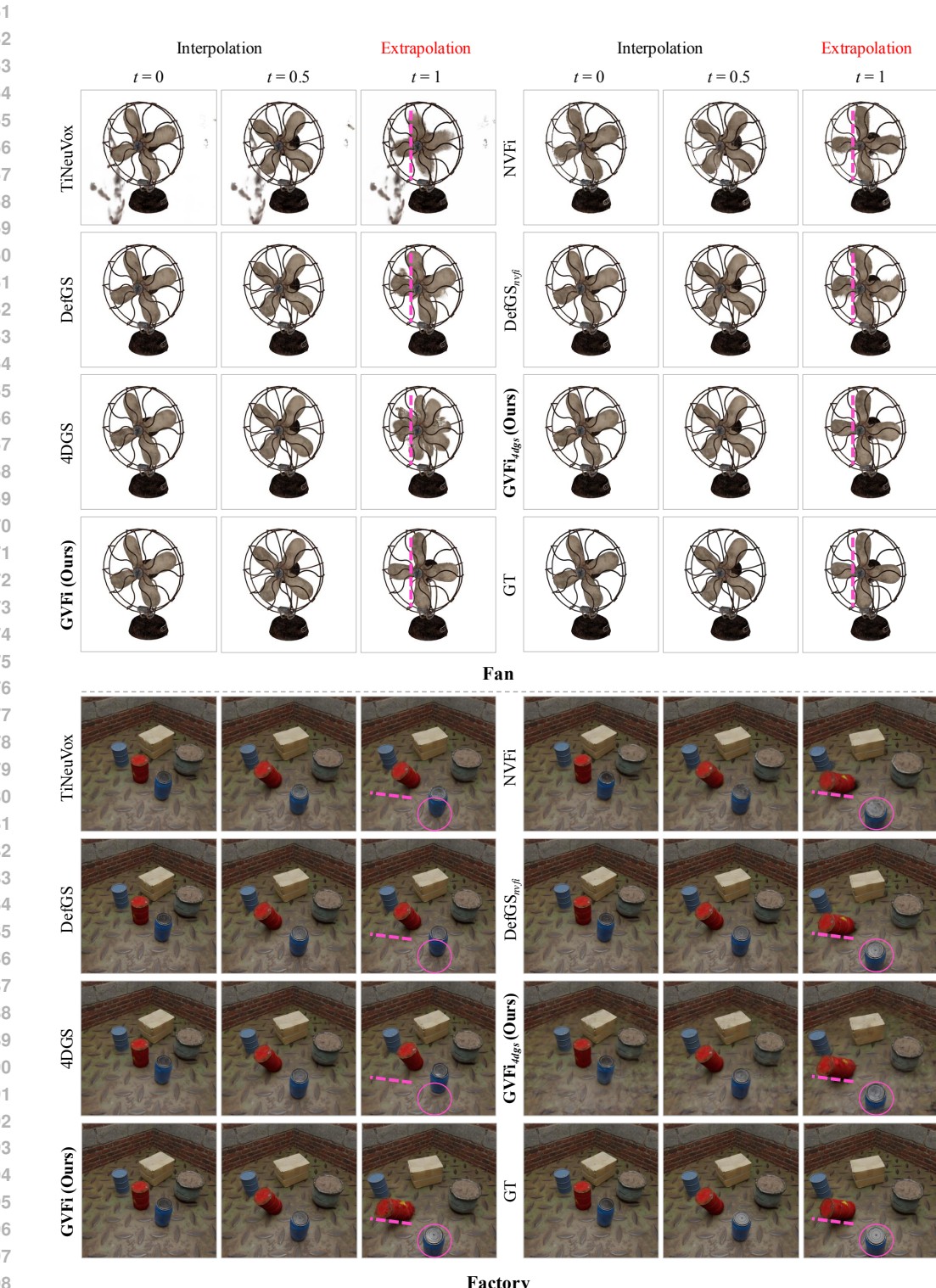

Figure 10: Qualitative results of RGB view synthesis for interpolation and extrapolation tasks on Dynamic Object and Dynamic Indoor Scene datasets.

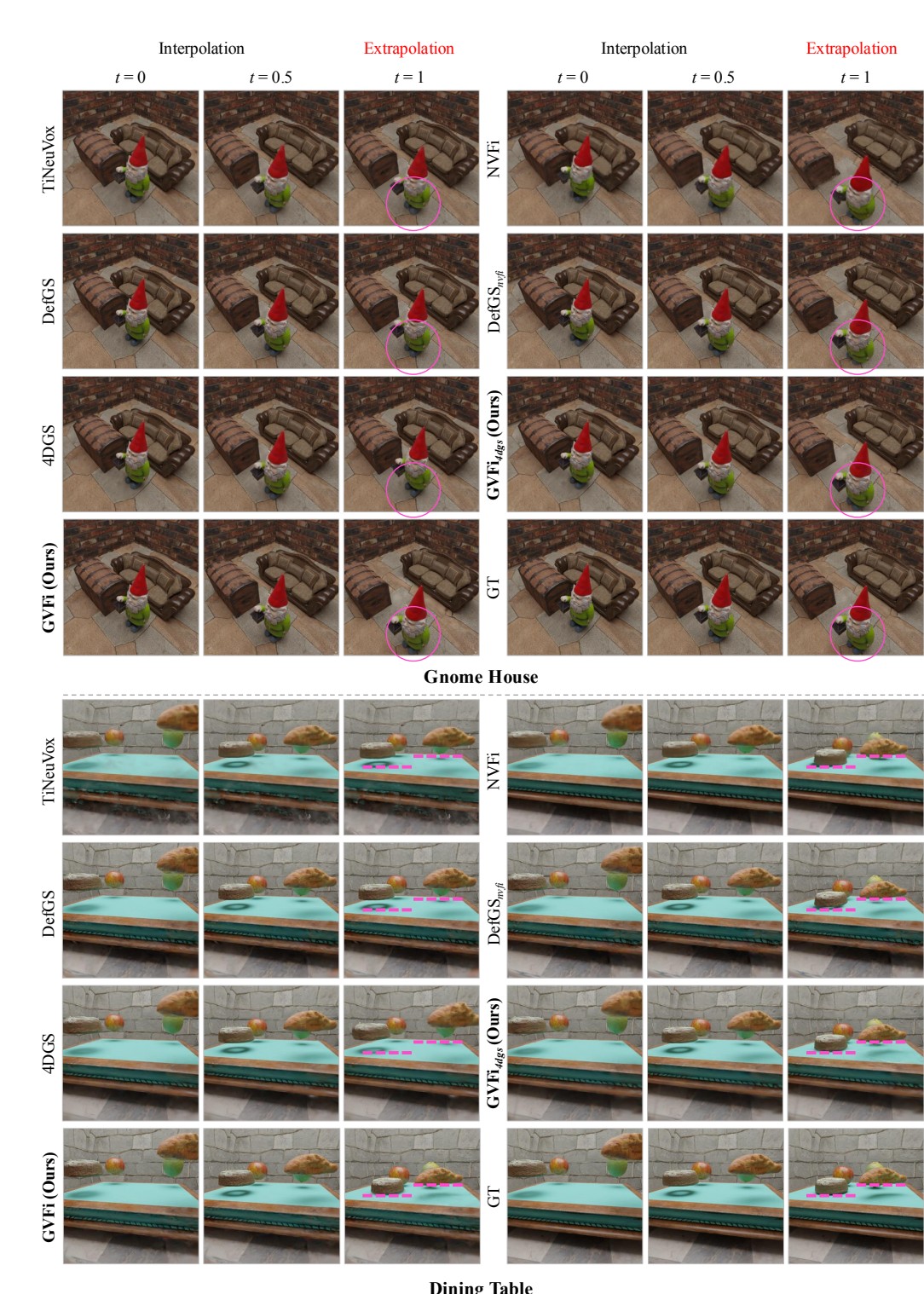

Figure 11: Qualitative results of RGB view synthesis for interpolation and extrapolation tasks on Dynamic Indoor Scene dataset.

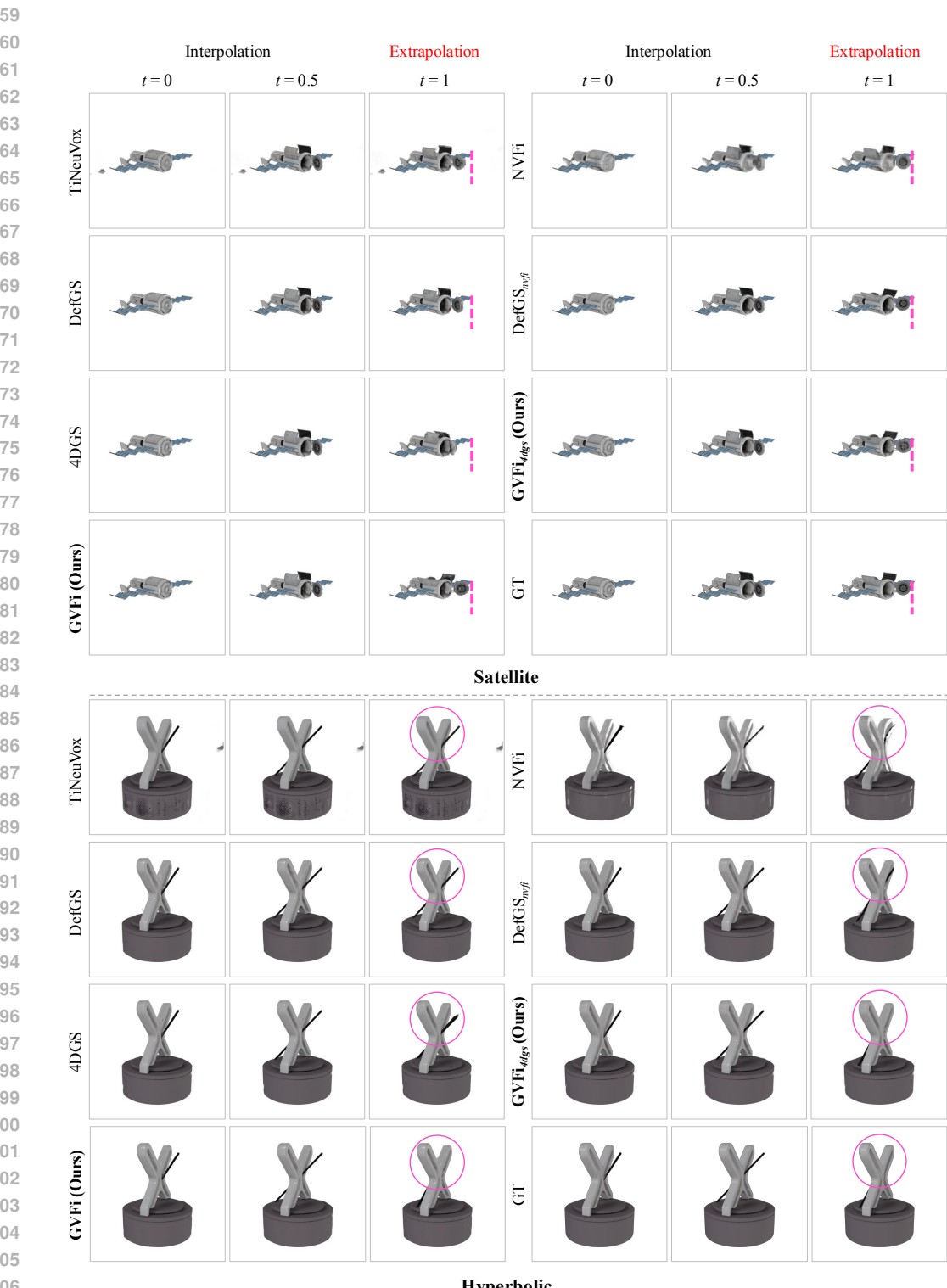

Figure 12: Qualitative results of RGB view synthesis for interpolation and extrapolation tasks on Dynamic Multipart dataset.

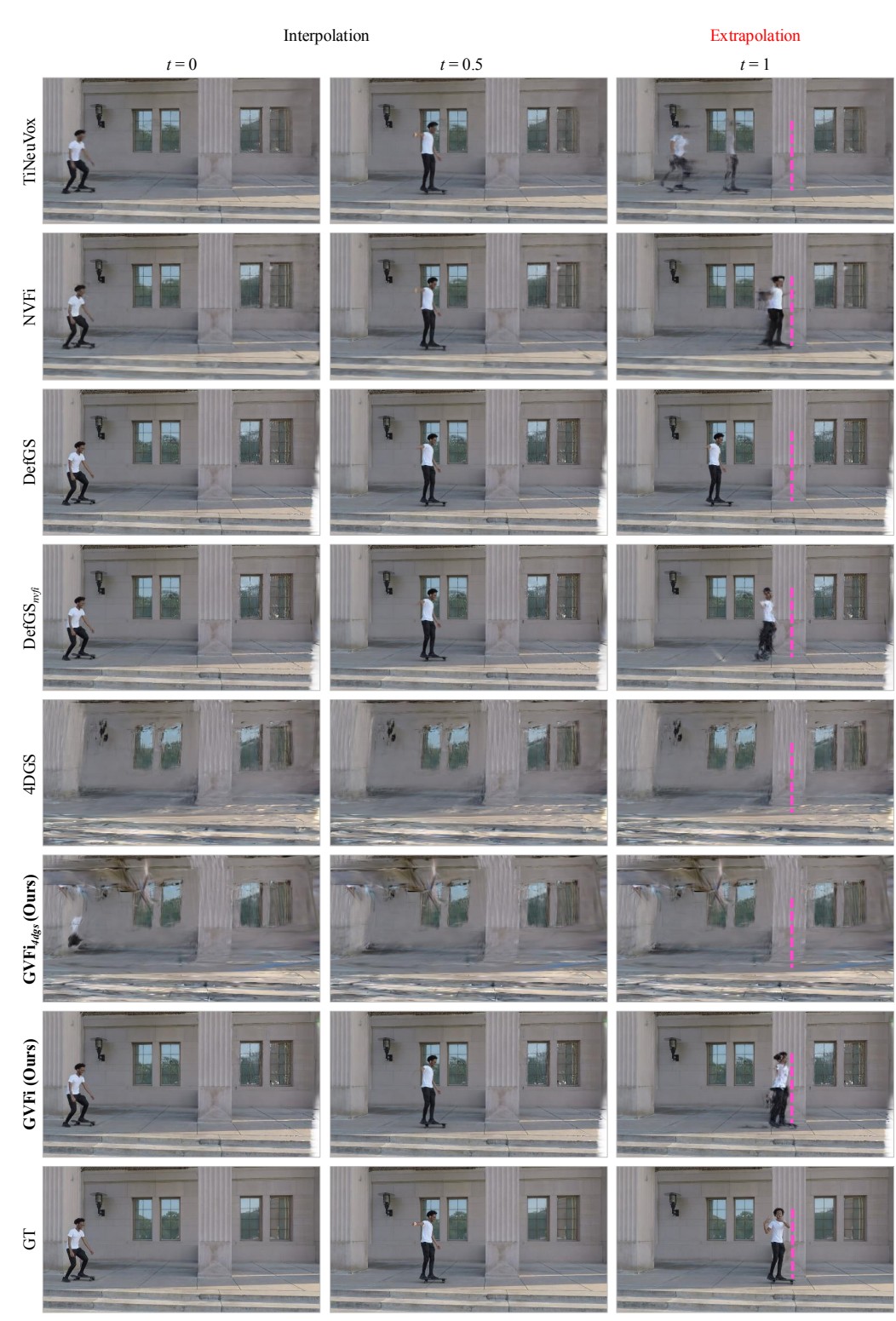

Figure 13: Qualitative results of RGB view synthesis for interpolation and extrapolation tasks on "Skating" scene of NVIDIA Dynamic Scene dataset.

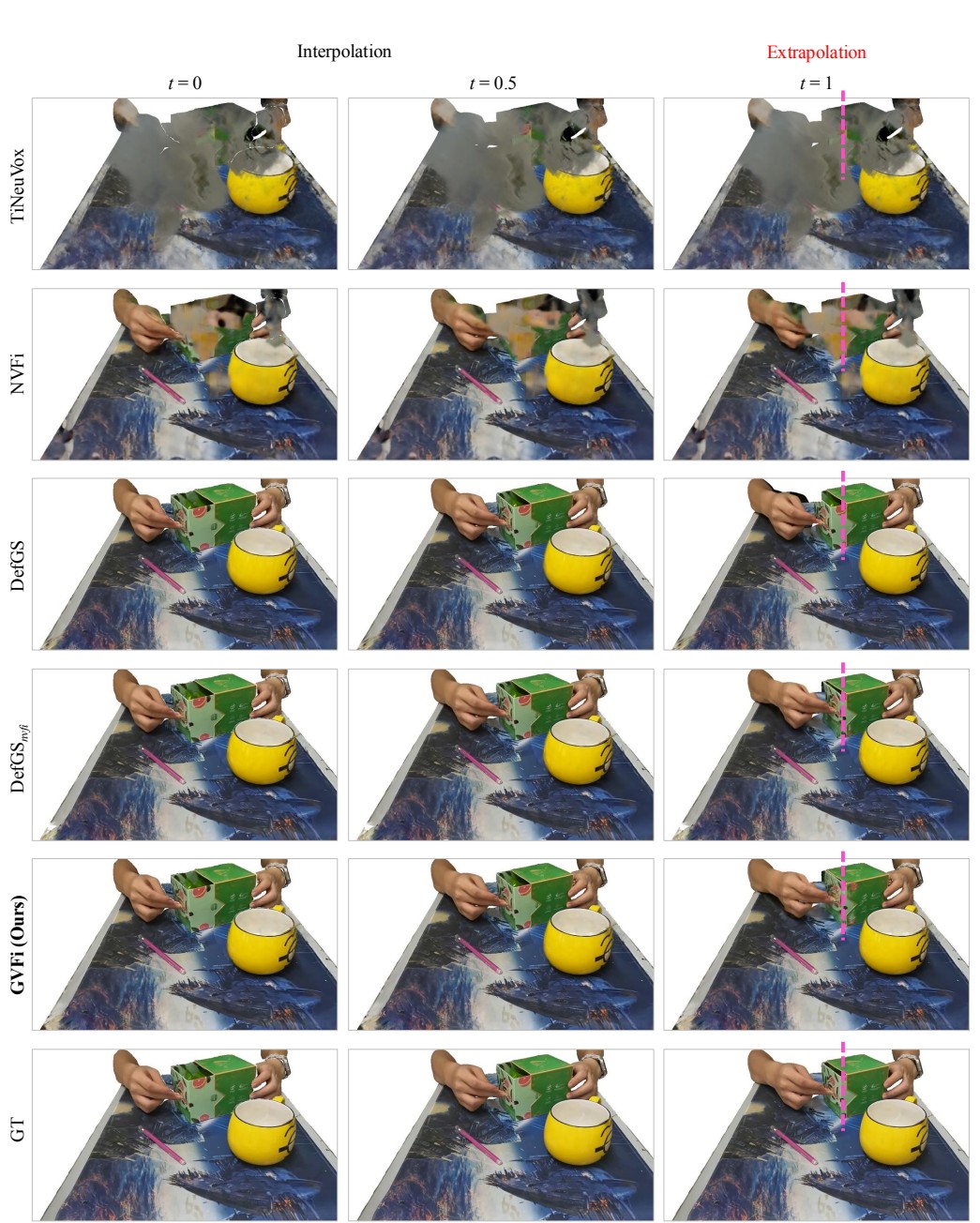

Figure 14: Qualitative results of RGB view synthesis for interpolation and extrapolation tasks on "Box" scene of GoPro dataset.

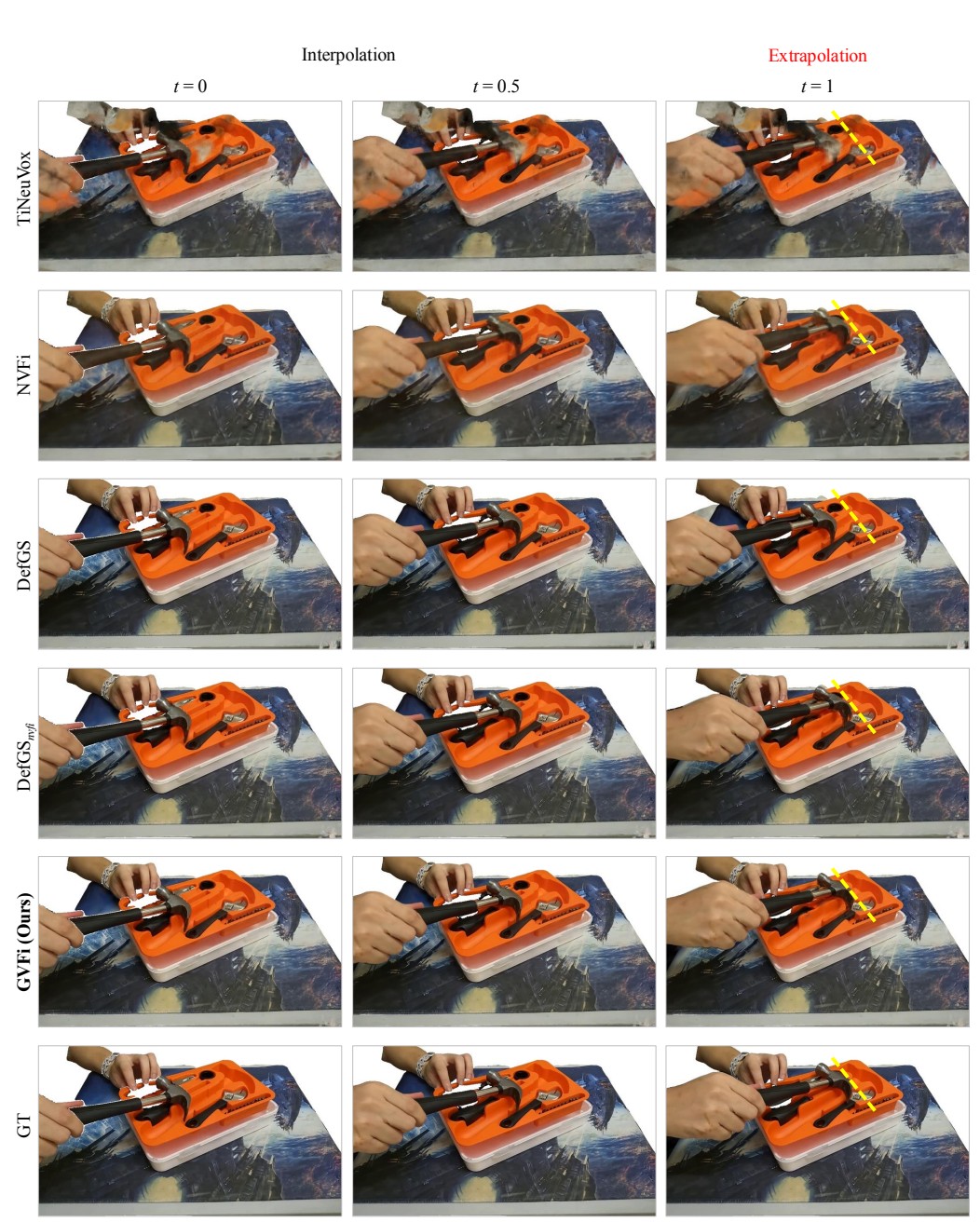

Figure 15: Qualitative results of RGB view synthesis for interpolation and extrapolation tasks on "Hammer" scene of GoPro dataset.

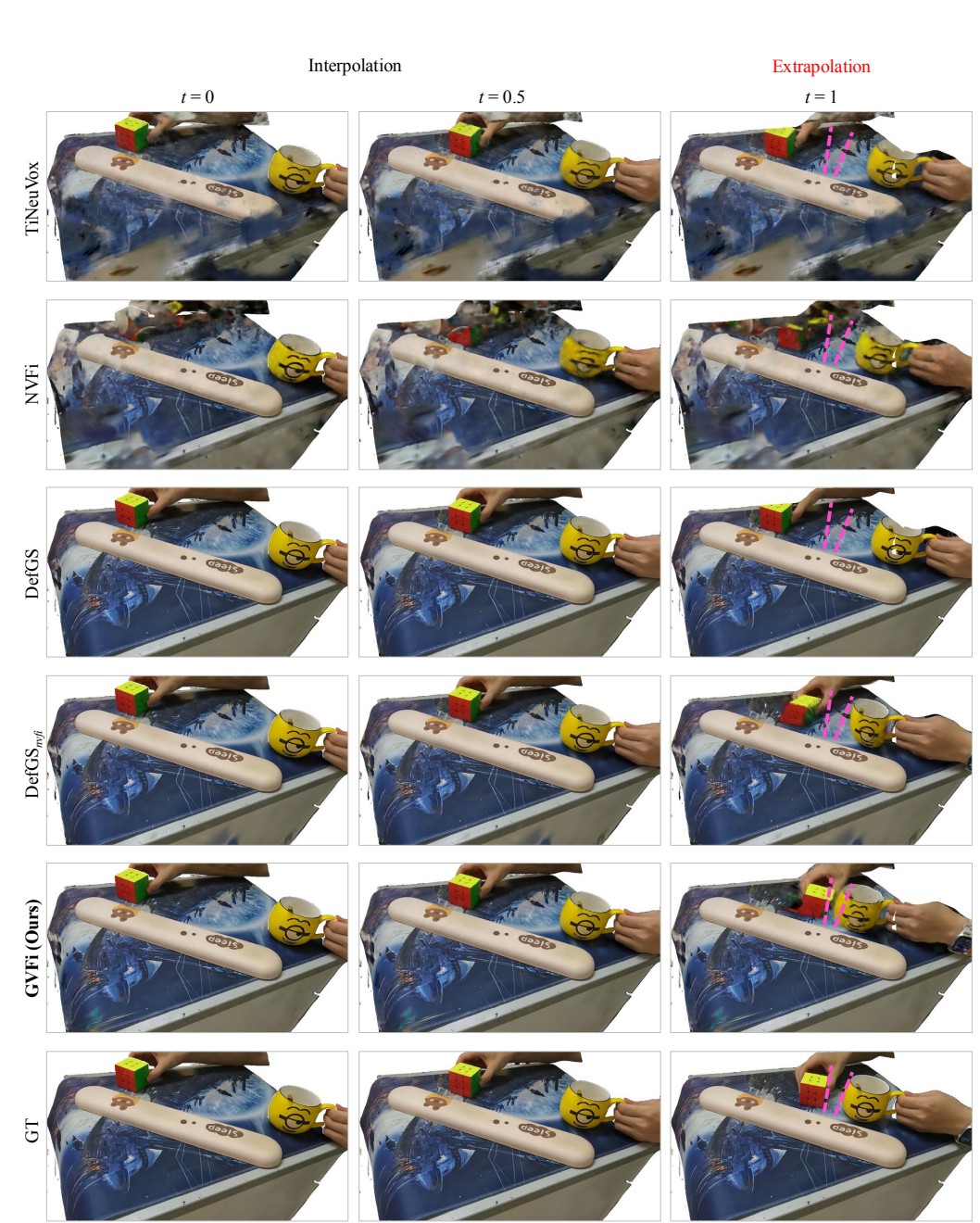

Figure 16: Qualitative results of RGB view synthesis for interpolation and extrapolation tasks on "Collision" scene of GoPro dataset.

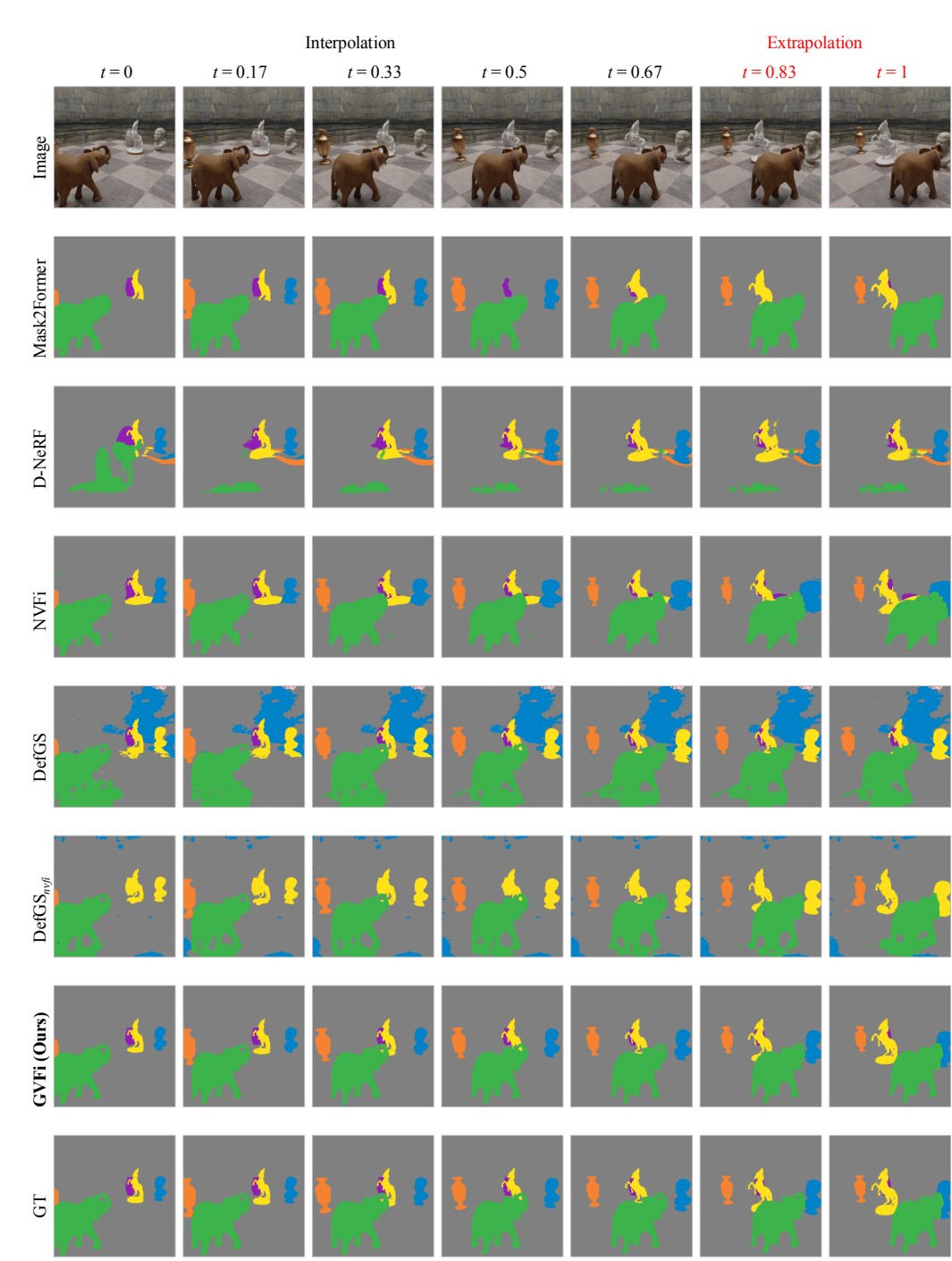

Figure 17: Qualitative results for object segmentation on "Chessboard" of Dynamic Indoor Scene dataset.

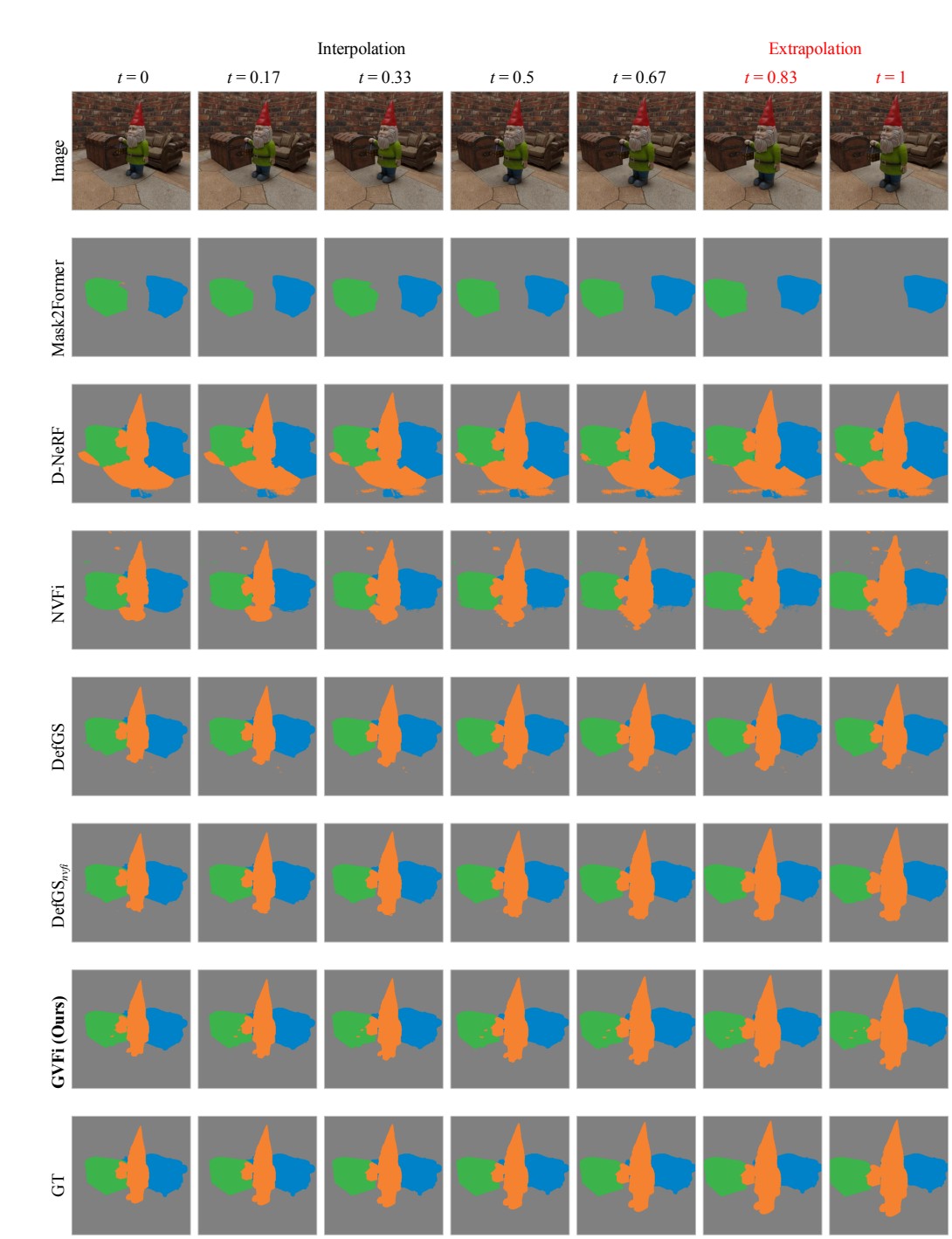

Figure 18: Qualitative results for object segmentation on "Gnome House" of Dynamic Indoor Scene dataset.

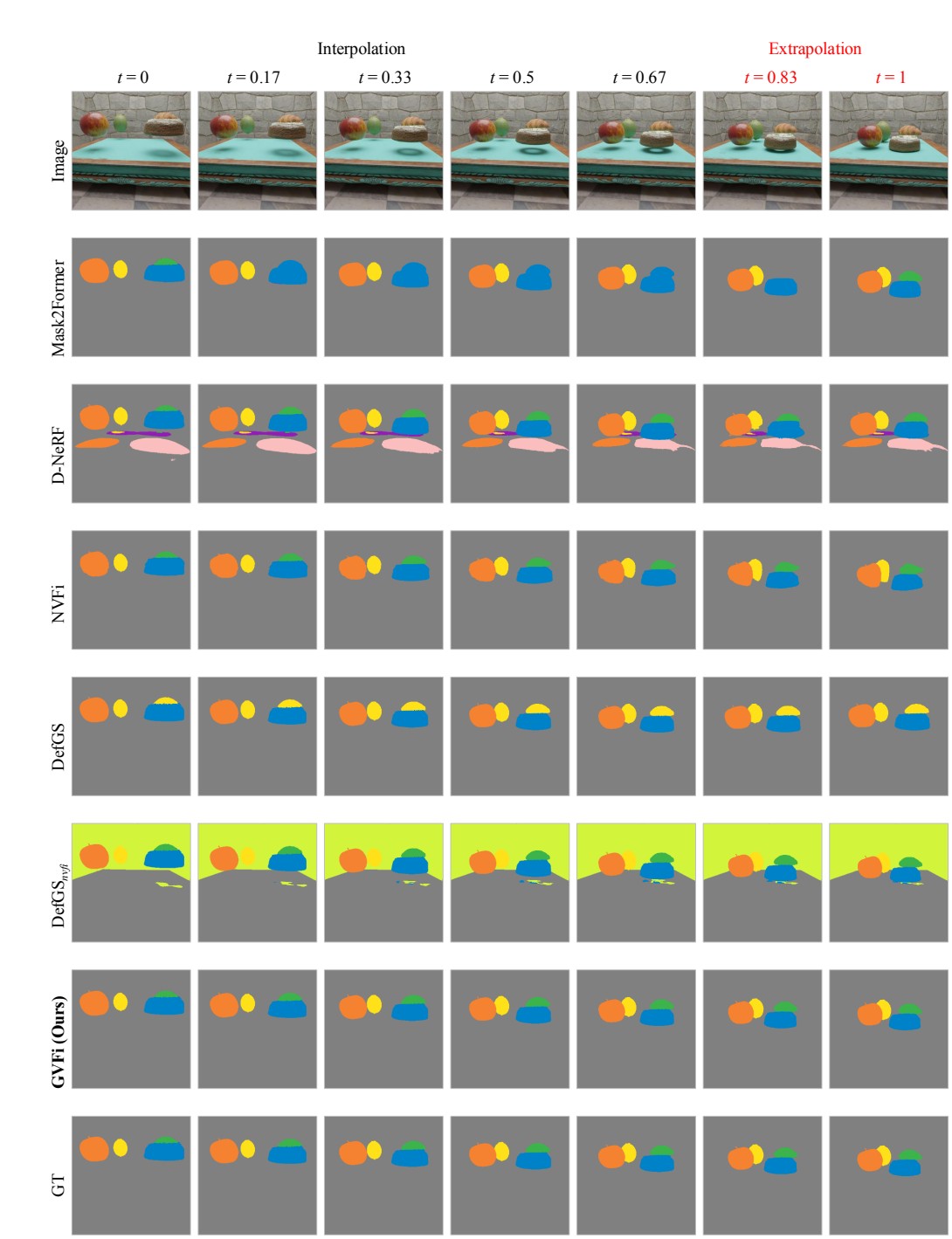

Figure 19: Qualitative results for object segmentation on "Dining Table" of Dynamic Indoor Scene dataset.

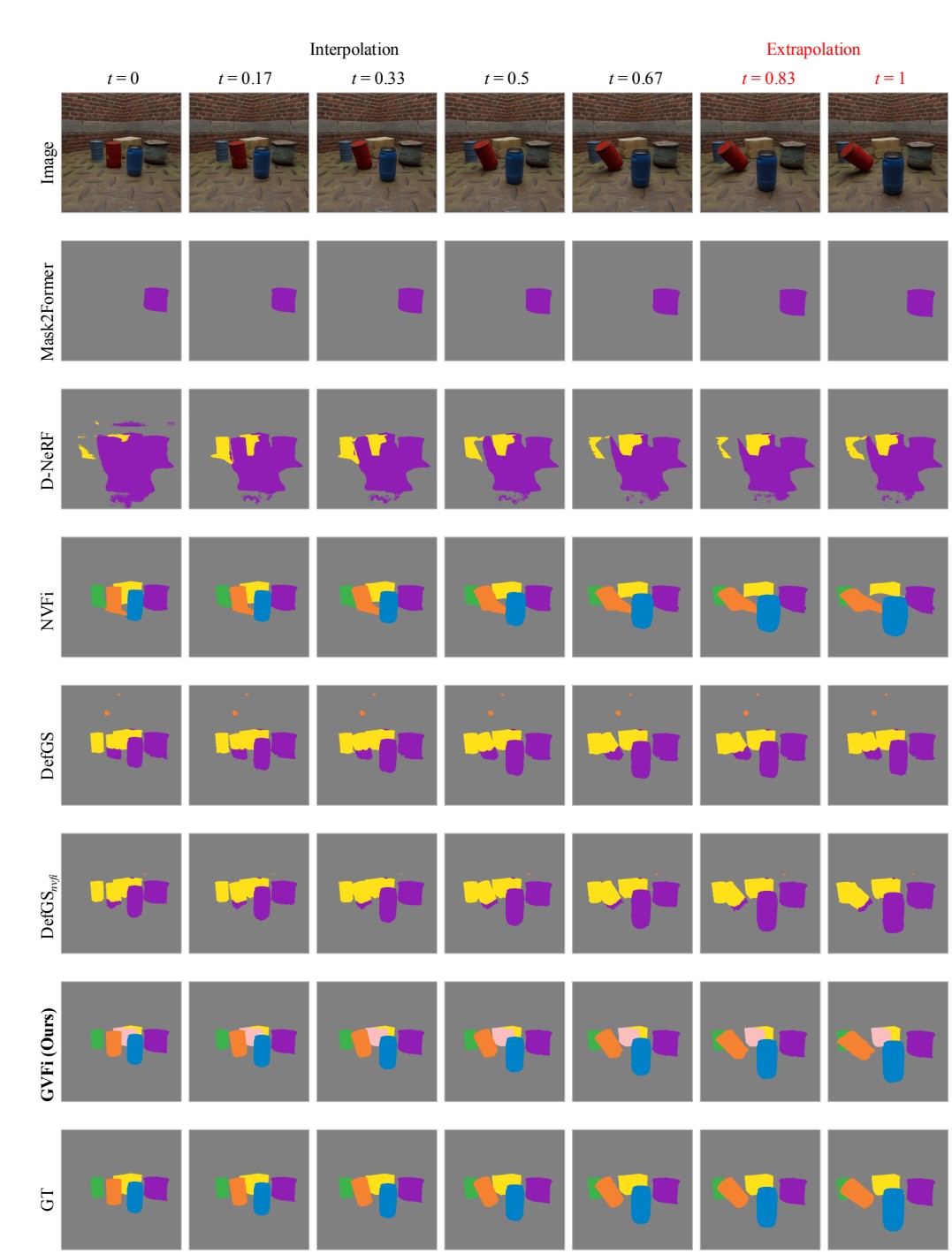

Figure 20: Qualitative results for object segmentation on "Factory" of Dynamic Indoor Scene dataset.

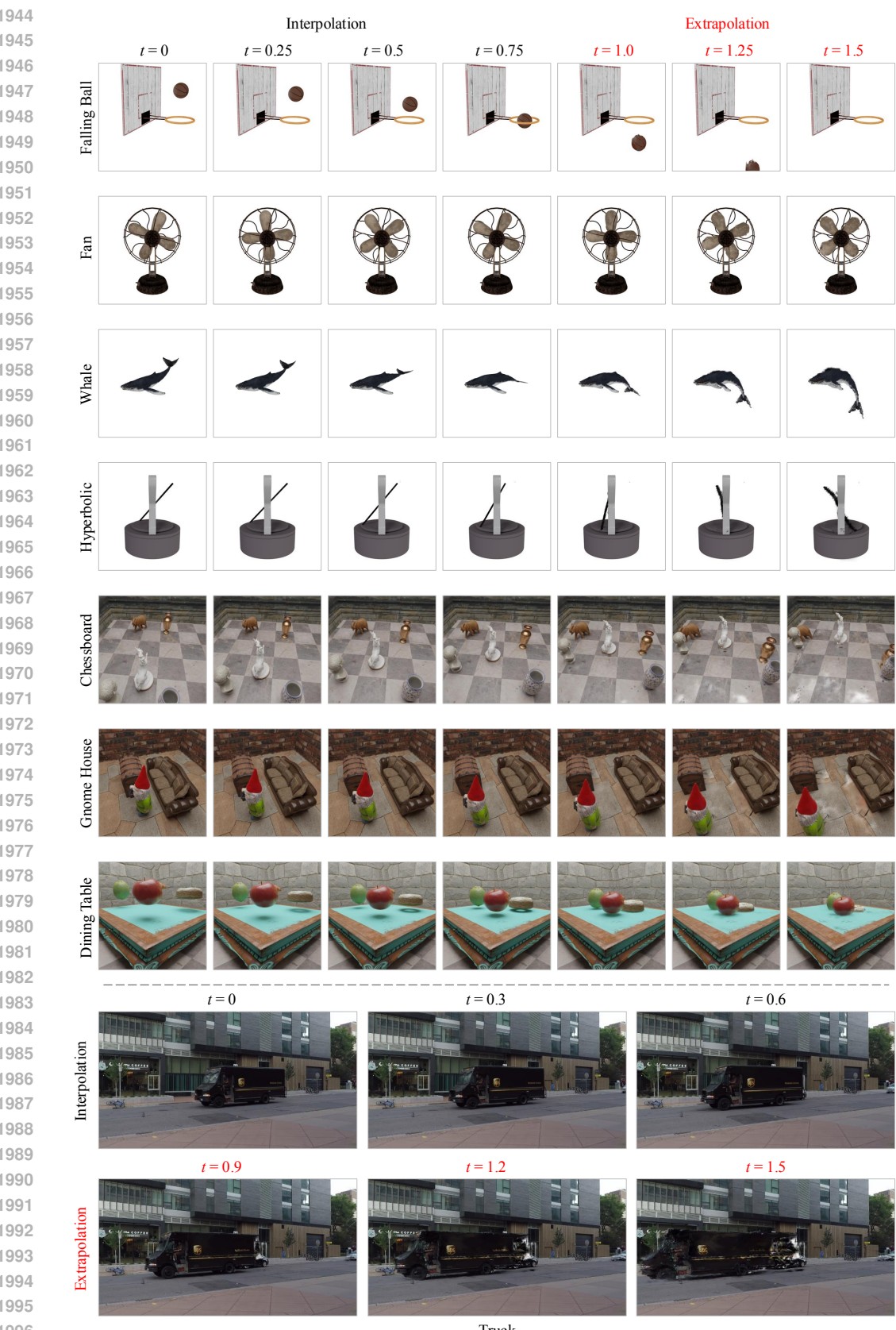

Figure 21: Qualitative results of RGB view synthesis for longer extrapolation from our method.

