# OpenReview forum: "GVFi: Learning 3D Gaussian Velocity Fields from Dynamic Videos"
_ICLR.cc/2025/Conference — Submitted to ICLR 2025_

### Official Review · Reviewer_i7XQ · 2024-10-28

**Soundness:** 3
**Presentation:** 2
**Contribution:** 3
**Rating:** 6
**Confidence:** 3

**Summary:**

Paper proposes a method "GVFi"  tackles the problem of estimating dynamic 3D scenes.

Broadly speaking, GVFi
 - Uses an off the shelf method (3DGS) to compute gaussian splats in a canonical frame
 - Uses an off the shelf method ("Deformable 3D Gaussians for High-Fidelity Monocular Dynamic Scene Reconstruction" Yang et al., CVPR 2024) to estimate a deformation field over position, rotation, and scale of each gaussian as a function of time
 - Uses these as inputs to then estimate the 3D gaussian's motion

Importantly, these gaussians are parameterized as rotation around a moving rotation centerpoint, and this centerpoint's motion is described entirely by an initial position, velocity, and acceleration estimate. These estimates are then optimized against the flow field as noisy ground truth and training observation reconstruction losses.

**Strengths:**

- Method at its core is quite simple (this is a good thing)
   - Learning a second order taylor series expansion of the full trajectory
 - The quantitative results seem good, even if only minor improvements in a number of cases

**Weaknesses:**

- Second order taylor series expansion seems quite limiting for arbitrary motion, or motion over non-trivial time horizons
 - Assuming I am interpreting the paper correctly, experiments seem to be only over short (~1 second) time horizons, which don't seem like they would challenge this assumption
 - Presentation quality is *extremely* poor
   - Core concept is quite simple, but it's heavily obfuscated for no apparent reason. It could be explained in 1 paragraph.
   - Core concepts seem poorly motivated; physics priors are common, but why only a second order expansion? Is this really a reasonable assumption in practice? There needs to be more motivation to this choice and more careful analysis of its limitations
   - Figure 1 and 2 are almost the same thing but not very informative. A better figure would be demonstrating the taylor series expansion of a single gaussian's trajectory
   - The math in section 3 does not feel like it was put there to be informative, but instead to intimidate the reader; after climbing through the notation its basically just saying to compose offsets together to estimate motion. If the authors feel this notational exercise is needed (don't think it is), it should go in the appendix and the main paper should have far more explanatory figures.
 - Ablations do not seem to address the core contribution, which is the assumption of the second order expansion --- what if you only do a first order expansion? Can you attempt to extend this to third order? They briefly mention replacing it with an MLP, but minimal details are provided.

I'm of the opinion that the paper has a neat idea but its presentation needs to be dramatically overhauled --- its assumptions need to be clearly stated and examined as reasonable or not, and it needs to have experiments where the method is pushed. Looking at the qualitative results, these datasets are very simple partwise rigid motion and the taylor series expansion is a nice trick to force smooth non-shattering motion, but it comes at the cost of generality --- nowhere does this seem to be addressed, considering the sometimes marginal performance improvements over far more flexible prior methods.

Nit:
"Cononical" -> Canonical misspelling is rampant

**Questions:**

- How long are each of the datasets scenes? Are they really long enough to meaningfully challenge the assumption of second order expansion?
- The NVIDIA Dynamic Scene Dataset (Yoon 2020) contains many dynamic scenes in the 2020 paper, but this paper claims "it consists of two real-world dynamic 3D scenes", what are those scenes?

---

> ### Author Response · Authors · 2024-11-25
>
> We appreciate the reviewer's valuable comments and address the concerns below.
>
> **Q1: Second order taylor series expansion seems quite limiting for arbitrary motion, or motion over non-trivial time horizons.**
>
> **A1:** For clarification, our scheme in Equation 4 indeed follows a second-order relationship to update dynamics parameters for each rigid particle from time $0$ to $t$. It captures up to a constant acceleration from $0$ to $t$, meaning that forces are allowed to generate accelerations and transfer energy.
>
> Theoretically, our updating scheme in Equation 4 can be easily extended to higher orders to capture extremely complex dynamics such as self-propelled objects (Refer to first-/third- order ablations in below response A7).
>
> In addition, as suggested by the reviewer **j1SG**, a simple sliding window based approach can also be applied to continuously and incrementally predict future frames given the newest visual observations from sensors, such that the complex dynamics can be well-captured.
>
> To validate this, we conduct experiments for incremental learning on three self-propelled objects from the Dynamic Object Dataset. To be specific, we first feed time $t=0\sim 0.15$ to train the network, and evaluate novel view interpolation on $t=0\sim 0.15$,  future frame extrapolation on $t=0.15\sim 0.30$. Next, we include $t=0.15\sim 0.30$ to train, and evaluate novel view interpolation on $t=0\sim 0.30$, future frame extrapolation on $t=0.30\sim 0.45$. We keep adding a time interval of 0.15 till we train from $t=0\sim 0.75$, and extrapolate from $t=0.75\sim 0.9$.
>
> The following Table (Table 5 in revised paper) shows quantitative results. It can be seen that DefGS suffers from overfitting the previous timestamps and its interpolation performance decreases, while our model can stably adapt to new observations and achieve excellent past and future frame predictions. This means that even though the internal forces are changing for self-propelled objects, our model can easily adapt to new observations.
>
> In the revised paper, we have rephrased the descriptions about Equation 4 in lines 235-254 of Section 3.2.
>
> **Table:** *Quantitative results (PSNR) of incremental learning.*
> | Interpolation | $0.15\rightarrow0.30$ | $0.30\rightarrow0.45$ | $0.45\rightarrow0.60$ | $0.60\rightarrow0.75$ | $0.75\rightarrow0.90$ | Average |
> |---:|:---:|:---:|:---:|:---:|:---:|:---:|
> | DefGS | 39.386 | 38.745 | 35.818 | 34.531 | 27.904 | 35.277 |
> | **GVFi (Ours)** | 40.032 | 40.706 | 41.013 | 40.466 | 39.971 | 40.438 |
>
> | Extrapolation | $0.15\rightarrow0.30$ | $0.30\rightarrow0.45$ | $0.45\rightarrow0.60$ | $0.60\rightarrow0.75$ | $0.75\rightarrow0.90$ | Average |
> |---:|:---:|:---:|:---:|:---:|:---:|:---:|
> | DefGS | 23.438 | 21.360 | 19.989 | 19.670 | 17.629 | 20.417 |
> | **GVFi (Ours)** | 29.958 | 32.260 | 31.384 | 29.527 | 28.958 | 30.417 |
>
> **Q2: Assuming I am interpreting the paper correctly, experiments seem to be only over short (\~1 second) time horizons, which don't seem like they would challenge this assumption.**
>
> **A2:** For clarification, the total time interval $0\sim 1$ is normalized (virtual) time for easy processing. In practice, it can surely be more than 1 second.
>
> As also requested by the reviewer **3rJ7**, we further conduct experiments for much longer extrapolation. Particularly, in our main experiments, training period lasts from $t=0\sim 0.75$ and extrapolation period lasts from $t=0.75\sim 1.0$. Here we show the results till $t=1.5$, which is already twice the training period.
>
> As shown in Figure 21 of Appendix A.15 in the revised paper, we provide qualitative results of longer extrapolation from the total four datasets. Note that, we are unable to provide quantitative results due to the lack of ground truth images. We can see that our method can still obtain physically meaningful future frame prediction in particularly high quality.
>
> **Q3: Presentation quality is extremely poor. Core concept is quite simple, but it's heavily obfuscated for no apparent reason. It could be explained in 1 paragraph.**
>
> **A3:** In the revised paper, in Sections 3.2\&3.3, we have condensed the core techniques of the proposed translation rotation dynamics system. The revised version is now more concise.

---

> ### Author Response · Authors · 2024-11-25
>
> **Q4: Core concepts seem poorly motivated; physics priors are common, but why only a second order expansion? Is this really a reasonable assumption in practice? There needs to be more motivation to this choice and more careful analysis of its limitations.**
>
> **A4:** As requested, in the revised paper, we have rephrased the assumption, motivations and scopes of our method, particularly clarified in lines 235-254 of Section 3.2.
>
> In addition, we also conduct two more groups of experiments to validate the effectiveness of our method on complex 3D scenes:  1) sliding window based incremental learning (refer to our response in A1), and 2) ablations of first-/third- order updating schemes in Equation 4 (refer to our response in below A7).
>
> **Q5: Figure 1 and 2 are almost the same thing but not very informative. A better figure would be demonstrating the taylor series expansion of a single gaussian's trajectory.**
>
> **A5:** This is a very helpful comment. In the revised paper, we have redrawn Figure 2, clearly illustrating the underlying trajectory governed by the learned physical parameters.
>
> **Q6: The math in section 3 does not feel like it was put there to be informative, but instead to intimidate the reader; after climbing through the notation its basically just saying to compose offsets together to estimate motion. If the authors feel this notational exercise is needed (don't think it is), it should go in the appendix and the main paper should have far more explanatory figures.**
>
> **A6:** Thanks for the suggestion. In the revised paper, in Sections 3.2\&3.3, we have consolidated and simplified the core equations, whilst keeping the logic flow fluent and sufficient for a broader audience.
>
> **Q7: Ablations do not seem to address the core contribution, which is the assumption of the second order expansion --- what if you only do a first order expansion? Can you attempt to extend this to third order? They briefly mention replacing it with an MLP, but minimal details are provided.**
>
> **A7:** This is a very insightful comment. As requested, we further conduct ablation experiments for choosing first-/third- order relationships in our Equation 4 on Dynamic Object Dataset and Dynamic Multipart Dataset. The following Table (Table 8 in revised paper) shows the results. We can see that, in Dynamic Object Dataset which has several self-propelled objects whose internal forces tend to change over time, not surprisingly, the third-order variant performs better. Nevertheless, due to the inherent over-parametrization, the third-order scheme tends to learn excessive rotation information to represent simple acceleration motions, thus incurring inferior performance on the Dynamic Multipart Dataset which does not have self-propelled objects. Overall, it is indeed interesting yet non-trivial to learn much higher-order relationships and we leave it for future exploration.
>
> Regarding our ablation of ``replacing it with an MLP", we originally aim to keep the physics parameters changing over time, thus making it more complex in theory. Particularly, we use the same network architecture of $f_{trd}$, except changing the input from $f_{trd}(\boldsymbol{x})$ to $f_{trd}(\boldsymbol{x}, t)$, to force the change of physics parameters. Nevertheless, its performance is inferior due to the lack of physics consistency over time, as detailed in Appendix A.7.
>
> In the revised paper, we have added the new first-/third- order ablations in Table 8 of Appendix A.8.
>
> **Table:** *Quantitative results of ablation studies about 3 orders of Taylor expansion on Dynamic Multipart dataset and Dynamic Object Dataset.*
> |  | **Dynamic Multipart Dataset** |  |  |  |  |  | **Dynamic Object Dataset** |  |  |  |  |  |
> |---|:---:|:---:|:---:|:---:|:---:|:---:|:---:|:---:|:---:|:---:|:---:|:---:|
> |  |  | Interpolation |  |  | Extrapolation |  |  | Interpolation |  |  | Extrapolation |  |
> |  | PSNR$\uparrow$ | SSIM$\uparrow$ | LPIPS$\downarrow$ | PSNR$\uparrow$ | SSIM$\uparrow$ | LPIPS$\downarrow$ | PSNR$\uparrow$ | SSIM$\uparrow$ | LPIPS$\downarrow$ | PSNR$\uparrow$ | SSIM$\uparrow$ | LPIPS$\downarrow$ |
> | $1^{st}$-order | 34.776 | 0.990 | 0.013 | 26.729 | 0.976 | 0.018 | 38.892 | **0.995** | **0.005** | 28.536 | **0.983** | 0.012 |
> | $2^{nd}$-order | 34.807 | **0.991** | **0.011** | **30.721** | **0.986** | **0.012** | 38.788 | **0.995** | 0.006 | 28.758 | 0.982 | **0.011** |
> | $3^{rd}$-order | **35.268** | **0.991** | 0.012 | 30.503 | 0.985 | 0.013 | **39.164** | **0.995** | **0.005** | **29.378** | **0.983** | **0.011** |

---

> > ### Author Response · Authors · 2024-11-25
> >
> > **Q8: I'm of the opinion that the paper has a neat idea but its presentation needs to be dramatically overhauled --- its assumptions need to be clearly stated and examined as reasonable or not, and it needs to have experiments where the method is pushed. Looking at the qualitative results, these datasets are very simple partwise rigid motion and the taylor series expansion is a nice trick to force smooth non-shattering motion, but it comes at the cost of generality --- nowhere does this seem to be addressed, considering the sometimes marginal performance improvements over far more flexible prior methods.**
> >
> > **A8:** As requested, in the revised paper, we have: 1) rephrased the assumption, motivation and scope of our method; 2) included more ablation results on first-/ third- order relationships; 3) collected and evaluated on a new challenging real-world dataset; 4) conducted incremental learning for complex dynamic 3D scenes; 5) conducted extremely long extrapolation on four datasets; 6) added two new recent baselines 4DGS and E-3DGS, and more.
> >
> > Most notably, compared with all existing NeRF-based and 3DGS-based methods including T-NeRF/D-NeRF/TiNeuVox/DefGS/4DGS/E-D3DGS, our method surpasses them by at least 5-10 points on PSNR for the core problem of future frame extrapolation on five datasets. The state-of-the-art method NVFi and another baseline built by us are also clearly inferior to our method for future frame extrapolation.
> >
> > To the best of our knowledge, there are no ``far more flexible prior methods" for future frame extrapolation. We are happy to compare if the reviewer suggests such new methods. Overall, we believe the reviewer's core concerns have been clearly addressed in our revised paper.
> >
> > **Q9: Nit: "Cononical" $\rightarrow$ Canonical misspelling is rampant.**
> >
> > **A9:** Typos fixed.
> >
> > **Q10: How long are each of the datasets scenes? Are they really long enough to meaningfully challenge the assumption of second order expansion?**
> >
> > **A10:** Refer to A2.
> >
> > **Q11: The NVIDIA Dynamic Scene Dataset (Yoon 2020) contains many dynamic scenes in the 2020 paper, but this paper claims "it consists of two real-world dynamic 3D scenes", what are those scenes?**
> >
> > **A11:** For clarification, our experiment setting on NVIDIA Dynamic Scene Dataset exactly follows prior work NVFi for a fair comparison. The two scenes (*skating man* and *moving truck*) are selected by NVFi, not us.
> >
> > To further validate the effectiveness of our method on challenging real-world scenes, we collect a new challenging real-world dataset by 20 GoPro cameras, named **GoPro Dataset**. Our dataset captures 4 dynamic scenes. For each dynamic scene, we select 89 frames from each view, and resize images to be a resolution of $960\times540$. We reserve the first 67 frames at 17 picked viewing angles as the training split, *i.e.*, 1139 frames, while leaving the 67 frames at the remaining 3 viewing angles for evaluating *novel view interpolation* within the training time period, *i.e.*, 201 frames. We keep the last 22 frames at all 20 viewing angles for evaluating *future frame extrapolation*, *i.e.*, 440 frames in total. More details are in Appendix A.6.
> >
> > The following Table (Table 3 and Table 14 in revised paper) shows the results of our method and baselines. We can see that our method is significantly better than DefGS/NVFi/TiNeuVox and also surpasses the strongest baseline built by us, demonstrating the effectiveness of our method on challenging real-world 3D scenes.
> >
> > In the revised paper, we have added our new real-world dataset and results in Section 4.1 and Appendix A.14 Table 14.
> >
> > **Table:** *Quantitative results of all methods for both novel view interpolation and future frame extrapolation on GoPro data.*
> > |  |  |  | GoPro | Dataset |  |  |
> > |:---:|:---:|:---:|:---:|:---:|:---:|:---:|
> > |  |  | Interpolation |  |  | Extrapolation |  |
> > |  | PSNR$\uparrow$ | SSIM$\uparrow$ | LPIPS$\downarrow$ | PSNR$\uparrow$ | SSIM$\uparrow$ | LPIPS$\downarrow$ |
> > | TiNeuVox | 15.306 | 0.588 | 0.516 | 20.323 | 0.738 | 0.318 |
> > | NVFi | 14.229 | 0.568 | 0.569 | 19.879 | 0.736 | 0.415 |
> > | DefGS | 20.018 | **0.838** | **0.167** | 21.193 | 0.842 | 0.185 |
> > | DefGS$_{nvfi}$ | **20.254** | **0.838** | **0.167** | _25.469_ | _0.882_ | _0.141_ |
> > | **GVFi** (Ours) | _20.124_ | _0.834_ | _0.168_ | **26.276** | **0.890** | **0.131** |

---

> > > ### Comment · Reviewer_i7XQ · 2024-12-01
> > >
> > > Thank you to the authors for their thorough rebuttal. The additional experiments assuage my concerns as to if the method actually reasonably works and can handle non-cherry picked scenes.
> > >
> > > Overall, I think the actual core contribution is very incremental but a neat idea nonetheless and possibly the basis for future work moving the ball forward on thus further. Additionally, the presentation has been improved but still needs a lot more work. Given the additional experiments, I have changed my review score from a 3 to a 6 on the condition that the authors further _significantly_ work on the paper presentation --- honestly, a significant rewrite might be in order. There's no enforcement mechanism to ensure the authors actually do this, but given how hard they worked for the rebuttal experiments I feel like it's possible they can also clean up the writing.

---

> > > > ### Author Response · Authors · 2024-12-01
> > > > **Thanks**
> > > >
> > > > Dear reviewer i7XQ,
> > > >
> > > > Thank you very much for your valuable time and positive rating on our paper. Your insightful suggestions have greatly helped us in enhancing the manuscript.
> > > >
> > > > Yes, with our new rebuttal materials at hand, we are committed to further improving the paper. Specifically, we will further clarify our neat core concept and techniques as you suggested, ensuring a more comprehensive yet concise presentation.
> > > >
> > > > Best,
> > > > Authors

---

### Official Review · Reviewer_3rJ7 · 2024-11-02

**Soundness:** 2
**Presentation:** 2
**Contribution:** 2
**Rating:** 3
**Confidence:** 5

**Summary:**

The paper introduces GVFi, a framework for modeling the motion physics of complex dynamic 3D scenes using multi-view RGB videos without requiring additional annotations such as object shapes, types, or masks.
Building on Deformable3DGS, GVFi incorporates constraints based on the laws of classical mechanics to guide motion predictions, ensuring that the Gaussian deformation estimated by the MLP aligns more closely with physical principles. By assuming that motion adheres to the laws of classical mechanics and explicitly learning the associated motion parameters, GVFi is capable of performing effective extrapolation rendering, allowing it to predict frames beyond the observed time span. Experimental results show that GVFi significantly outperforms existing methods, particularly excelling in future frame extrapolation tasks.

**Strengths:**

1. Modeling the motion of Gaussians through a Translation Rotation Dynamics System grounded in classical mechanics, resulting in a concise and conceptually elegant framework with solid mathematical and physical foundations.
2. Introducing an effective method to train the motion parameters of the Translation Rotation Dynamics System, enabling the accurate estimation of translation and rotation dynamics for each particle in the scene.
4. By explicitly learning motion parameters under classical mechanics, enabling effective extrapolation to unobserved frames and presenting potential for generation tasks that require plausible future frames in dynamic 3D scenes.
3. The proposed approach is validated on two tasks, demonstrating superior performance compared to previous methods, highlighting its effectiveness in modeling motion dynamics in 3D scenes.

**Weaknesses:**

1. The contributions of this work are somewhat incremental, as most of the methodological design heavily overlaps with the baseline method, Deformable3DGS [1]. The key difference lies in the incorporation of dynamical principles, primarily to enable extrapolation capabilities rather than introducing fundamentally novel approaches.
2. The proposed motion modeling framework is overly restrictive, relying on an strong assumption of no external forces, disregarding energy transfer processes, and lacking the ability to handle non-rigid or nonlinear motion. These limitations significantly reduce the model’s applicability to real-world physics.
3. Due to its reliance on idealized assumptions and limited scope, the model struggles to handle complex, real-world motion dynamics where varied forces, interactions, and non-rigid behaviors are prevalent, limiting its utility for practical applications in diverse environments.

[1] Ziyi Yang, Xinyu Gao, Wen Zhou, Shaohui Jiao, Yuqing Zhang, and Xiaogang Jin. Deformable 3D Gaussians for High-Fidelity Monocular Dynamic Scene Reconstruction. CVPR, 2024.

**Questions:**

1. Based on the methodology, there seem to be three possible approaches for interpolation rendering: (1) directly using $f_{defo}$ to predict the deformation at the given time $t$, (2) progressively calculating the Gaussian deformation at the given time $t$ from time 0 using the motion parameters predicted by $f_{trd}$, or (3) following the steps described in lines L261-L269. Which approach was used in the experiments? Are the results consistent across these three methods?
2. For extrapolation rendering according to lines L261-L269, it seems feasible to use either the second or third approach from question 1. Which method was actually used by the authors? If the third approach was used, how does it perform over longer extrapolation periods? Could the authors provide visual results for extrapolations that extend beyond the time span covered in the dataset?
3. The choice of baseline methods for comparison appears limited. For a comprehensive evaluation, it would be beneficial to compare against state-of-the-art methods in dynamic scene reconstruction, such as 4D-GS[2] and more recent work like E-D3DGS [3], which both have architectures similar to Deformable3DGS but differ in their motion representation. Could the authors verify if the proposed Translation Rotation Dynamics System can be integrated into these methods and whether it would yield similar performance gains?
4. The authors claim that their framework is a general approach for modeling motion physics in complex dynamic 3D scenes. However, the datasets used, with only 60 frames in total, limit the complexity and extent of motion. Could the authors validate this claim by testing on more challenging synthetic and real-world datasets, such as the ParticleNeRF and PanopticSports datasets, to provide a more comprehensive evaluation of the framework’s effectiveness on complex scenes?
5. In the ablation study, the authors provide a rationale for their choice of $\delta t$, which is somewhat reasonable. However, this conclusion is based on results from only one dataset, which may not be sufficient, as each dataset could exhibit different motion characteristics. Could the authors clarify how to select an appropriate $\delta t$ in practice across diverse datasets?
6. The experimental details are insufficient, particularly regarding training time, required resources, storage size, and rendering speed. Could the authors provide more comprehensive information on these aspects?
7. Please ensure that all abbreviations and technical terms are clearly defined, with full explanations and necessary citations. In the related work section, it would be helpful to explicitly clarify the differences from relevant works wherever possible.

[2] Guanjun Wu, Taoran Yi, Jiemin Fang, Lingxi Xie, Xiaopeng Zhang, Wei Wei, Wenyu Liu, Qi Tian, and Xinggang Wang. 4d gaussian splatting for real-time dynamic scene rendering. In Proceedings of the IEEE/CVF Conference on Computer Vision and Pattern Recognition (CVPR), 2024.

[3] Jeongmin Bae, Seoha Kim, Youngsik Yun, Hahyun Lee, Gun Bang, and Youngjung Uh. Per- gaussian embedding-based deformation for deformable 3d gaussian splatting. In Proceedings of the European Conference on Computer Vision (ECCV), 2024.

---

> ### Author Response · Authors · 2024-11-25
>
> We appreciate the reviewer's valuable comments and address the concerns below.
>
> **Q1: The contributions of this work are somewhat incremental, as most of the methodological design heavily overlaps with the baseline method, Deformable3DGS [1]. The key difference lies in the incorporation of dynamical principles, primarily to enable extrapolation capabilities rather than introducing fundamentally novel approaches.**
>
> **A1:** For clarification, our core novelty is the introduced translation rotation dynamics system together with its effective optimization strategy, which allows us to truly learn physical parameters, ultimately achieving future frame extrapolation. By comparison, existing works such as DefGS/4DGS all fail to do so, fundamentally because they do do not learn underlying physics priors, though they perform well for past frame interpolation, as extensively verified in Tables 1\&2 in our paper.
>
> In addition, the use of DefGS as our auxiliary deformation field is actually not our novelty. In fact, our introduced translation rotation dynamics system is also amenable to other deformation fields such as 4DGS, achieving satisfactory performance as shown in the following Table 1 (hyperparameters not tuned due to limited time for rebuttal).
>
> To the best of our knowledge, we are the first to learn such a translation rotation dynamics system for modeling dynamic 3D scenes in literature, and we achieve state-of-the-art performance for future frame extrapolation on five datasets. This clearly demonstrates our significant novelty in the field of study.
>
> In the revised paper, we highlight our novelty in lines 93-99 of Section 1.
>
> **Table 1:** _Quantitative results of our method with 4DGS as the auxiliary deformation field on four datasets._
> |  | **Dynamic Multipart Dataset** |  |  |  |  |  | **Dynamic Object Dataset** |  |  |  |  |  |
> |---|:---:|:---:|:---:|:---:|:---:|:---:|:---:|:---:|:---:|:---:|:---:|:---:|
> |  |  | Interpolation |  |  | Extrapolation |  |  | Interpolation |  |  | Extrapolation |  |
> |  | PSNR$\uparrow$ | SSIM$\uparrow$ | LPIPS$\downarrow$ | PSNR$\uparrow$ | SSIM$\uparrow$ | LPIPS$\downarrow$ | PSNR$\uparrow$ | SSIM$\uparrow$ | LPIPS$\downarrow$ | PSNR$\uparrow$ | SSIM$\uparrow$ | LPIPS$\downarrow$ |
> | GVFi$_{4dgs}$ | **36.542** | **0.991** | 0.015 | **30.801** | _0.983_ | _0.016_ | 35.961 | 0.985 | 0.021 | _28.316_ | _0.978_ | _0.023_ |
> | GVFi | 34.807 | **0.991** | **0.011** | _30.721_ | **0.986** | **0.012** | **38.788** | **0.995** | **0.006** | **28.758** | **0.982** | **0.011** |
> |  | **Dynamic Indoor Scene Dataset** |  |  |  |  |  | **NVIDIA Dynamic Scenes Dataset** |  |  |  |  |  |
> |  |  | Interpolation |  |  | Extrapolation |  |  | Interpolation |  |  | Extrapolation |  |
> |  | PSNR$\uparrow$ | SSIM$\uparrow$ | LPIPS$\downarrow$ | PSNR$\uparrow$ | SSIM$\uparrow$ | LPIPS$\downarrow$ | PSNR$\uparrow$ | SSIM$\uparrow$ | LPIPS$\downarrow$ | PSNR$\uparrow$ | SSIM$\uparrow$ | LPIPS$\downarrow$ |
> | GVFi$_{4dgs}$ | 27.932 | 0.860 | 0.252 | _31.590_ | _0.909_ | _0.194_ | 18.995 | 0.448 | 0.544 | _22.706_ | _0.714_ | _0.400_ |
> | GVFi | **32.202** | **0.928** | **0.089** | **34.556** | **0.964** | **0.046** | **26.943** | **0.891** | **0.102** | **29.388** | **0.938** | **0.067** |
>
> **Q2: The proposed motion modeling framework is overly restrictive, relying on an strong assumption of no external forces, disregarding energy transfer processes, and lacking the ability to handle non-rigid or nonlinear motion. These limitations significantly reduce the model’s applicability to real-world physics.**
>
> **A2:** Thank you for pointing out the inaccurate descriptions about our assumption in the original paper.
>
> For clarification, our scheme in Equation 4 follows a second-order relationship to update dynamics parameters for each rigid particle from time $0$ to $t$. It captures up to a constant acceleration from $0$ to $t$, meaning that forces are indeed allowed to generate accelerations and transfer energy.
>
> Theoretically, our updating scheme in Equation 4 can be easily extended to higher orders to capture extremely complex dynamics such as self-propelled objects. In addition, as suggested by the reviewer **j1SG**, a simple sliding window based approach can be applied to continuously and incrementally predict future frames given the newest visual observations from sensors, such that the complex dynamics can be well-captured.
>
> More experiment results for learning complex dynamics are provided in the following response A3. In the revised paper, we have rephrased the descriptions about our assumption and scope in lines 235-254 of Section 3.2.

---

> ### Author Response · Authors · 2024-11-25
>
> **Q3: Due to its reliance on idealized assumptions and limited scope, the model struggles to handle complex, real-world motion dynamics where varied forces, interactions, and non-rigid behaviors are prevalent, limiting its utility for practical applications in diverse environments.**
>
> **A3:** Following our response in A2, we further validate the applicability of our updating scheme in Equation 4. In particular, we conduct the following two groups of experiments to learn complex dynamics: 1) sliding window based incremental learning (also requested by the reviewer **j1SG**), and 2) ablations of first-/third- order updating schemes in Equation 4 (also requested by the reviewer **i7XQ**).
>
> **1) incremental learning**: We conduct experiments on three self-propelled objects from the Dynamic Object Dataset. To be specific, we first feed time $t=0\sim 0.15$ to train the network, and evaluate novel view interpolation on $t=0\sim 0.15$,  future frame extrapolation on $t=0.15\sim 0.30$. Next, we include $t=0.15\sim 0.30$ to train, and evaluate novel view interpolation on $t=0\sim 0.30$, future frame extrapolation on $t=0.30\sim 0.45$. We keep adding a time interval of 0.15 till we train from $t=0\sim 0.75$, and extrapolate from $t=0.75\sim 0.9$.
>
> The following Table 3 (Table 5 in revised paper) shows quantitative results. It can be seen that DefGS suffers from overfitting the previous timestamps and  its interpolation performance decreases, while our model can stably adapt to new observations and achieve excellent past and future frame predictions. This means that even though the internal forces are changing for self-propelled objects, our model can easily adapt to new observations.
>
> **Table 2:** _Quantitative results (PSNR) of incremental learning._
> | Interpolation | $0.15\rightarrow0.30$ | $0.30\rightarrow0.45$ | $0.45\rightarrow0.60$ | $0.60\rightarrow0.75$ | $0.75\rightarrow0.90$ | Average |
> |---:|:---:|:---:|:---:|:---:|:---:|:---:|
> | DefGS | 39.386 | 38.745 | 35.818 | 34.531 | 27.904 | 35.277 |
> | **GVFi (Ours)** | 40.032 | 40.706 | 41.013 | 40.466 | 39.971 | 40.438 |
> | **Extrapolation** | $0.15\rightarrow0.30$ | $0.30\rightarrow0.45$ | $0.45\rightarrow0.60$ | $0.60\rightarrow0.75$ | $0.75\rightarrow0.90$ | **Average** |
> | DefGS | 23.438 | 21.360 | 19.989 | 19.670 | 17.629 | 20.417 |
> | **GVFi (Ours)** | 29.958 | 32.260 | 31.384 | 29.527 | 28.958 | 30.417 |
>
> **2) first-/third- order ablations**: We conduct ablation experiments for our Equation 4 on Dynamic Object Dataset and Dynamic Multipart Dataset.
>
> The following Table 2 (Table 8 in revised paper) shows the results. We can see that, in Dynamic Object Dataset which has several self-propelled objects whose internal forces tend to change over time, not surprisingly, the third-order variant performs better. Nevertheless, due to the inherent over-parametrization, the third-order scheme tends to learn excessive rotation information to represent simple acceleration motions, thus incurring inferior performance on the Dynamic Multipart Dataset which does not have self-propelled objects.
>
> **Table 3:** _Quantitative results of ablation studies about 3 orders of Taylor expansion in Equation 4 on Dynamic Multipart Dataset and Dynamic Object Dataset._
> |  | **Dynamic Multipart Dataset** |  |  |  |  |  | **Dynamic Object Dataset** |  |  |  |  |  |
> |---|:---:|:---:|:---:|:---:|:---:|:---:|:---:|:---:|:---:|:---:|:---:|:---:|
> |  |  | Interpolation |  |  | Extrapolation |  |  | Interpolation |  |  | Extrapolation |  |
> |  | PSNR$\uparrow$ | SSIM$\uparrow$ | LPIPS$\downarrow$ | PSNR$\uparrow$ | SSIM$\uparrow$ | LPIPS$\downarrow$ | PSNR$\uparrow$ | SSIM$\uparrow$ | LPIPS$\downarrow$ | PSNR$\uparrow$ | SSIM$\uparrow$ | LPIPS$\downarrow$ |
> | $1^{st}$-order | 34.776 | 0.990 | 0.013 | 26.729 | 0.976 | 0.018 | 38.892 | **0.995** | **0.005** | 28.536 | **0.983** | 0.012 |
> | $2^{nd}$-order | 34.807 | **0.991** | **0.011** | **30.721** | **0.986** | **0.012** | 38.788 | **0.995** | 0.006 | 28.758 | 0.982 | **0.011** |
> | $3^{rd}$-order | **35.268** | **0.991** | 0.012 | 30.503 | 0.985 | 0.013 | **39.164** | **0.995** | **0.005** | **29.378** | **0.983** | **0.011** |
>
> Overall, our method can actually tackle complex dynamics just by applying a sliding window based incremental learning, or simply extending to high-order relationships if needed. In the revised paper, all these new materials have been updated.

---

> ### Author Response · Authors · 2024-11-25
>
> **Q4: Questions 1) Based on the methodology, there seem to be three possible approaches for interpolation rendering: (1) directly using $f_{defo}$ to predict the deformation at the given time $t$, (2) progressively calculating the Gaussian deformation at the given time $t$ from time 0 using the motion parameters predicted by $f_{trd}$, or (3) following the steps described in lines L261-L269. Which approach was used in the experiments? Are the results consistent across these three methods?**
>
> **A4:** We use the third approach in the experiments. As requested, we also provide interpolation results using the three approaches in the following Table 4 (Table 9 in revised paper). We can see that, the first and the third approaches are not strictly consistent, but achieve very similar performance. However, for the second approach, the performance clearly decreases, we hypothesize that this is due to  accumulated errors in the autoregressive process.
>
> In the revised paper, we have clarified our interpolation settings in lines 368-373 of Section 4.1, and added these new results in Table 9 of Appendix A.9.
>
> **Table 4:** _Quantitative results of different interpolation approaches of our method on all four datasets._
> |  |  | **Dynamic Multipart** |  |  | **Dynamic Object** |  |  | **Indoor Scene** |  |  | **Dynamic Scenes** |  |
> |---|:---:|:---:|:---:|:---:|:---:|:---:|:---:|:---:|:---:|:---:|:---:|:---:|
> |  | PSNR$\uparrow$ | SSIM$\uparrow$ | LPIPS$\downarrow$ | PSNR$\uparrow$ | SSIM$\uparrow$ | LPIPS$\downarrow$ | PSNR$\uparrow$ | SSIM$\uparrow$ | LPIPS$\downarrow$ | PSNR$\uparrow$ | SSIM$\uparrow$ | LPIPS$\downarrow$ |
> | (1)$f_{defo}$ | **35.040** | **0.991** | **0.011** | 38.406 | **0.995** | **0.005** | **32.569** | **0.930** | **0.088** | **26.951** | **0.891** | **0.102** |
> | (2)$f_{trd}$ | 30.310 | 0.984 | 0.017 | 33.527 | 0.991 | 0.009 | 31.776 | 0.926 | 0.092 | 25.899 | 0.875 | 0.118 |
> | (3)$f_{defo}+f_{trd}$ | 34.807 | **0.991** | **0.011** | **38.788** | **0.995** | 0.006 | 32.202 | 0.928 | 0.089 | 26.943 | **0.891** | **0.102** |
>
> **Q5: Questions 2) For extrapolation rendering according to lines L261-L269, it seems feasible to use either the second or third approach from question 1. Which method was actually used by the authors? If the third approach was used, how does it perform over longer extrapolation periods? Could the authors provide visual results for extrapolations that extend beyond the time span covered in the dataset?**
>
> **A5:** We use the third approach. Our primary goal is to extrapolate meaningful future frames as a continuum of the last training observations, which is achieved by the third approach (_i.e._, following Steps \#1\#2\#3\#4 in Section 3.3 lines 305-313), whereas the progressively accumulated parameters used in the second approach are less meaningful due to accumulated errors.
>
> As requested, we further conduct experiments for much longer extrapolation. Particularly, in our main experiments, training period lasts from $t=0\sim 0.75$ and extrapolation period lasts from $t=0.75\sim 1.0$. Here we show the results till $t=1.5$, which is already twice the training period.
>
> As shown in Figure 21 of Appendix A.15 in the revised paper, we provide qualitative results of longer extrapolation from the total four datasets. Note that, we are unable to provide quantitative results due to the lack of ground truth images. We can see that our method can still obtain physically meaningful future frame prediction in particularly high quality.
>
> In the revised paper, we have clarified our extrapolation settings in lines 368-373 of Section 4.1, and added the new extrapolation results in Appendix A.15.

---

> ### Author Response · Authors · 2024-11-25
>
> **Q6: Questions 3) The choice of baseline methods for comparison appears limited. For a comprehensive evaluation, it would be beneficial to compare against state-of-the-art methods in dynamic scene reconstruction, such as 4D-GS[2] and more recent work like E-D3DGS [3], which both have architectures similar to Deformable3DGS but differ in their motion representation. Could the authors verify if the proposed Translation Rotation Dynamics System can be integrated into these methods and whether it would yield similar performance gains?**
>
> **A6:** Thank you for the valuable suggestions. As requested, in Tables 1\&2 of the revised paper (also shown in the following Table 5), we have added the recent 4DGS an E-D3DGS as additional baselines on all four datasets. We can see that, not surprisingly, both methods clearly fail to predict meaningful future frames due to the lack of physics learning, though they can achieve excellent performance for past frame interpolation.
>
> As requested, to demonstrate the flexibility of our framework, we also adopt 4DGS as our auxiliary deformation field, denoted as GVF$i_{4dgs}$. As showing in the following Table, our GVFi$_{4dgs}$ (hyperparameters not tuned due to the limited time for rebuttal) also achieves very good results for future frame extrapolation on most datasets.
>
> In the revised paper, all these new results are added to Section 4.1, further demonstrating the superiority of our method.
>
> **Table 5:** *Quantitative results of new baselines and our GVFi$_{4dgs}$  on all four datasets.*
> |  | **Dynamic Multipart Dataset** |  |  |  |  |  | **Dynamic Object Dataset** |  |  |  |  |  |
> |---|:---:|:---:|:---:|:---:|:---:|:---:|:---:|:---:|:---:|:---:|:---:|:---:|
> |  |  | Interpolation |  |  | Extrapolation |  |  | Interpolation |  |  | Extrapolation |  |
> |  | PSNR$\uparrow$ | SSIM$\uparrow$ | LPIPS$\downarrow$ | PSNR$\uparrow$ | SSIM$\uparrow$ | LPIPS$\downarrow$ | PSNR$\uparrow$ | SSIM$\uparrow$ | LPIPS$\downarrow$ | PSNR$\uparrow$ | SSIM$\uparrow$ | LPIPS$\downarrow$ |
> | E-D3DGS | 26.180 | 0.955 | 0.062 | 18.615 | 0.904 | 0.114 | 28.075 | 0.963 | 0.049 | 18.526 | 0.923 | 0.087 |
> | 4D-GS | **37.021** | **0.992** | _0.014_ | 20.564 | 0.935 | 0.067 | _37.285_ | _0.986_ | _0.020_ | 20.354 | 0.950 | 0.052 |
> | GVFi$_{4dgs}$ | _36.542_ | _0.991_ | 0.015 | **30.801** | _0.983_ | _0.016_ | 35.961 | 0.985 | 0.021 | _28.316_ | _0.978_ | _0.023_ |
> | GVFi | 34.807 | _0.991_ | **0.011** | _30.721_ | **0.986** | **0.012** | **38.788** | **0.995** | **0.006** | **28.758** | **0.982** | **0.011** |
> |  | **Dynamic Indoor Scene Dataset** |  |  |  |  |  | **NVIDIA Dynamic Scenes Dataset** |  |  |  |  |  |
> |  |  | Interpolation |  |  | Extrapolation |  |  | Interpolation |  |  | Extrapolation |  |
> |  | PSNR$\uparrow$ | SSIM$\uparrow$ | LPIPS$\downarrow$ | PSNR$\uparrow$ | SSIM$\uparrow$ | LPIPS$\downarrow$ | PSNR$\uparrow$ | SSIM$\uparrow$ | LPIPS$\downarrow$ | PSNR$\uparrow$ | SSIM$\uparrow$ | LPIPS$\downarrow$ |
> | E-D3DGS | 29.267 | 0.874 | 0.222 | 20.374 | 0.772 | 0.307 | _20.848_ | _0.541_ | _0.532_ | 20.301 | 0.565 | 0.522 |
> | 4D-GS | _29.381_ | _0.889_ | _0.212_ | 21.107 | 0.793 | 0.274 | 19.411 | 0.462 | _0.532_ | 22.510 | 0.703 | 0.408 |
> | GVFi$_{4dgs}$ | 27.932 | 0.860 | 0.252 | _31.590_ | _0.909_ | _0.194_ | 18.995 | 0.448 | 0.544 | _22.706_ | _0.714_ | _0.400_ |
> | GVFi | **32.202** | **0.928** | **0.089** | **34.556** | **0.964** | **0.046** | **26.943** | **0.891** | **0.102** | **29.388** | **0.938** | **0.067** |

---

> ### Author Response · Authors · 2024-11-25
>
> **Q7: Questions 4) The authors claim that their framework is a general approach for modeling motion physics in complex dynamic 3D scenes. However, the datasets used, with only 60 frames in total, limit the complexity and extent of motion. Could the authors validate this claim by testing on more challenging synthetic and real-world datasets, such as the ParticleNeRF and PanopticSports datasets, to provide a more comprehensive evaluation of the framework’s effectiveness on complex scenes?**
>
> **A7:** For clarification, the total 60 frames are sampled frames, and the total time interval $0\sim 1$ is also normalized (virtual) time for easy processing with limited computation resources. In principle, our method does not have specific requirements on the actual frame rate.
>
> Thank you for suggesting ParticleNeRF and PanopticSports datasets. After a close investigation of the two datasets, we found that ParticleNeRF involves springs or cloth and PanopticSports involves random human interactions. These dynamics are rather chaotic and beyond the scope of this paper given the limited rebuttal time. Nevertheless, we agree that it is interesting and we leave it for future exploration.
>
> Instead, we collect a new challenging real-world dataset by 20 GoPro cameras, named **GoPro Dataset**. Our dataset captures 4 dynamic scenes. For each dynamic scene, we select 89 frames from each view, and resize images to be a resolution of $960\times540$. We reserve the first 67 frames at 17 picked viewing angles as the training split, _i.e._, 1139 frames, while leaving the 67 frames at the remaining 3 viewing angles for evaluating _novel view interpolation_ within the training time period, _i.e._, 201 frames. We keep the last 22 frames at all 20 viewing angles for evaluating _future frame extrapolation_, _i.e._, 440 frames in total. More details are in Appendix A.6.
>
> The following Table 6 (Table 3 in revised paper) shows the results of our method and baselines. We can see that our method is significantly better than DefGS/NVFi/TiNeuVox and also surpasses the strongest baseline built by us, demonstrating the effectiveness of our method on challenging real-world 3D scenes.
>
> In the revised paper, we have added our newly collected dataset and the new results in Section 4, significantly improving the quality of our paper.
>
> **Table 6:** _Quantitative results for both novel view interpolation and future frame extrapolation on GoPro Dataset._
> |  |  |  | GoPro | Dataset |  |  |
> |:---:|:---:|:---:|:---:|:---:|:---:|:---:|
> |  |  | Interpolation |  |  | Extrapolation |  |
> |  | PSNR$\uparrow$ | SSIM$\uparrow$ | LPIPS$\downarrow$ | PSNR$\uparrow$ | SSIM$\uparrow$ | LPIPS$\downarrow$ |
> | TiNeuVox | 15.306 | 0.588 | 0.516 | 20.323 | 0.738 | 0.318 |
> | NVFi | 14.229 | 0.568 | 0.569 | 19.879 | 0.736 | 0.415 |
> | DefGS | 20.018 | **0.838** | **0.167** | 21.193 | 0.842 | 0.185 |
> | DefGS$_{nvfi}$ | **20.254** | **0.838** | **0.167** | _25.469_ | _0.882_ | _0.141_ |
> | **GVFi** (Ours) | _20.124_ | _0.834_ | _0.168_ | **26.276** | **0.890** | **0.131** |

---

> ### Author Response · Authors · 2024-11-25
>
> **Q8: Questions 5) In the ablation study, the authors provide a rationale for their choice of $\delta t$, which is somewhat reasonable. However, this conclusion is based on results from only one dataset, which may not be sufficient, as each dataset could exhibit different motion characteristics. Could the authors clarify how to select an appropriate $\delta t$ in practice across diverse datasets?**
>
> **A8:** Thanks for this insightful comment and advice. As requested, we further conduct extensive ablations about different choices of $\Delta t$ on all 4 datasets, and the results are listed in the following Table 7 (Table 7 in revised paper). We observe that $3\delta t$ works better in extrapolation on three datasets (Dynamic Object/ Dynamic Indoor Scene/ NVIDIA Dynamic Scenes). The basic rule to select an appropriate $\delta t$ is based on the motion range. If the motion changes fast, so the motion between two consecutive frames is apparent enough, then a smaller $\delta t$ is good enough. Otherwise, if the motion is rather slow, then a larger $\delta t$ is preferred.
>
> In the revised paper, we have added these new results in Table 7 of Appendix A.8.
>
> **Table 7:** Quantitative results of ablation studies for $\delta t$ on all four datasets.
> |  | **Dynamic Multipart Dataset** |  |  |  |  |  | **Dynamic Object Dataset** |  |  |  |  |  |
> |---|:---:|:---:|:---:|:---:|:---:|:---:|:---:|:---:|:---:|:---:|:---:|:---:|
> |  |  | Interpolation |  |  | Extrapolation |  |  | Interpolation |  |  | Extrapolation |  |
> |  | PSNR$\uparrow$ | SSIM$\uparrow$ | LPIPS$\downarrow$ | PSNR$\uparrow$ | SSIM$\uparrow$ | LPIPS$\downarrow$ | PSNR$\uparrow$ | SSIM$\uparrow$ | LPIPS$\downarrow$ | PSNR$\uparrow$ | SSIM$\uparrow$ | LPIPS$\downarrow$ |
> | $\delta t$ | 35.128 | **0.991** | **0.011** | 29.441 | 0.984 | 0.013 | **38.929** | **0.995** | **0.005** | 28.506 | 0.981 | 0.013 |
> | $2\delta t$ | 34.807 | **0.991** | **0.011** | **30.721** | **0.986** | **0.012** | 38.788 | **0.995** | 0.006 | 28.758 | 0.982 | **0.011** |
> | $3\delta t$ | **35.223** | **0.991** | **0.011** | 30.246 | 0.985 | **0.012** | 38.693 | **0.995** | 0.006 | **29.414** | **0.983** | 0.012 |
> |  | **Dynamic Indoor Scene Dataset** |  |  |  |  |  | **NVIDIA Dynamic Scenes Dataset** |  |  |  |  |  |
> |  |  | Interpolation |  |  | Extrapolation |  |  | Interpolation |  |  | Extrapolation |  |
> |  | PSNR$\uparrow$ | SSIM$\uparrow$ | LPIPS$\downarrow$ | PSNR$\uparrow$ | SSIM$\uparrow$ | LPIPS$\downarrow$ | PSNR$\uparrow$ | SSIM$\uparrow$ | LPIPS$\downarrow$ | PSNR$\uparrow$ | SSIM$\uparrow$ | LPIPS$\downarrow$ |
> | $\delta t$ | 32.179 | **0.929** | **0.089** | 34.387 | 0.964 | 0.046 | 26.823 | **0.891** | **0.101** | 28.781 | 0.934 | 0.070 |
> | $2\delta t$ | 32.202 | 0.928 | **0.089** | 34.556 | 0.964 | 0.046 | 26.943 | **0.891** | 0.102 | 29.388 | **0.938** | **0.067** |
> | $3\delta t$ | **32.296** | 0.928 | **0.089** | **35.242** | **0.967** | **0.045** | **27.099** | 0.890 | 0.103 | **29.440** | **0.938** | **0.067** |
>
> **Q9: Questions 6) The experimental details are insufficient, particularly regarding training time, required resources, storage size, and rendering speed. Could the authors provide more comprehensive information on these aspects?**
>
> **A9:** As requested, we have added comprehensive details in Appendix A.4 in the revised paper. Particularly:
>
> As the complexity of different scenes varies, the total number of Gaussians learned for each scene varies from 40k to 1.6M. In general, our training time is 1.05 times longer than DefGS (or 4DGS if built on it). For example, on the _bat_ of Dynamic Object Dataset, DefGS/4DGS need 25 minutes, while we need 27 minutes, with a slight training cost addition. Since our additional module is a tiny MLPs, we only need 367.4kB larger storage. Our rendering speed is 0.85 times slower than DefGS (or 0.8 times slower than 4DGS if built on it). For example, on the _bat_ of Dynamic Object Dataset, they achieve 40fps and ours 32fps. We train all our models on a single NVIDIA 3090 24G GPU.
>
> **Q10: Questions 7) Please ensure that all abbreviations and technical terms are clearly defined, with full explanations and necessary citations. In the related work section, it would be helpful to explicitly clarify the differences from relevant works wherever possible.**
>
> **A10:** In the revised paper, all abbreviations and terms are clearly defined. In lines 132-134 and 146-149, we have clarified the differences from related works.

---

> > ### Comment · Reviewer_3rJ7 · 2024-11-27
> >
> > Thank you for your detailed responses to my comments as well as those of the other reviewers, and for your efforts to improve the manuscript. I have carefully reviewed your replies, the revised paper, as well as the comments and feedback from the other reviewers. After thorough consideration, I have identified some critical issues that significantly limit the generality and originality of the proposed method. Consequently, I have decided to adjust my score from 5 to 3. Below, I outline the main reasons for this decision:
> >
> > ---
> >
> > ### 1. Limitations of the Core Assumptions
> >
> > This work relies on a strong assumption that the motion occurs without external forces and that the objects in motion are rigid. While this assumption has led to favorable performance in the reported experiments, particularly in the extrapolation mode, I believe that the performance advantage is largely due to the characteristics of the chosen dataset rather than the generalizability of the method.
> >
> > Nonetheless, the actual results reveal significant limitations that question the validity of this assumption. For instance, in Figure 13 of the revised paper, while the overall predictions for the skateboarder’s body are relatively accurate, closer examination reveals that the skateboarder’s hand undergoes noticeable deformations during motion, which clearly violates the rigidity assumption. This results in poor predictions for finer details and highlights the method’s inability to handle non-rigid components effectively in real-world scenarios. Furthermore, in the extrapolation mode, the skateboard itself is not reconstructed, which is a critical failure given its integral role in the motion context. These results cast doubt on the method’s robustness and its applicability beyond datasets that closely conform to the rigid-body assumption.
> >
> > ---
> >
> > ### 2. Applicability of the Method and Alternative Approaches
> >
> > I remain skeptical about the practicality of extrapolation based on such a strong assumption. In scenarios where this assumption holds true, simpler reconstruction-based editing approaches may achieve similar or even better outcomes. For example, after reconstruction, it is relatively straightforward to calculate motion properties such as velocity and momentum, which can then estimate the approximate future positions of objects.
> >
> > From the visual results provided by the authors (again, referring to Figure 13), the overall visual quality of the extrapolated objects is very poor, even when compared to what could potentially be achieved using the simpler editing method mentioned above. This raises serious doubts about the significance and practical utility of the extrapolation mode, and I strongly question whether it is meaningful in its current form.
> >
> > ---
> >
> > ### 3. Concerns Shared by Other Reviewers
> >
> > In addition to my own comments, I note that other reviewers have raised similar concerns. These include:
> >
> > - Limited Contribution: For example, Reviewer gAFC mentioned that “the novelty of this addition may be somewhat limited.”
> > - Rationale for Strong Assumptions: Reviewer j1SG stated, “My main concern about this work is the assumption made.”
> > - Experimental Design and Dataset Choices: Several reviewers, including myself, have highlighted that the dataset and experimental settings may not adequately validate the method’s applicability to more general or realistic scenarios.
> >
> > These consistent concerns suggest a broader consensus that the work, in its current state, does not sufficiently address its limitations or justify its assumptions.
> >
> > ---
> >
> > Based on these considerations, I believe the current work lacks sufficient evidence to demonstrate its generality, practicality, or originality. While the direction of the study has potential, significant improvements are necessary to strengthen its contributions. I hope these comments are helpful for your revisions, and I remain open to further discussion if needed.

---

> > > ### Author Response · Authors · 2024-11-30
> > >
> > > **Comment \#1: Limitations of the Core Assumptions -**
> > >
> > > **This work relies on a strong assumption that the motion occurs without external forces and that the objects in motion are rigid. While this assumption has led to favorable performance in the reported experiments, particularly in the extrapolation mode, I believe that the performance advantage is largely due to the characteristics of the chosen dataset rather than the generalizability of the method.**
> > >
> > > **Nonetheless, the actual results reveal significant limitations that question the validity of this assumption. For instance, in Figure 13 of the revised paper, while the overall predictions for the skateboarder’s body are relatively accurate, closer examination reveals that the skateboarder’s hand undergoes noticeable deformations during motion, which clearly violates the rigidity assumption. This results in poor predictions for finer details and highlights the method’s inability to handle non-rigid components effectively in real-world scenarios. Furthermore, in the extrapolation mode, the skateboard itself is not reconstructed, which is a critical failure given its integral role in the motion context. These results cast doubt on the method’s robustness and its applicability beyond datasets that closely conform to the rigid-body assumption.**
> > >
> > > **Response \#1:** We would bring attention to the key facts that:
> > > - First, we **never** assume motion without external forces, but with constant (or constantly changing in third order type) forces. Such examples include falling balls in gravity and all self-propelled objects in the datasets.
> > > - Second, we **never** assume the object's motion is rigid. An object comprises numerous independent particles. A single particle's motion is rigid, but the resulting compounded object motion can be extremely complex. In the datasets, our method can exactly model many self-propelled deformable objects.
> > >
> > > As to Figure 13, it should be noticed that all baselines achieve much worse results than ours. Such a scenarios is extremely challenging due to many potential factors such as limited training views and rapidly changing appearances, instead of being deformable. For example, 4DGS, the powerful baseline recommended by you, totally fails to reconstruct the skater, but our method still achieves reasonable future extrapolation.
> > >
> > > We respect the reviewer's strong desire to see a groundbreaking solution in this field of study. However, a single failure case should not be a reason to deny the value of a method which shows the best results.
> > >
> > > ---
> > >
> > > **Comment \#2: Applicability of the Method and Alternative Approaches -**
> > >
> > > **I remain skeptical about the practicality of extrapolation based on such a strong assumption. In scenarios where this assumption holds true, simpler reconstruction-based editing approaches may achieve similar or even better outcomes. For example, after reconstruction, it is relatively straightforward to calculate motion properties such as velocity and momentum, which can then estimate the approximate future positions of objects.**
> > >
> > > **From the visual results provided by the authors (again, referring to Figure 13), the overall visual quality of the extrapolated objects is very poor, even when compared to what could potentially be achieved using the simpler editing method mentioned above. This raises serious doubts about the significance and practical utility of the extrapolation mode, and I strongly question whether it is meaningful in its current form.**
> > >
> > > **Response \#2:** Regarding the mentioned reconstruction-based editing approach, it is virtually impossible in practice.
> > >
> > > - First, it always requires human knowledge in the loop, because you need to manually segment the interested parts and then apply your personally estimated motion onto them. The underlying physics is never learned, but from user's experience.
> > > - Second, the mentioned approach can only work for linear motions without any rotations. If we need to estimate  rotation information from the calculated velocity, we have to decompose the velocities into object groups and regress the rotation information, which requires accurate motion segmentation. However, as demonstrated by Table 3 and Figures $17\sim 20$, all existing baselines fail to accurately segment objects, let alone making accurate dynamics estimation.
> > >
> > > The reviewer constantly criticizes a single failure case, while ignoring the fact that all existing baselines are much worse than ours on that case.
> > >
> > > Again, we respect the reviewer's opinion, but also question whether such a judgment without considering the current development of the field is valid and professional.

---

> > > > ### Comment · Reviewer_3rJ7 · 2024-12-01
> > > >
> > > > Regarding your emphasis on the experimental comparisons in Figure 13, I have two additional suggestions. First, including visualizations of the boundary frames between interpolation and extrapolation could better illustrate the complexity of extrapolation and help readers understand the challenges in transitioning from known to unknown motion. Second, your emphasis on 4DGS highlights some unexpected results: the significant degradation in both the reconstructed static elements (e.g., the building and ground) and the dynamic components (e.g., the skateboarder and the skateboard) seems inconsistent with 4DGS’s typical performance. This raises questions about whether these results are influenced by experimental settings or implementation details, and further clarification on this point would be helpful.

---

> > > > > ### Author Response · Authors · 2024-12-02
> > > > >
> > > > > **Comment \#4: Regarding your emphasis on the experimental comparisons in Figure 13, I have two additional suggestions. First, including visualizations of the boundary frames between interpolation and extrapolation could better illustrate the complexity of extrapolation and help readers understand the challenges in transitioning from known to unknown motion. Second, your emphasis on 4DGS highlights some unexpected results: the significant degradation in both the reconstructed static elements (e.g., the building and ground) and the dynamic components (e.g., the skateboarder and the skateboard) seems inconsistent with 4DGS’s typical performance. This raises questions about whether these results are influenced by experimental settings or implementation details, and further clarification on this point would be helpful.**
> > > > >
> > > > > **Response \#4:** Thank you very much for the two detailed suggestions.
> > > > >
> > > > > As to the first suggestion, sure, we will add additional visualizations of adjacent frames between interpolation and extrapolation in the next version. We agree that this will clearly highlight the difficulty of extrapolation.
> > > > >
> > > > > Regarding your second comment about the implementation details of 4DGS, we would make the following clarifications:
> > > > >
> > > > > - As shown in Table 2 of our paper, our implementation of 4DGS shows the best performance for novel-view synthesis on the Dynamic Multipart dataset and achieves comparable performances on other datasets as well. Therefore, we are sure there is no issue in our implementation.
> > > > > - In particular, we follow the official configuration of the deformation HexPlane for the real-world dataset, _i.e._, with a resolution of $64\times64\times64\times150$. Then, we train it for 60000 iterations, where the coarse iteration is set as 3000 and we keep densifying the Gaussians till iteration 15000 (this setting exactly follows the official setting). For reference, we train DefGS for 40000 iterations, where the coarse iteration is also 3000 and the identifying iteration is also 15000. All canonical Gaussians are initialized by SfM, which is consistent for all baselines.
> > > > >
> > > > > Originally, we found the official settings for both DefGS and 4DGS fail to reconstruct the challenging skater scene, primarily due to the very violent dynamics and relatively small size of the skater. To tackle this issue, we turn to a gradual feeding strategy for all Gaussian-based models. To be specific, we feed $0\sim0.1$ (virtual) seconds for the first 1000 iterations of dynamic training, and gradually add the succeeding timestamps into the training set, _i.e._, $0\sim0.2$ for 1000 to 2000 iterations, $0\sim0.3$ for 2000 to 3000 iterations, and keep doing so till all training samples are added into the training set. Only in this way, DefGS can achieve successful reconstruction, but 4DGS still fails either with or without this gradual feeding strategy, primarily because:
> > > > >
> > > > > - Since 4DGS uses the HexPlane representation as a deformation field, the grid features are only trained locally (when a Gaussian appears at the grid), making the training signals unstable at earlier iterations for large motions. This will introduce artifacts and influence the reconstruction quality, resulting in unexpected distortions, especially for the novel view interpolation and extrapolation (the visualization in Figure 13 is from a novel view).
> > > > > - 4DGS itself struggles in learning relatively thin parts with large motions, which is also analyzed in the original paper: the performance of 4DGS dramatically falls for Hellwarrior scene compared to other scenes, whereas its concurrent work DefGS achieves much higher performance on that scene. Therefore, the skater scene which has thin parts poses challenges to 4DGS.
> > > > >
> > > > > Overall, we appreciate the reviewer's two suggestions and our concrete implementation details show that the skater scene is truly hard for 4DGS, and actually for all other baselines as well.

---

> > > ### Author Response · Authors · 2024-11-30
> > >
> > > **Comment \#3: Concerns Shared by Other Reviewers -**
> > >
> > > **In addition to my own comments, I note that other reviewers have raised similar concerns. These include:**
> > >
> > > - **Limited Contribution: For example, Reviewer gAFC mentioned that “the novelty of this addition may be somewhat limited.”**
> > >
> > > - **Rationale for Strong Assumptions: Reviewer j1SG stated, “My main concern about this work is the assumption made.”**
> > >
> > > - **Experimental Design and Dataset Choices: Several reviewers, including myself, have highlighted that the dataset and experimental settings may not adequately validate the method’s applicability to more general or realistic scenarios.**
> > >
> > > **These consistent concerns suggest a broader consensus that the work, in its current state, does not sufficiently address its limitations or justify its assumptions.**
> > >
> > > **Response \#3:** Regarding the novelty, the reviewer **gAFC** holds the view on the ground that our method is built on DefGS[1] and thus lacks novelty. However, we would clarify the core differences and our novelty as follows:
> > >
> > > - DefGS focuses on the problem of interpolation, while our method tackles a rather different problem of physics learning and future extrapolation. This means that our learning objectives (*i.e.*, the set of physical parameters) are fundamentally different from DefGS.
> > > - DefGS is just our backbone network, not our contribution. In our rebuttal materials (Response A1 for **gAFC**, Response A1 for you), we have clearly demonstrated that our method can adopt another backbone 4DGS[2]. This means that downplaying our novelty grounding on the used backbone is unfair.
> > > - Lastly, our method clearly outperforms all baselines by large margins on 5 datasets for accurate future extrapolation and motion segmentation, showing the superiority of our method.
> > >
> > > Regarding the assumption, please refer to our above Response \#1.
> > >
> > > Regarding the experiments and datasets, we exactly follow the established experimental settings on public or our newly collected datasets in the community, conducting an extensive and adequate assessment on five datasets, ultimately achieving the state-of-the-art performance for future frame extrapolation on various general and realistic 3D dynamic scenes.
> > >
> > > Overall, we respect the reviewer's opinions, but an unbiased judgment which fully takes into account the current literature of 3D physics learning should be more beneficial to the field of study.
> > >
> > >
> > > [1] Ziyi Yang, Xinyu Gao, Wen Zhou, Shaohui Jiao, Yuqing Zhang, and Xiaogang Jin. Deformable 3D Gaussians for High-Fidelity Monocular Dynamic Scene Reconstruction. CVPR, 2024.
> > >
> > > [2] Guanjun Wu, Taoran Yi, Jiemin Fang, Lingxi Xie, Xiaopeng Zhang, Wei Wei, Wenyu Liu, Qi Tian, and Xinggang Wang. 4d gaussian splatting for real-time dynamic scene rendering. In Proceedings of the IEEE/CVF Conference on Computer Vision and Pattern Recognition (CVPR), 2024.

---

> > > > ### Comment · Reviewer_3rJ7 · 2024-12-01
> > > >
> > > > Dear Authors,
> > > >
> > > > Thank you for your detailed response to my previous comments. I appreciate the substantial effort you have devoted to addressing the reviewers’ feedback and improving the manuscript. However, I must point out that the revisions and your responses have not fully resolved my primary concerns, which are centered around the following points: (1) the validity of the physical assumptions underlying your method, and (2) the representativeness of the datasets used to evaluate the effectiveness and generalizability of the proposed method, and (3) the limited contribution of the proposed method. Below, I provide a detailed discussion of your responses.
> > > >
> > > > ---
> > > >
> > > > 1. On the Physical Assumptions
> > > >
> > > > You stated that “we never assume motion without external forces.” However, this is inconsistent with the original manuscript (Lines 219–221), where it is explicitly stated: “Here we make an assumption that there is no additional force involved after t = 0.” While this statement has been removed in the revised version, and you clarified that the assumption is now one of “constant force” rather than “no additional force,” it suggests that the method’s motivation and foundation are still fundamentally rooted in this assumption.
> > > >
> > > > While I acknowledge your clarification that the method assumes rigidity at the level of the Gaussian primitives rather than the entire object. However, from a physical perspective, if the components of an object are rigidly connected and subjected to a constant force, the overall motion is typically expected to be rigid as well. This is consistent with your experimental results, where the method performs well under scenarios of overall rigid motion but struggles significantly in cases of non-rigid motion, such as those illustrated in Figure 13 (a scenario that is notably different from other dataset in the paper and more representative of real-world conditions). This suggests that while the method explicitly models local rigidity at the component level, its practical effectiveness appears more aligned with an implicit assumption of overall rigidity.
> > > >
> > > > If the goal of the method is to generalize beyond rigid motion, the reliance on this implicit assumption under constant force significantly limits its applicability to more complex, real-world motion scenarios.
> > > >
> > > > ---
> > > >
> > > > 2. On the Significance and Implementation of Extrapolation
> > > >
> > > > In my previous comments, I suggested that under the strong physical priors assumed in your method, reconstruction followed by editing could achieve extrapolation over short time windows. While this may not be the optimal solution, it could still perform well given the datasets and experimental settings in the paper.
> > > >
> > > > The proposed strategy does not rely on manual segmentation and more human knowledge than your method. Instead, it leverages the learned 4D representations to estimate the motion attributes of individual Gaussian primitives. These representations inherently model the motion trajectories of Gaussians in canonical space, making it possible to extrapolate motion without additional segmentation or human intervention.
> > > >
> > > > However, since the revised version has de-emphasized these physical assumptions, it is crucial to demonstrate the effectiveness of your method in more practical, real-world scenarios. Currently, the experiments are constrained by datasets with limited temporal frames and overly simple motion patterns. If your method can demonstrate satisfactory performance on more diverse and realistic datasets, such as the two I suggested, I would be more inclined to reconsider my evaluation.
> > > >
> > > > ---
> > > >
> > > > 3. On Feedback from Other Reviewers
> > > >
> > > > While all reviewers, myself included, acknowledge the substantial effort and extensive work you have invested in this paper, there remains a shared concern regarding the method’s contribution and novelty. The current experimental design and results do not adequately establish the method’s generalizability. Although you have constructed a new dataset, the scenarios included are overly simplistic and fail to significantly enhance the diversity of the experiments.
> > > >
> > > > ---
> > > >
> > > > In summary, I commend your dedication to improving the manuscript and your thoughtful responses to the reviewers’ feedback. I have also invested significant time and effort into carefully reading, analyzing, and reflecting on your paper and its revisions, as I genuinely hope to see this work evolve into a more complete and robust contribution.
> > > >
> > > > Despite the extensive experiments, my concerns remain regarding the method’s reliance on above-mentioned implicit assumptions, the limited representativeness of the datasets, and the lack of a significant contribution over existing methods. If you can further demonstrate that your method is not constrained by these implicit assumptions and represents a generalizable approach to learning motion dynamics, with meaningful extrapolation results in real-world scenarios, I would be happy to reassess the paper and consider raising my score.

---

> > > > > ### Author Response · Authors · 2024-12-02
> > > > >
> > > > > **Comment \#2: On the Significance and Implementation of Extrapolation**
> > > > >
> > > > > **In my previous comments, I suggested that under the strong physical priors assumed in your method, reconstruction followed by editing could achieve extrapolation over short time windows. While this may not be the optimal solution, it could still perform well given the datasets and experimental settings in the paper.**
> > > > >
> > > > > **The proposed strategy does not rely on manual segmentation and more human knowledge than your method. Instead, it leverages the learned 4D representations to estimate the motion attributes of individual Gaussian primitives. These representations inherently model the motion trajectories of Gaussians in canonical space, making it possible to extrapolate motion without additional segmentation or human intervention.**
> > > > >
> > > > > **However, since the revised version has de-emphasized these physical assumptions, it is crucial to demonstrate the effectiveness of your method in more practical, real-world scenarios. Currently, the experiments are constrained by datasets with limited temporal frames and overly simple motion patterns. If your method can demonstrate satisfactory performance on more diverse and realistic datasets, such as the two I suggested, I would be more inclined to reconsider my evaluation.**
> > > > >
> > > > > **Response \#2:** First, we appreciate the reviewer for suggesting a potential method \``estimating motion attributes or trajectories, making it possible to extrapolate", yet the core issue is _what attributes to learn and how to extrapolate_. In fact, our method can be regarded as one instantiation of such a pipeline: ``estimating motion attributes (translation rotation parameters), followed by second- or third- order extrapolation via our Equation (4)".
> > > > >
> > > > > Second, regarding the assumption, again, we hope to reach a consensus that, as shown in Equation (4), our basic physical assumption is up to a second-order relationship, and it can be easily extended to a third-order relationship.
> > > > >
> > > > > Third, regarding the experiments, here is the summary: we evaluate our method on five synthetic and real-world datasets comprising multiple rigid and deformable scenarios. Exactly following the fair and extensive evaluation protocols established by baselines in the community, our method clearly achieves the best performance in future frame extrapolation and motion segmentation compared to all existing works.
> > > > >
> > > > > Nevertheless, the field of 3D physical learning is still in its infancy, and the benchmarking datasets are yet to be large-scale and diverse, ideally being similar as ImageNet for image classification in the coming years. Apparently, such a desired goal shared by the reviewer is hardly achievable in a single paper like ours. We respectfully hope the reviewer reconsiders your evaluation from the actual progress of the specific field of study.

---

> > > > > ### Author Response · Authors · 2024-12-02
> > > > >
> > > > > **Comment \#3: On Feedback from Other Reviewers**
> > > > >
> > > > > **While all reviewers, myself included, acknowledge the substantial effort and extensive work you have invested in this paper, there remains a shared concern regarding the method’s contribution and novelty. The current experimental design and results do not adequately establish the method’s generalizability. Although you have constructed a new dataset, the scenarios included are overly simplistic and fail to significantly enhance the diversity of the experiments.**
> > > > >
> > > > > **In summary, I commend your dedication to improving the manuscript and your thoughtful responses to the reviewers’ feedback. I have also invested significant time and effort into carefully reading, analyzing, and reflecting on your paper and its revisions, as I genuinely hope to see this work evolve into a more complete and robust contribution.**
> > > > >
> > > > > **Despite the extensive experiments, my concerns remain regarding the method’s reliance on above-mentioned implicit assumptions, the limited representativeness of the datasets, and the lack of a significant contribution over existing methods. If you can further demonstrate that your method is not constrained by these implicit assumptions and represents a generalizable approach to learning motion dynamics, with meaningful extrapolation results in real-world scenarios, I would be happy to reassess the paper and consider raising my score.**
> > > > >
> > > > > **Response \#3:** We highly appreciate the reviewer's time and effort on our paper over the past weeks. Your insightful comments and thought-provoking questions have significantly improved our manuscript.
> > > > >
> > > > > We also thank all four responsible and professional reviewers for acknowledging our substantial efforts invested in this paper. Like all researchers, we hope that our own contributions - namely, the neat idea of learning physical parameters and two new datasets - could be truly valued by reviewers.
> > > > >
> > > > > Lastly, regarding the mentioned implicit assumption, to be short, we do not apply an implicit assumption of overall rigidity, because we do not assume that neighboring particles should predict the same physical parameters in our design, but leaving the network to learn per-particle physical parameters separately from RGB images, thus adapting to either rigid or non-rigid motions automatically. Ultimately, the learned per-particle physical parameters enable meaningful future frame extrapolation.

---

> ### Author Response · Authors · 2024-12-02
>
> **Comment \#1: On the Physical Assumptions**
>
> **You stated that “we never assume motion without external forces.” However, this is inconsistent with the original manuscript (Lines 219–221), where it is explicitly stated: “Here we make an assumption that there is no additional force involved after t = 0.” While this statement has been removed in the revised version, and you clarified that the assumption is now one of “constant force” rather than “no additional force,” it suggests that the method’s motivation and foundation are still fundamentally rooted in this assumption.**
>
> **While I acknowledge your clarification that the method assumes rigidity at the level of the Gaussian primitives rather than the entire object. However, from a physical perspective, if the components of an object are rigidly connected and subjected to a constant force, the overall motion is typically expected to be rigid as well. This is consistent with your experimental results, where the method performs well under scenarios of overall rigid motion but struggles significantly in cases of non-rigid motion, such as those illustrated in Figure 13 (a scenario that is notably different from other dataset in the paper and more representative of real-world conditions). This suggests that while the method explicitly models local rigidity at the component level, its practical effectiveness appears more aligned with an implicit assumption of overall rigidity.**
>
> **If the goal of the method is to generalize beyond rigid motion, the reliance on this implicit assumption under constant force significantly limits its applicability to more complex, real-world motion scenarios.**
>
> **Response \#1:** Thank you for sharing your points, and we would make the following clarifications as brief as possible to save your precious time.
>
> Regarding the assumption, it appears that the core issue is caused by different interpretations of the language term ``no additional force" used before, but it's now removed to avoid confusion in our new version. After two rounds of discussion, we hope to reach a consensus that, as shown in Equation (4), our basic physical assumption is up to a second-order relationship, and it can be easily extended to a third-order relationship (Response A3 to **j1SG**, Response A3 to you, Response A7 to **i7XQ**). This clearly means that our assumption is **constant (or constantly changing in third order type) forces**.
>
> Regarding the local rigidity, by treating each particle separately, our method does not apply an implicit assumption of overall rigidity, because we do not assume that neighboring particles should predict the same physical parameters in our design, unless the network learns to achieve so just from training images by itself. And it can achieve so, as verified on scenarios of rigid motions acknowledged by the reviewer. In addition to rigid motions, as demonstrated on many self-propelled deformable objects, our method can indeed effectively learn non-rigid motions, thanks to our neat per-particle physical formulation.
>
> Regarding the particularly challenging scenario of Figure 13, we acknowledge that all methods achieve less satisfactory results, though ours is still better. For any newly designed algorithm, failure cases are always unavoidable, which typically moves the ball forward and inspires more effective future work. We hope the reviewer could reweight the results of Figure 13.

---

### Official Review · Reviewer_j1SG · 2024-11-04

**Soundness:** 3
**Presentation:** 3
**Contribution:** 3
**Rating:** 6
**Confidence:** 3

**Summary:**

The authors extend multi-view dynamical scene modeling by predicting motion physics parameters without additional supervision. Specifically, they directly predict a translation rotation dynamics system for each 3D particle, which gives the model capabilities in future predictions of trajectories and rigid part discovery via clustering. Quantitative and qualitative results show superior performance against prior arts on three existing and one proposed benchmarks.

**Strengths:**

[+] The paper is well-organized.

[+] The proposed methodology of predicting translation rotation dynamics is straight-forward and well-presented.

[+] The emerged behavior of rigid parts through motion clustering is interesting and show be highlighted further.

[+] Extensive empirical evaluation on multiple benchmarks demonstrates superior performance, along with proper ablation study and demo video in supplementary.

**Weaknesses:**

[-] My main concern about this work is the assumption made (L219) that "there is no additional force involved after $t=0$." Although the author give a justification that "a rolling ball suddenly exploding is not learnable," I am not sure if the scope of the research is sufficiently broad given this constraint:
- First, while some moveable objects cannot move of their own volition, many dynamical (interesting) objects do have the ability to move on their own (e.g. humans, vehicles, animals, etc). By assuming no additional forces after $t=0$, the formulation assumes the presence of no dynamical objects, which conflicts with some of the qualitative results (whale, skater and van). Are we simply modeling these objects in a time window where no force is applied? It would be great if the authors can clarify on how the assumption impacts the modeling of self-propelled objects.
- Second, due to the strict assumption made about applied forces, the dynamical scene valid for this method would be rather simple and cannot contain more complex motion with evolving accelerations. The authors should elaborate on the types of motion that can / cannot be handled by GVFi.
- Finally, since I do not work on this topic, I am not sure how significant is my concern above and I am happy to change my recommendation as I await to read other reviewer’s comments and the author's response to my review.

**Questions:**

Please refer to weaknesses above.

---

> ### Author Response · Authors · 2024-11-25
>
> We appreciate the reviewer's valuable comments and address the concerns below.
>
> **Q1: My main concern about this work is the assumption made (L219) that "there is no additional force involved after $t=0$." Although the author give a justification that "a rolling ball suddenly exploding is not learnable," I am not sure if the scope of the research is sufficiently broad given this constraint.**
>
> **A1:** Thank you for pointing out the inaccurate descriptions about our assumption in the original paper.
>
> For clarification, our scheme in Equation 4 follows a second-order relationship to update dynamics parameters for each rigid particle from time $0$ to $t$. It captures up to a constant acceleration from $0$ to $t$, meaning that forces are indeed allowed to generate accelerations. More explanations are in the following response A2.
>
> In the revised paper, we have rephrased the descriptions in lines 235-254 of Section 3.2.
>
> **Q2: First, while some moveable objects cannot move of their own volition, many dynamical (interesting) objects do have the ability to move on their own (e.g. humans, vehicles, animals, etc). By assuming no additional forces after $t=0$, the formulation assumes the presence of no dynamical objects, which conflicts with some of the qualitative results (whale, skater and van). Are we simply modeling these objects in a time window where no force is applied? It would be great if the authors can clarify on how the assumption impacts the modeling of self-propelled objects.**
>
> **A2:** We appreciate this thought-provoking comment. Theoretically, our updating scheme in Equation 4 can be naturally extended to higher orders or reduced to lower orders with regard to future time $t$. Intuitively, a higher order relationship from time $0$ to $t$ is expected to capture extremely complex dynamics such as self-propelled objects.
>
> In the paper, one reason for choosing the second-order scheme to update dynamics parameters is that: In many applications such as robot manipulation, the need for future prediction typically involves a relatively short interval, *i.e.*, $|t-0|$ is rather small, *e.g.*, in milliseconds. In this case, a second-order relationship is usually sufficient to achieve decent approximations.
>
> In addition, as suggested by the reviewer, a simple sliding window based approach can be applied to continuously and incrementally predict future frames given the newest visual observations from sensors, such that the dynamics of self-propelled objects can be well-captured.
>
> To validate this, we conduct experiments for incremental learning on three self-propelled objects from the Dynamic Object Dataset. To be specific, we first feed time $t=0\sim 0.15$ to train the network, and evaluate novel view interpolation on $t=0\sim 0.15$,  future frame extrapolation on $t=0.15\sim 0.30$. Next, we include $t=0.15\sim 0.30$ to train, and evaluate novel view interpolation on $t=0\sim 0.30$, future frame extrapolation on $t=0.30\sim 0.45$. We keep adding a time interval of 0.15 till we train from $t=0\sim 0.75$, and extrapolate from $t=0.75\sim 0.9$.
>
> The following Table (Table 5 in revised paper) shows quantitative results. It can be seen that DefGS suffers from overfitting the previous timestamps and  its interpolation performance decreases, while our model can stably adapt to new observations and achieve excellent past and future frame predictions. This means that even though the internal forces are changing for self-propelled objects, our model can easily adapt to new observations.
>
> More explanations are in the following response A3. Details of the new incremental experiments are in Appendix A.5 in the revised paper.
>
> **Table:** *Quantitative results (PSNR) of incremental learning.*
> | Interpolation | $0.15\rightarrow0.30$ | $0.30\rightarrow0.45$ | $0.45\rightarrow0.60$ | $0.60\rightarrow0.75$ | $0.75\rightarrow0.90$ | Average |
> |---:|:---:|:---:|:---:|:---:|:---:|:---:|
> | DefGS | 39.386 | 38.745 | 35.818 | 34.531 | 27.904 | 35.277 |
> | **GVFi (Ours)** | 40.032 | 40.706 | 41.013 | 40.466 | 39.971 | 40.438 |
>
> | Extrapolation | $0.15\rightarrow0.30$ | $0.30\rightarrow0.45$ | $0.45\rightarrow0.60$ | $0.60\rightarrow0.75$ | $0.75\rightarrow0.90$ | Average |
> |---:|:---:|:---:|:---:|:---:|:---:|:---:|
> | DefGS | 23.438 | 21.360 | 19.989 | 19.670 | 17.629 | 20.417 |
> | **GVFi (Ours)** | 29.958 | 32.260 | 31.384 | 29.527 | 28.958 | 30.417 |

---

> > ### Author Response · Authors · 2024-11-25
> >
> > **Q3: Second, due to the strict assumption made about applied forces, the dynamical scene valid for this method would be rather simple and cannot contain more complex motion with evolving accelerations. The authors should elaborate on the types of motion that can \/ cannot be handled by GVFi.**
> >
> > **A3:** Thanks for this helpful suggestion. To validate the effectiveness of our method on more complex motions with evolving accelerations, as also requested by the reviewer **i7XQ**, we further conduct ablation experiments for choosing first-/third- order relationships in our Equation 4 on Dynamic Object Dataset and Dynamic Multipart Dataset.
> >
> > The following Table (Table 8 in revised paper) shows the results. We can see that, in Dynamic Object Dataset which has several self-propelled objects whose internal forces tend to change over time, not surprisingly, the third-order variant performs better. Nevertheless, due to the inherent over-parametrization, the third-order scheme tends to learn excessive rotation information to represent simple acceleration motions, thus incurring inferior performance on the Dynamic Multipart Dataset which does not have self-propelled objects.
> >
> > Overall, it is indeed interesting yet non-trivial to learn much higher-order relationships and we leave it for future exploration.
> >
> > In the revised paper, we have clarified the descriptions in lines 235-254 of Section 3.2, and added the new first-/third- order ablations in Table 8 of Appendix A.8.
> >
> > **Table:** *Quantitative results of ablation studies about 3 orders of Taylor expansion on Dynamic Multipart dataset and Dynamic Object Dataset.*
> > |  | **Dynamic Multipart Dataset** |  |  |  |  |  | **Dynamic Object Dataset** |  |  |  |  |  |
> > |---|:---:|:---:|:---:|:---:|:---:|:---:|:---:|:---:|:---:|:---:|:---:|:---:|
> > |  |  | Interpolation |  |  | Extrapolation |  |  | Interpolation |  |  | Extrapolation |  |
> > |  | PSNR$\uparrow$ | SSIM$\uparrow$ | LPIPS$\downarrow$ | PSNR$\uparrow$ | SSIM$\uparrow$ | LPIPS$\downarrow$ | PSNR$\uparrow$ | SSIM$\uparrow$ | LPIPS$\downarrow$ | PSNR$\uparrow$ | SSIM$\uparrow$ | LPIPS$\downarrow$ |
> > | $1^{st}$-order | 34.776 | 0.990 | 0.013 | 26.729 | 0.976 | 0.018 | 38.892 | **0.995** | **0.005** | 28.536 | **0.983** | 0.012 |
> > | $2^{nd}$-order | 34.807 | **0.991** | **0.011** | **30.721** | **0.986** | **0.012** | 38.788 | **0.995** | 0.006 | 28.758 | 0.982 | **0.011** |
> > | $3^{rd}$-order | **35.268** | **0.991** | 0.012 | 30.503 | 0.985 | 0.013 | **39.164** | **0.995** | **0.005** | **29.378** | **0.983** | **0.011** |
> >
> > **Q4: Finally, since I do not work on this topic, I am not sure how significant is my concern above and I am happy to change my recommendation as I await to read other reviewer’s comments and the author's response to my review.**
> >
> > **A4:** Thank you for your willingness to discuss. Your concerns are very insightful and have significantly helped us to improve the quality of our revised paper. To sum up, in the revised paper, we have clarified the motivations of choosing a second-order updating scheme and the scopes of our method. Most notably, we have further conducted experiments about the suggested incremental experiments, and the ablation of first-/third- order updating schemes.

---

### Official Review · Reviewer_gAFC · 2024-11-05

**Soundness:** 3
**Presentation:** 3
**Contribution:** 3
**Rating:** 6
**Confidence:** 3

**Summary:**

This paper introduces GVFi, a novel approach for modeling 3D scene geometry, appearance, and dynamics from multi-view images without the need for human annotations, such as bounding boxes or segmentations. The authors highlight that previous 3D Gaussian Splatting models struggled to capture the underlying motion physics of dynamic scenes. In contrast, GVFi treats 3D points as particles in space, each with a learnable size and orientation, enabling the model to learn particle rotation and translation to represent a dynamic system effectively. Experimental results on three diverse datasets show that GVFi significantly outperforms prior 3D Gaussian Splatting models on both interpolation and extrapolation tasks.

**Strengths:**

1. It is novel to represent the 3D points as particles, which is a well-established concept in robotics. This representation could open up further research topics to improve dynamics modeling.
2. This model does not rely on human annotations for motion estimation. It can autonomously group meaningful objects based on motion patterns without requiring any labeled data.
3. The authors provide both quantitative and qualitative results across multiple datasets, demonstrating GVFi’s improvements in both interpolation and extrapolation tasks.

**Weaknesses:**

1. This model builds upon DefGS (Yang et al., 2024), with its main contribution being the translation-rotation dynamics system module. However, the novelty of this addition may be somewhat limited.
2. The performance of DefGS (Yang et al., 2024) and GVFi is quite similar, and there appears to be no significant visual difference between the outputs of the two models. Could the authors clarify specific scenarios where the translation-rotation dynamics system module leads to performance improvements?
3. There are no quantitative results for object segmentation. Would it be possible to evaluate this and compare it to models that rely on human annotations?

**Questions:**

1. The performance and visual results of DefGS and GVFi appear very similar. Could the authors specify scenarios where the translation-rotation dynamics module offers clear advantages?
2. Could quantitative results for object segmentation be provided, and how does GVFi compare to models that rely on human annotations for this task?
3. Could the authors highlight the novelty compare to DefGS?

---

> ### Author Response · Authors · 2024-11-25
>
> We appreciate the reviewer's valuable comments and address the concerns below.
>
> **Q1: This model builds upon DefGS (Yang et al., 2024), with its main contribution being the translation-rotation dynamics system module. However, the novelty of this addition may be somewhat limited.**
>
> **A1:** For clarification, our core novelty is the introduced translation rotation dynamics system together with its effective optimization strategy, which allows us to truly learn physical parameters, ultimately achieving future frame extrapolation. By comparison, existing works such as DefGS/4DGS[1] all fail to do so, fundamentally because they do not learn underlying physics priors, though they perform well for past frame interpolation, as extensively verified in Tables 1\&2 in our paper.
>
> In addition, the use of DefGS as our auxiliary deformation field is actually not our novelty. In fact, our introduced translation rotation dynamics system is also amenable to other deformation fields such as 4DGS[1], achieving satisfactory performance as shown in the following table (hyperparameters not tuned due to the limited time for rebuttal).
>
> To the best of our knowledge, we are the first to learn such a translation rotation dynamics system for modeling dynamic 3D scenes in literature, and we achieve state-of-the-art performance for future frame extrapolation on five datasets. This clearly demonstrates our significant novelty in the field of study.
>
> In the revised paper, we highlight our novelty in lines 93-99 of Section 1.
>
> **Table:** _Quantitative results of our method with 4DGS as the auxiliary deformation field on four datasets._
> |  | **Dynamic Multipart Dataset** |  |  |  |  |  | **Dynamic Object Dataset** |  |  |  |  |  |
> |---|:---:|:---:|:---:|:---:|:---:|:---:|:---:|:---:|:---:|:---:|:---:|:---:|
> |  |  | Interpolation |  |  | Extrapolation |  |  | Interpolation |  |  | Extrapolation |  |
> |  | PSNR$\uparrow$ | SSIM$\uparrow$ | LPIPS$\downarrow$ | PSNR$\uparrow$ | SSIM$\uparrow$ | LPIPS$\downarrow$ | PSNR$\uparrow$ | SSIM$\uparrow$ | LPIPS$\downarrow$ | PSNR$\uparrow$ | SSIM$\uparrow$ | LPIPS$\downarrow$ |
> | GVFi$_{4dgs}$ | **36.542** | **0.991** | 0.015 | **30.801** | _0.983_ | _0.016_ | 35.961 | 0.985 | 0.021 | _28.316_ | _0.978_ | _0.023_ |
> | GVFi | 34.807 | **0.991** | **0.011** | _30.721_ | **0.986** | **0.012** | **38.788** | **0.995** | **0.006** | **28.758** | **0.982** | **0.011** |
> |  | **Dynamic Indoor Scene Dataset** |  |  |  |  |  | **NVIDIA Dynamic Scenes Dataset** |  |  |  |  |  |
> |  |  | Interpolation |  |  | Extrapolation |  |  | Interpolation |  |  | Extrapolation |  |
> |  | PSNR$\uparrow$ | SSIM$\uparrow$ | LPIPS$\downarrow$ | PSNR$\uparrow$ | SSIM$\uparrow$ | LPIPS$\downarrow$ | PSNR$\uparrow$ | SSIM$\uparrow$ | LPIPS$\downarrow$ | PSNR$\uparrow$ | SSIM$\uparrow$ | LPIPS$\downarrow$ |
> | GVFi$_{4dgs}$ | 27.932 | 0.860 | 0.252 | _31.590_ | _0.909_ | _0.194_ | 18.995 | 0.448 | 0.544 | _22.706_ | _0.714_ | _0.400_ |
> | GVFi | **32.202** | **0.928** | **0.089** | **34.556** | **0.964** | **0.046** | **26.943** | **0.891** | **0.102** | **29.388** | **0.938** | **0.067** |
>
> **Q2: the performance of DefGS (Yang et al., 2024) and GVFi is quite similar, and there appears to be no significant visual difference between the outputs of the two models. Could the authors clarify specific scenarios where the translation-rotation dynamics system module leads to performance improvements?**
>
> **A2:** As shown in Tables 1\&2 in our main paper, the performance of DefGS actually lags far behind our method for future frame extrapolation on all datasets. Particularly, our method has 10 points higher on PSNR than DefGS on three datasets (Dynamic Object / Dynamic Indoor Scene / Dynamic Multipart), and 5 points higher on PSNR on two datasets (NVIDIA Dynamic Scene, and our newly collected real-world GoPro dataset).
>
> As discussed in above A1, DefGS fundamentally cannot predict future frames because it does not learn physics priors, though it can achieve good performance for past frame interpolation.
>
> [1] Guanjun Wu, Taoran Yi, Jiemin Fang, Lingxi Xie, Xiaopeng Zhang, Wei Wei, Wenyu Liu, Qi Tian, and Xinggang Wang. 4d gaussian splatting for real-time dynamic scene rendering. In Proceedings of the IEEE/CVF Conference on Computer Vision and Pattern Recognition (CVPR), 2024.

---

> ### Author Response · Authors · 2024-11-25
>
> **Q3: There are no quantitative results for object segmentation. Would it be possible to evaluate this and compare it to models that rely on human annotations?**
>
> **A3:** Thanks for this valuable suggestion. As requested, we include extensive quantitative results on the Dynamic Indoor Scene dataset in Table 3 of Section 4.2 in our revised paper.
>
> In particular, we follow Gaussian Grouping [2] to render 2D object segmentation masks for all 30 views over 60 timestamps on all 4 scenes, _i.e._, 7200 images in total. We compare with **D-NeRF**, **NVFi**, **DefGS** and **DefGS$_{nvfi}$**. We follow NVFi to obtain segmentation results of D-NeRF and NVFi. For the 3DGS-based baselines, we also adopt OGC [3] to segment Gaussians based on scene flows induced from their learned deformation fields. All implementation details are in Appendix. Additionally, we include a strong image-based 2D object segmentation method, Mask2Former [4] pre-trained by human annotations on COCO dataset [5] as a fully-supervised baseline.
>
> As shown in the following Table (Table 3 in revised paper), our method achieves almost perfect object segmentation results on all metrics, significantly outperforming all baselines. This shows that our learned physical parameters correctly model object physical motion patterns and can be easily leveraged to identify individual objects according to their motions, without needing any human annotations.
>
> **Table:** _Quantitative results of motion segmentation results on Dynamic Indoor Scene dataset._
> |  | AP$\uparrow$ | PQ$\uparrow$ | F1$\uparrow$ | Pre$\uparrow$ | Rec$\uparrow$ | mIoU$\uparrow$ |
> |---:|:---:|:---:|:---:|:---:|:---:|:---:|
> | Mask2Former [4] | 65.37 | 73.14 | 78.29 | _94.83_ | 68.88 | 64.42 |
> | D-NeRF | 57.26 | 46.15 | 59.02 | 56.55 | 62.94 | 46.58 |
> | NVFi | _91.21_ | _78.74_ | _93.75_ | 93.76 | _93.74_ | _67.64_ |
> | DefGS | 51.73 | 57.60 | 66.43 | 63.21 | 70.07 | 54.46 |
> | DefGS$_{nvfi}$ | 55.26 | 62.75 | 69.83 | 69.39 | 72.91 | 56.82 |
> | GVFi (Ours) | **95.82** | **93.28** | **97.90** | **96.21** | **99.86** | **79.55** |
>
> **Q4: Questions 1) The performance and visual results of DefGS and GVFi appear very similar. Could the authors specify scenarios where the translation-rotation dynamics module offers clear advantages?**
>
> **A4:** Refer to A1 and A2.
>
> **Q5: Questions 2) Could quantitative results for object segmentation be provided, and how does GVFi compare to models that rely on human annotations for this task?**
>
> **A5:** Refer to A3.
>
> **Q6: Questions 3) Could the authors highlight the novelty compare to DefGS?**
>
> **A6:** Refer to A1.
>
> [2] Mingqiao Ye, Martin Danelljan, Fisher Yu, and Lei Ke. Gaussian grouping: Segment and edit anything in 3d scenes. In ECCV, 2024.
>
> [3] Ziyang Song and Bo Yang. OGC: Unsupervised 3D Object Segmentation from Rigid Dynamics of Point Clouds. NeurIPS, 2022
>
> [4] Bowen Cheng, Ishan Misra, Alexander G. Schwing, Alexander Kirillov, and Rohit Girdhar. Masked-attention Mask Transformer for Universal Image Segmentation. CVPR, 2022
>
> [5] Tsung-Yi Lin, Michael Maire, Serge Belongie, Lubomir Bourdev, Ross Girshick, James Hays, Pietro Perona, Deva Ramanan, C. Lawrence Zitnick, and Piotr Doll´ar. Microsoft COCO: Common Objects in Context. ECCV, 2014.

---

> > ### Comment · Reviewer_gAFC · 2024-11-28
> >
> > Thank the authors for their valuable feedback and for addressing my questions. While my concerns have been resolved, I share the generalization and novelty concerns raised by reviewers 3rJ7, i7XQ, and j1SG. I will maintain my score as a borderline accept.

---

> ### Author Response · Authors · 2024-11-30
>
> **Comment: Thank the authors for their valuable feedback and for addressing my questions. While my concerns have been resolved, I share the generalization and novelty concerns raised by reviewers 3rJ7, i7XQ, and j1SG. I will maintain my score as a borderline accept.**
>
> **Response:** Thank you very much for acknowledging that your concerns have been addressed, and maintaining your positive score. We would make further clarifications.
>
> Regarding our novelty, the reviewers hold their view on the ground that our method is built on DefGS[1] and thus lacks novelty. However, we would clarify the core differences and our novelty as follows:
> -  DefGS focuses on the problem of interpolation, while our method tackles a rather different problem of physics learning and future extrapolation. This means that our learning objectives (*i.e.*, the set of physical parameters) are fundamentally different from DefGS.
> - DefGS is just our backbone network, not our contribution. In our rebuttal materials (Response A1 for you, Response A1 for **3rJ7**), we have clearly demonstrated that our method can adopt another backbone 4DGS[2]. This means that downplaying our novelty grounding on the used backbone is unfair.
> - Lastly, our method clearly outperforms all baselines by large margins on 5 datasets for accurate future extrapolation and motion segmentation, showing the superiority of our method.
>
> Regarding the generalization or assumptions, reviewers ignore the fact that:
> - First, we **never** assume motion without external forces, but with constant (or constantly changing in third order type) forces. Such examples include falling balls in gravity and all self-propelled objects in the datasets.
> - Second, we **never** assume the object's motion is rigid. An object comprises numerous independent particles. A single particle's motion is rigid, but the resulting compounded object motion can be extremely complex. In the datasets, our method can exactly model many self-propelled deformable objects.
>
>
>  [1] Ziyi Yang, Xinyu Gao, Wen Zhou, Shaohui Jiao, Yuqing Zhang, and Xiaogang Jin. Deformable 3D Gaussians for High-Fidelity Monocular Dynamic Scene Reconstruction. CVPR, 2024.
>
>  [2] Guanjun Wu, Taoran Yi, Jiemin Fang, Lingxi Xie, Xiaopeng Zhang, Wei Wei, Wenyu Liu, Qi Tian, and Xinggang Wang. 4d gaussian splatting for real-time dynamic scene rendering. In Proceedings of the IEEE/CVF Conference on Computer Vision and Pattern Recognition (CVPR), 2024.

---

### Author Response · Authors · 2024-11-26
**Summary of Updates**

We would like to express our gratitude to all reviewers for your valuable comments, and we have made significant improvements to our paper. Below is a consolidated summary of the changes made.

- **Evaluation on newly captured real-world scenes:** We capture 4 real-world dynamic scenes with 20 GoPro cameras and evaluate our method and baselines on them for future frame extrapolation. The leading performance on these new challenging real-world scenes consolidate the superiority of our method in physics learning.
- **Quantitative evaluation of object segmentation:** We follow NVFi to evaluate the rendered 2D object segmentation masks on Dynamic Indoor Scene dataset, where our method produces almost perfect results and significantly outperforms all baselines.
- **Evaluation with incremental learning:** We conduct incremental learning by taking gradually increasing observation frames for training. Our method can stably adapt to new observations and make reasonable predictions within different observation time window, demonstrating strong potentials to applications like robot planning.
- **Additional baselines \& adaption with baselines:** We add two baseline methods, 4DGS and E-D3DGS, both of which fail to extrapolate future frames as ours. However, a combination of our translation rotation dynamics system with 4DGS shows strong capability in extrapolation, demonstrating the flexibility of our proposed physics learning module.
- **Experiments for long-term extrapolation:** We experiment on longer extrapolation with our model, demonstrating physically meaningful results on all datasets for extremely long-term extrapolation.
- **Additional ablation studies:** We complete ablation studies about $\delta t$ on all datasets. Besides, we ablate learning different orders of velocity representations, where higher-order representations benefit complex motions with evolving accelerations as expected.
- **Experiments with different ways of interpolation:** We experiment with different ways of interpolation within our framework. The similar results to another way consolidates the accurate physical motion representations learned by our method.

Overall, our additional experiments clearly demonstrate the superiority of our method over all baselines on five datasets. We believe that our rebuttal materials adequately address all your concerns. We will release all our code, datasets, trained models to the community.

---

### Meta-Review · Area_Chair_8fDp · 2024-12-21

**Metareview:**

The paper addresses the problem of modeling both 3D scene geometry, appearance, and physical information (this last one being the primary novelty). The paper adopts the DefGS representation for 3D scene geometry, appearance, and motion and then adds a translation-rotation dynamics system module along with an optimization strategy. The authors note that this allows for future frame extrapolation. The paper evaluate on several datasets (Dynamic Object, Dynamic Indoor Scene, NVIDIA Dynamic Scene, Dynamic Multipart, GoPro) and demonstrate strong quantitative results, in particular with respect to the extrapolation setting.

The main strength is the interesting problem setting and the quantitative performance on the extrapolation setting. The main concern from reviewers is the lack of technical novelty, in particular over the DefGS framework used, and the strong assumptions being made for the translation-rotation dynamic system which can limited the applicability of this system - noted by reviewers 3rJ7, i7XQ, and j1SG. None of the reviewers advocated to champion for this paper since they felt that the limited impact and novelty made the paper borderline.

I slightly lean towards rejection, though acceptance as poster would be fine.

**Additional Comments On Reviewer Discussion:**

Reviewer 3rJ7 rated the paper a 3. All other reviewers rated the paper a 6 but none felt it was strong enough to advocate for it. Many reviewers shared similar concerns, limited novelty and limited impact due to the strong assumptions being made. In almost all the author rebuttals, the authors emphasized their extrapolation results (as well as noting that their method works for a different representation (4dgs).

---

### Decision · Program_Chairs · 2025-01-22

Reject

---

> ### Public Comment · ~Bo_Yang7 · 2025-09-14
> **Accepted by ICCV 2025**
>
> The updated version has been published at ICCV 2025. Welcome to check out our paper and code.
>
> Full paper: https://arxiv.org/abs/2508.09811
>
> Code and data: https://github.com/vLAR-group/TRACE